# Investigation of Microstructural Features and Mechanical Characteristics of the Pressureless Sintered B_4_C/C(Graphite) Composites and the B_4_C-SiC-Si Composites Fabricated by the Silicon Infiltration Process

**DOI:** 10.3390/ma15144853

**Published:** 2022-07-12

**Authors:** Tao Jiang

**Affiliations:** School of Materials Science and Engineering, Xi’an Shiyou University, Xi’an 710065, China; jiangtaoxsyu@xsyu.edu.cn; Tel.: +86-134-8458-5478

**Keywords:** composites, hardness, mechanical properties, wear resistance, silicon carbide

## Abstract

The B_4_C/C(graphite) composites were fabricated by employing a pressureless sintering process. The pressureless sintered B_4_C/C(graphite) composites exhibited extremely low mechanical characteristics. The liquid silicon infiltration technique was employed for enhancing the mechanical property of B_4_C/C(graphite) composites. Since the porosity of the B_4_C/C(graphite) composites was about 25–38%, the liquid silicon was able to infiltrate into the interior composites, thereby reacting with B_4_C and graphite to generate silicon carbide. Thus, boron carbide, silicon carbide, and residual silicon were sintered together forming B_4_C-SiC-Si composites. The pressureless sintered B_4_C/C(graphite) composites were transformed into the B_4_C-SiC-Si composites following the silicon infiltration process. This work comprises an investigation of the microstructure, phase composition, and mechanical characteristics of the pressureless sintered B_4_C/C(graphite) composites and B_4_C-SiC-Si composites. The XRD data demonstrated that the pressureless sintered bulks were composed of the B_4_C phase and graphite phase. The pressureless sintered B_4_C/C(graphite) composites exhibited a porous microstructure, an extremely low mechanical property, and low wear resistance. The XRD data of the B_4_C-SiC-Si specimens showed that silicon infiltrated specimens comprised a B_4_C phase, SiC phase, and residual Si. The B_4_C-SiC-Si composites manifested a compact and homogenous microstructure. The mechanical property of the B_4_C-SiC-Si composites was substantially enhanced in comparison to the pressureless sintered B_4_C/C(graphite) composites. The density, relative density, fracture strength, fracture toughness, elastic modulus, and Vickers hardness of the B_4_C-SiC-Si composites were notably enhanced as compared to the pressureless sintered B_4_C/C(graphite) composites. The B_4_C-SiC-Si composites also manifested outstanding resistance to wear as a consequence of silicon infiltration. The B_4_C-SiC-Si composites demonstrated excellent wear resistance and superior mechanical characteristics.

## 1. Introduction

As a promising material for structural applications, Boron carbide (B_4_C) ceramics have been recognized for application in several engineering fields due to their remarkable characteristics [1,2,3]; for instance, high toughness, high strength, high melting point (2450 °C), low density (2.52 g/cm^3^), high Vickers hardness (>30 GPa) and outstanding wear resistance [3,4,5]. Nonetheless, B_4_C ceramics suffer from poor machinability [5,6,7]. The mechanical processing cost of boron carbide ceramics is extremely high [7,8,9,10]. It is thus crucial to develop strategies for the improvement of the machinability of boron carbide-based ceramics [7,8,9,10].

The machinable boron carbide matrix composites can be possibly fabricated by incorporating a machinable phase such as graphite into the boron carbide matrix [1,2,3,4,5]. Graphite materials are low in terms of strength and toughness; however, they exhibit superb thermal shock resistance and exceptional machinability. Following graphite incorporation into boron carbide, it is fabricated into machinable B_4_C/C(graphite) composites [5,6,7,8,9,10]. Hence, a combination of the B_4_C and graphite yields B_4_C/C(graphite) composites [5,6,7,8,9,10]. The B_4_C/C(graphite) composites were fabricated in this work via a pressureless sintering process [5,6,7,8,9,10]. The B_4_C/C(graphite) composites are anticipated to improve the B_4_C ceramics in terms of their thermal shock resistance and machinability [5,6,7,8,9,10].

In one of our previous works, the mechanical property and machinability of the pressureless sintered B_4_C/C(graphite) composites were evaluated. The research findings revealed that the pressureless sintered B_4_C/C(graphite) composites exhibited excellent machinability, while their mechanical characteristics decreased steadily with the rise in graphite content. The surface hardness and wear resistance of the pressureless sintered B_4_C/C(graphite) composites decreased remarkably. Thus, the lack of wear resistance, low surface hardness, and poor mechanical property of the pressureless sintered B_4_C/C(graphite) composites are downsides to their utilization for practical applications. The porosity of the composites was found to be 25–38%. Hence, in an attempt to bring about an improvement in the surface hardness, wear-resistance, and mechanical property of the pressureless sintered B_4_C/C(graphite) composites, a high-temperature liquid silicon infiltration process was employed. Following machining, the pressureless sintered B_4_C/C(graphite) composites were processed via a silicon infiltration process. Following liquid silicon infiltration, the boron carbide and graphite react with the liquid silicon, producing silicon carbide [11,12,13,14,15,16,17,18,19,20]. Because the pressureless sintered B_4_C/C(graphite) composites have a porosity of about 25–38%, the liquid silicon could be conveniently infiltrated into the interior of the composites [11,12,13,14,15,16,17,18,19,20], subsequently reacting with B_4_C and graphite to produce the silicon carbide [20,21,22,23,24,25,26,27,28,29,30]. While the boron carbide reacted only partially [30,31,32,33,34,35,36,37,38,39,40], the graphite reacted completely with silicon producing silicon carbide [40,41,42,43,44,45,46,47,48,49,50]. The residual boron carbide and silicon existed within the matrix of the composite [40,41,42,43,44,45,46,47,48,49,50] and together with silicon carbide gave rise to the B_4_C-SiC-Si composites [40,41,42,43,44,45,46,47,48,49,50,51,52,53,54,55]. The pressureless sintered B_4_C/C(graphite) composites are thus transformed into the B_4_C-SiC-Si composites via the silicon infiltration process [40,41,42,43,44,45,46,47,48,49,50,51,52,53,54,55]. The microstructural details, phase composition, and mechanical characteristics of the B_4_C/C(graphite) composites as well as B_4_C-SiC-Si composites fabricated by the silicon infiltration process were then investigated and compared. We also looked into the mechanism of the reaction between B_4_C/C(graphite) composites and liquid silicon. The density and relative density, fracture strength, fracture toughness, Vickers hardness, wear-resistance, and elastic modulus of the silicon infiltered B_4_C-SiC-Si composites were evaluated and subsequently compared with the pressureless sintered B_4_C/C(graphite) composites. B_4_C-SiC-Si composites, upon silicon infiltration, exhibited a homogenous and compact microstructure, an extremely high relative density, a low porosity, outstanding mechanical characteristics, high surface hardness, and superb wear resistance when compared with the pressureless sintered B_4_C/C(graphite) composites.

In particular, there has been much work using first-principles approaches to understand the strength of these and other materials [56]. In particular, these prior studies show that the density functional theory calculations can be used to understand atomistic details of structural mechanics and strength [57]. These approaches can reveal mechanisms for the strength of structures in a more unbiased way. The first principle calculation played a very important role in studying the relationship between microstructure analysis and mechanical property of the composites materials [56]. At the same time, the density functional calculation played a very important role in studying the relationship between microstructure analysis and the mechanical property of the composite’s materials [57]. The first principle calculation and the density functional calculation could completely understand the relationship between the microstructure characterization and mechanical property of the materials by using the calculation methods [56]. So, it was very important to use the first principle calculation and the density functional calculation to understand the relationship between the microstructure characterization and the mechanical property of the composites materials [57]. According to the principle of interaction between an atomic nucleus and an electron and its basic motion law, using the principle of quantum mechanics, starting from the specific requirements, the algorithm to directly solve the Schrödinger equation after some approximation processing is customarily called the first principle [56]. The density functional theory (DFT) is a method for studying the electronic structure of multi-electron systems. The density functional theory has a wide range of applications in physics and chemistry, especially for studying the properties of molecules and condensed matter. It is one of the most commonly used methods in the fields of computational materials science and computational chemistry in condensed matter physics [57].

## 2. Experimental Procedures

### 2.1. Materials and Fabrication

#### 2.1.1. Fabrication of the B_4_C/C(Graphite) Composites by Pressureless Sintering

A pressureless sintering method was used for fabricating B_4_C/C(graphite) composites. The starting materials included B_4_C powder and graphite powder having average particle dimensions of 3.5 μm and 10–15 μm, respectively. The graphite powder contents of the B_4_C/C(graphite) composite powders were kept as 10 wt%, 20 wt%, 30 wt%, and 40 wt%. In total, 10 wt% sintering aids comprising Y_2_O_3_ and Al_2_O_3_ were then mixed with B_4_C and graphite powders. Subsequently, ethanol was added to the powder mixture and the mixture was ball milled and wet mechanically milled for 24 h and allowed to dry slowly. The B_4_C/C(graphite) composite powders were thus prepared. The powders were then put into the steel moulds and fabricated into the strip samples under 200 MPa pressure, with the holding time being 2 min. The strip sample size was about 5 mm × 6 mm × 50 mm. The strip specimens were put into a graphite crucible and set inside a vacuum sintering furnace. A pressureless sintering process was carried out at 1800 °C for 2 h in vacuum conditions yielding the composite’s bulks based on B_4_C/C(graphite). The B_4_C/C(graphite) composite’s bulks thus produced were cut into strip specimens, with the size of the sample being 5 mm × 6 mm × 50 mm, which were subsequently processed by silicon infiltration.

#### 2.1.2. Fabrication of the B_4_C Monolith by Pressureless Sintering

The B_4_C monolith was fabricated by a pressureless sintering process. For fabricating B_4_C monolith, the raw materials included B_4_C powders having an average particle dimension of 3.5 μm. In total, 10 wt% sintering aids of Al_2_O_3_ and Y_2_O_3_ powder were mixed with B_4_C powders. Ethanol was added into the powdered mixture which was subsequently balled milled and wet mechanically milled for 24 h. Following milling, the powder was subjected to slow drying yielding B_4_C, Al_2_O_3_, and Y_2_O_3_ composite powders. The composite powders were then poured into a steel mould and fabricated into the stripe samples under 200 MPa. The holding time was 2 min, and the strip sample size was about 5 mm × 6 mm × 50 mm. These stripe samples were then put into a graphite crucible and set within a vacuum sintering furnace. The fabrication of B_4_C monolith bulks was achieved using pressureless sintering for 2 h at 1800 °C under vacuum conditions.

#### 2.1.3. Fabrication Process of the B_4_C-SiC-Si Composites by Silicon Infiltration Process

The B_4_C monolith specimens fabricated by pressureless sintering were cut into the strip samples. These strip specimens were coated with silicon powders, placed within a graphite crucible, and set inside a vacuum sintering furnace. The silicon infiltration process was performed for 2 h at 1550 °C under a vacuum. The solid silicon was completely transformed into liquid silicon at 1550 °C (MP of silicon = 1410 °C) during the silicon infiltration process. Thus, the stripe samples were completely immersed in the liquid silicon. Since the porosity of pressureless sintered specimens was about 20–30%, the liquid silicon thoroughly infiltrated into the interior of the specimens. In the process of infiltration, the liquid silicon underwent a reaction with B_4_C, producing silicon carbide and the boron carbide partially reacted with the liquid silicon generating silicon carbide. The residual boron carbide and silicon were found within the matrix of the composite. The boron carbide, silicon carbide, and residual silicon were sintered together and formed the B_4_C-SiC-Si composites. Following the silicon infiltration process, the silicon infiltrated specimens were ground, polished, and fabricated into strip samples of 5 mm × 6 mm × 50 mm. The strip silicon infiltrated specimens were then processed and fabricated into smaller strips with a size of 3 mm × 4 mm × 30 mm. The microstructure and phase composition of the silicon infiltrated specimens were subsequently examined.

The B_4_C/C(graphite) composite samples were fabricated by a pressureless sintering process and then cut into stripe samples. The B_4_C/C(graphite) composites fabricated by a pressureless sintering process have a graphite content of 10 wt%, 20 wt%, 30 wt%, and 40 wt% and were then subjected to a silicon infiltration process by being placed into a vacuum sintering furnace in a graphite crucible. The specimens were covered with silicon powder and the silicon infiltration process was performed for 2 h at 1550 °C under vacuum conditions. The solid silicon was completely transformed into liquid silicon at 1550 °C (MP of silicon = 1410 °C) during the silicon infiltration process. In this way, the stripe samples were completely immersed in the liquid silicon. Since the porosity of pressureless sintered specimens was about 20–30%, the liquid silicon thoroughly infiltrated the interior of the composites. The liquid silicon reacted with B_4_C and graphite producing silicon carbide, the boron carbide partially reacted with the liquid silicon to produce silicon carbide, and the graphite was completely combined with the liquid silicon to produce silicon carbide. The residual boron carbide and silicon, however, were present in the matrix of the composite and sintered together with silicon carbide forming the B_4_C-SiC-Si composites. Following silicon infiltration, the silicon infiltrated specimens were ground, polished, and fabricated first into strip samples of about 5 mm × 6 mm × 50 mm and then cut into smaller sized specimens of 3 mm × 4 mm × 30 mm. The phase composition and microstructure of the B_4_C-SiC-Si composites specimens yielded via silicon infiltration were then investigated.

### 2.2. Characterizations

#### 2.2.1. The Microstructure and Phase Composition of Pressureless Sintered Specimens and Silicon Infiltrated Specimens

The phase composition of the B_4_C/C(graphite) composites fabricated via a pressureless sintering process and that of silicon infiltrated specimens (B_4_C-SiC-Si composites) was ascertained using X-ray diffraction analysis (XRD, D/MAX-2400, Nippon Rigaku Corporation Limited, Tokyo, Japan). The microstructural features of the B_4_C/C(graphite) composites obtained via the pressureless sintering process and that of the silicon infiltrated specimens (B_4_C-SiC-Si composites) were examined and analyzed by a scanning electron microscope (SEM, JSM-6390 A, JEOL Scanning Microscope, Tokyo, Japan). An energy-dispersive spectroscopic analysis (EDS, JEOL Scanning Microscope, Tokyo, Japan) of the silicon infiltrated specimens (B_4_C-SiC-Si composites) was conducted using the scanning electron microscope (SEM, JSM-6390 A, JEOL Scanning Microscope, Tokyo, Japan).

#### 2.2.2. The Mechanical Property and Wear Resistance of Pressureless Sintered Specimens and Silicon Infiltrated Specimens

The strip samples of the B_4_C/C(graphite) composites fabricated by pressureless sintering and the B_4_C-SiC-Si composites by silicon infiltration process were evaluated for their mechanical property, density, relative density, fracture toughness, fracture strength, Vickers hardness, elastic modulus, and wear resistance. Archimedes’ methods were used to estimate the density and relative density of the composites. The composites were evaluated for their fracture strength by a three-point bending test utilizing specimens of sizes 3 mm × 4 mm × 30 mm. The span was kept at 16 mm and the speed of the crosshead was 0.5 mm/min (Testing machine: Instron1195). The fracture toughness of the composites was estimated using a single edge notch beam (SENB) using samples of 3 mm × 4 mm × 30 mm. The respective depth and width of the notch were 1.5 mm and 0.2 mm, the span was 16 mm, and the crosshead speed was 0.05 mm/min (Testing machine: Instron1195). The elastic modulus of the pressureless sintered B_4_C/C(graphite) composites, as well as B_4_C-SiC-Si composites obtained via silicon infiltration, was estimated using a three-point bending test with the dimensions of the specimen being 3 mm × 4 mm × 30 mm, the span and the crosshead speed being 16 mm and 0.5 mm/min (Testing machine: Instron1195), respectively. A Vickers hardness meter equipped with a load force of 49 N and a holding time of 20 s was employed to test the composites for their Vickers’s hardness (HV-5 Vickers hardness meter). The wear experiments were performed at MM-W1 B friction and wear tester. Friction wear tests were carried out to measure the wear resistance of the pressureless sintered B_4_C/C(graphite) composites and B_4_C-SiC-Si composites obtained via silicon infiltration, wherein the load force was 20 N and the rotational speed was 220 rpm. The wear loss rates of pressureless sintered specimens were also measured. The wear loss rate (R) was the ratio of wear weight loss(M) to wear time(t), thus the wear loss rate is computed using the following equation: R = M/t. The mechanical property and wear resistance of the pressureless sintered B_4_C monolith were measured similarly using the methods employed for the pressureless sintered B_4_C/C(graphite) composites.

## 3. Results and Discussion

### 3.1. The Phase Composition of the B_4_C/C(Graphite) Composites Fabricated by the Pressureless Sintering Process

#### 3.1.1. The Phase Composition of the B_4_C Monolith Fabricated Using a Pressureless Sintering Process

The XRD patterns of the B_4_C monolith bulks produced by pressureless sintering have been shown in Figure 1. For the pressureless sintered B_4_C monolith, Figure 1 illustrates that the diffraction peaks corresponding to the B_4_C phase were visible in the observed XRD data, at 19.71°, 22.02°, 23.50°, 31.90°, 34.95°, and 37.82°. The sintering aids, i.e., Al_2_O_3_ and Y_2_O_3_ reacted and produced the Y_3_Al_5_O_12_ (YAG) during the pressureless sintering process. The corresponding peaks were also evident in the XRD patterns.

#### 3.1.2. The Phase Composition of the B_4_C/C(Graphite) Composites Produced by Pressureless Sintering

The characteristic XRD pattern of the B_4_C/C(graphite) composite’s bulks produced by pressureless sintering have been shown in Figure 2. Figure 2a–d respectively depict the XRD patterns of the (10 wt%, 20 wt%, 30 wt%, 40 wt%C(graphite)) B_4_C/C(graphite) composites sintered bulks. The diffraction peaks of the B_4_C phase and graphite phase are apparent in the XRD patterns. Whereas the diffractive peaks of the B_4_C phase were at 19.71°, 22.02°, 23.50°, 31.90°, 34.95°, and 37.82°, the peaks at 26.60°, 42.71°, 43.45°, 44.67°, 46.31°, 54.79°, 56.66°, 60.02°, and 77.69° arise from the graphite phase. The sintering aids, i.e., Al_2_O_3_ and Y_2_O_3_ upon reaction produced Y_3_Al_5_O_12_ (YAG) during the pressureless sintering process. The diffractive peaks of the low content of Y_3_Al_5_O_12_ (YAG) are also obvious in the XRD patterns.

### 3.2. The Microstructural Features of the B_4_C/C(Graphite) Composites Fabricated by the Pressureless Sintering Process

#### 3.2.1. The Microstructural Features of the B_4_C Monolith Fabricated by the Pressureless Sintering Process

Figure 3 depicts the SEM photographs of the fracture surface of the B_4_C monolith bulks fabricated by pressureless sintering. The pressureless sintered B_4_C monolith manifested a porous microstructure, with porosity as high as 20–30%. The B_4_C particles were disseminated within the composite’s matrix and the average particle dimension of boron carbide was approximately 3–5 µm. The porous microstructure of the B_4_C monolith bulks was presumably because of their fabrication from the pressureless sintering process. The pressureless sintered B_4_C monolith also exhibited an extremely low relative density; precisely about 70–80%.

#### 3.2.2. The Microstructural Features of the B_4_C/C(Graphite) Composites Fabricated by the Pressureless Sintering Process

Figure 4 presents the SEM images depicting the fracture surface of the pressureless sintered B_4_C/C(graphite) composite’s bulks. Figure 4a–d depict the SEM micrographs of the (10 wt%, 20 wt%, 30 wt%, 40 wt%C(graphite)) B_4_C/C(graphite) composites, respectively. It is obvious from the micrographs that the pressureless sintered B_4_C/C(graphite) composites have a porous microstructure, ranging between 25 and 38%. The B_4_C and graphite particles are distributed within the composite matrix and the graphite particles generated a laminated structure. The average particle dimensions of boron carbide are about 3–5 µm, whereas the mean particle dimensions of graphite were about 10–15 µm. It has also been observed that with the increase in the amount of graphite, the microstructure of pressureless sintered B_4_C/C(graphite) composites became increasingly porous. The relative density of the pressureless sintered B_4_C/C(graphite) composites was found to be extremely low at approximately 62–75%, and their porosity was extremely high.

### 3.3. The Phase Composition of the B_4_C-SiC-Si Composites Fabricated by the Silicon Infiltration Process

#### 3.3.1. The Phase Composition of the B_4_C-SiC-Si Composites Fabricated by Silicon Infiltration of Pressureless Sintered B_4_C Monolith

Figure 5 presents the XRD patterns corresponding to the silicon infiltrated B_4_C monolith (B_4_C-SiC-Si composites). The pressureless sintered B_4_C monolith was transformed into the B_4_C-SiC-Si composites through the silicon infiltration process. Figure 5 shows that following the silicon infiltration process, the diffractive peaks of the B_4_C phase disappeared partially. Besides, some new diffraction peaks were apparent in the XRD pattern, which presumably arise due to the existence of silicon carbide and silicon phase. As seen before, the diffraction peaks of the B_4_C phase were observed at 19.71°, 22.02°, 23.50°, 31.90°, 34.95°, and 37.82°. The diffraction peaks of the SiC phase and Si phase were observed, respectively, at 20.78°, 26.67°, 35.60°, 41.38°, 59.97°, 71.77°, 75.49°, and 28.44°, 47.30°, 56.12°, 69.13°, 76.37°. The XRD results have also demonstrated the existence of a boron carbide phase, silicon carbide phase, and residual silicon phase on the surface of the specimen obtained after silicon infiltration. These XRD analysis results also confirmed our assumption that the liquid silicon upon reaction with the boron carbide produced silicon carbide, leaving the residual silicon on the surface of silicon infiltrated specimens. The residual boron carbide also existed in the matrix of the composite. The boron carbide, silicon carbide, and residual silicon were sintered together yielding composites of the B_4_C-SiC-Si type. This way the pressureless sintered B_4_C monolith was transformed into the B_4_C-SiC-Si composites following the silicon infiltration process. The porosity of the pressureless sintered B_4_C monolith was about 20–30%. The apparent decrease in porosity can be ascribed to the silicon carbide and residual silicon (generated following silicon infiltration), filling in the pores of the matrix of the composite.

#### 3.3.2. The Phase Composition of the B_4_C-SiC-Si Composites Fabricated by Silicon Infiltration of Pressureless Sintered B_4_C/C(Graphite) Composites

Figure 6 represents the X-ray diffraction patterns of the silicon infiltrated B_4_C/C(graphite) composites (B_4_C-SiC-Si composites). Figure 6a–d, respectively, depict the XRD patterns of the (10 wt%, 20 wt%, 30 wt%, 40 wt%C(graphite)) silicon infiltrated B_4_C/C(graphite) composites. The pressureless sintered B_4_C/C(graphite) composites transformed into the B_4_C-SiC-Si composites following silicon infiltration. The silicon infiltrated B_4_C/C(graphite) composites have a precise chemical composition of the type B_4_C-SiC-Si. Figure 6a–d demonstrated that the diffractive peaks of the B_4_C phase and graphite phase vanished following the infiltration of silicon. While the diffraction peaks of the B_4_C phase disappeared only partially, those of graphite vanished completely. The appearance of some new diffraction peaks in the data corresponds to the existence of the silicon carbide phase and silicon phase. The diffraction peaks of the B_4_C phase appear at 19.71°, 22.02°, 23.50°, 31.90°, 34.95°, and 37.82°, the diffraction peaks of the SiC phase can be observed at 20.78°, 26.67°, 35.60°, 41.38°, 59.97°, 71.77°, 75.49°, whereas those of the Si phase are apparent at 28.44°, 47.30°, 56.12°, 69.13°, 76.37°. The XRD data point towards the existence of the boron carbide phase, silicon carbide phase, and residual silicon phase upon the surface of the specimens obtained after silicon infiltration. The XRD analysis suggests the infiltration of liquid silicon into the inner composite matrix and subsequent reaction with boron carbide and graphite producing silicon carbide. Some residual silicon stayed on the specimen surface following silicon infiltration. The boron carbide, silicon carbide, and residual silicon were sintered together forming the B_4_C-SiC-Si composites. The porosity of the pressureless sintered B_4_C/C(graphite) composites was about 25–38%. The apparent decrease in the porosity is because produced silicon carbide and residual silicon filled the pores in the matrix of the composite during the silicon infiltration process.

### 3.4. The Microstructural Features of the B_4_C-SiC-Si Composites Fabricated by the Silicon Infiltration Process

#### 3.4.1. The Microstructural Features of the B_4_C-SiC-Si Composites Fabricated by Silicon Infiltration of Pressureless Sintered B_4_C Monolith

Figure 7 presents the SEM photographs representing the fracture surface of the silicon infiltrated B_4_C monolith (B_4_C-SiC-Si composites). The pressureless sintered B_4_C monolith was transformed into the B_4_C-SiC-Si composites through the silicon infiltration process. These SEM images depict that the silicon infiltrated products B_4_C-SiC-Si composites possessed a compact and homogenous microstructure. The B_4_C particles and SiC particles were homogenously dispersed within the composite matrix. The mean particle size of B_4_C particles and SiC particles was about 4–5 µm. The existence of some residual silicon in the matrix of the composite is also evident. Because the pressureless sintered B_4_C monolith was a porous material; the porosity being 20–30%, the liquid silicon could infiltrate into the interior of the composite matrix during the silicon infiltration process. Boron carbide reacted with liquid silicon producing silicon carbide particles, and the residual boron carbide, as well as silicon, existed in the composite’s matrix. B_4_C-SiC-Si composites exhibited an exceptionally high relative density (98–99%), whereas their porosity was only about 1–2%. The silicon carbide was produced following silicon infiltration and the residual silicon filled the pores of the composites, so the porosity of the silicon infiltrated specimens decreased remarkably, and the B_4_C-SiC-Si composites became more compact and denser.

#### 3.4.2. The Microstructural Features of the B_4_C-SiC-Si Composites Fabricated by Silicon Infiltration of Pressureless Sintered B_4_C/C(Graphite) Composites

Figure 8 presents the SEM images depicting the fracture surface of the silicon infiltrated B_4_C/C(graphite) composites (B_4_C-SiC-Si composites). Figure 8a–d, respectively, depict the SEM micrographs of the (10 wt%, 20 wt%, 30 wt%, 40 wt%C(graphite)) silicon infiltrated B_4_C/C(graphite) composites. These SEM data present a compact and homogenous microstructure of the silicon infiltrated B_4_C-SiC-Si composites. The B_4_C particles and SiC particles were distributed homogeneously in the composite’s matrix, and some residual silicon also existed in the composite’s matrix. The mean particle size of B_4_C particles and SiC particles was about 4–5 µm. Because the pressureless sintered B_4_C/C(graphite) composites were porous (the porosity of the pressureless sintered B_4_C/C(graphite) composites was about 25–38%), the liquid silicon could infiltrate into the inner composite’s matrix during silicon infiltration. The reaction between boron carbide/graphite and liquid silicon produced silicon carbide particles, and the residual boron carbide and silicon existed in the composite’s matrix. The relative density of the B_4_C-SiC-Si composites was extremely high (98–99%), whereas the porosity of the B_4_C-SiC-Si composites was about 1–2%. The produced silicon carbide and residual silicon filled the pores, so the porosity of the silicon infiltrated specimens decreased remarkably, and the B_4_C-SiC-Si composites became more compact and denser.

### 3.5. The EDS Spectrum of the B_4_C-SiC-Si Composites Fabricated by the Silicon Infiltration Process

Figure 9 presents the SEM micrographs and Energy Dispersive Spectroscopy (EDS) spectrum of the B_4_C-SiC-Si composites fabricated by the silicon infiltration process. As already discussed, the B_4_C-SiC-Si composites manifested compact and homogenous microstructural features as shown in Figure 9a. The boron carbide and silicon carbide particles were dispersed homogeneously within the composite’s matrix, and the residual silicon also existed within the matrix. Figure 9b is a presentation of the EDS spectrum of the B_4_C-SiC-Si composites fabricated by the silicon infiltration process. The EDS spectroscopic analysis depicted that the silicon infiltrated composites were composed primarily of boron, silicon, and carbon, thereby demonstrating that the composites obtained following silicon infiltration comprised boron carbide and silicon carbide along with some content of silicon. It can therefore be assumed that the liquid silicon infiltrated into the interior of the composite and produce silicon carbide upon reaction with boron carbide, according to the following reaction equations [20,21,22,23,24,25,26,27,28,29,30,31,32,33,34,35,36,37,38,39,40,41,42,43,44,45,46,47,48,49,50,51,52,53,54,55]:
5 Si + 3 B_4_C = 3 SiC + 2 SiB_6_
2 Si + B_4_C = SiC + SiB_4_

The liquid silicon underwent a chemical reaction with graphite producing silicon carbide as shown by the following chemical reactions [20,21,22,23,24,25,26,27,28,29,30,31,32,33,34,35,36,37,38,39,40,41,42,43,44,45,46,47,48,49,50,51,52,53,54,55]:Si + C = SiC

The residual boron carbide and silicon sintered together with the product silicon carbide to give rise to B_4_C-SiC-Si composites. The residual silicon and silicon carbide produced during the process filled the pores in the composite’s matrix, hence the porosity of the specimens obtained following silicon infiltration decreased remarkably. Consequently, the B_4_C-SiC-Si composites exhibited an extremely high relative density and low porosity. The relative density of the B_4_C-SiC-Si composites was about 98–99%, and the porosity of the B_4_C-SiC-Si composites was about 1–2%.

### 3.6. The Mechanical Property of the Pressureless Sintered B_4_C/C(Graphite) Composites and the B_4_C-SiC-Si Composites

#### 3.6.1. The Density and Relative Density of the Pressureless Sintered B_4_C/C(Graphite) Composites and the B_4_C-SiC-Si Composites

Figure 10 represents the impact of the graphite content on the density of the pressureless sintered B_4_C/C(graphite) composites and the B_4_C-SiC-Si composites. The density of the pressureless sintered B_4_C/C(graphite) composites demonstrated a gradual decrease with the increase of graphite content. The density of the pressureless sintered B_4_C monolith was relatively high, but the density of the pressureless sintered B_4_C/C(graphite) composites was rather low. The underlying reason for this observation is that B_4_C materials have a density of 2.52 g/cm^3^, whereas graphite has a density of about 2.26 g/cm^3^, hence with the increase in the amount of graphite, the density of the pressureless sintered B_4_C/C(graphite) composites demonstrated a gradual decline. The porous microstructure of the pressureless sintered specimens also contributed to the extremely low density. The density of the pressureless sintered B_4_C/C(graphite) composites was about 1.5–1.9 g/cm^3^.

On the contrary, the density of the B_4_C-SiC-Si composites was phenomenally improved as compared to the pressureless sintered B_4_C/C(graphite) composites as depicted in Figure 10. The density of the B_4_C-SiC-Si composites progressively increased with the increase in the amount of added graphite in the B_4_C/C(graphite) composites. This observation was mainly because the liquid silicon infiltrated into the interior of the pressureless sintered B_4_C/C(graphite) composites and reacted with B_4_C and graphite to produce silicon carbide. Thus, with the increment in the amount of graphite in the B_4_C/C(graphite) composites, the graphite completely reacted with liquid silicon to produce silicon carbide, thus markedly increasing the content of produced silicon carbide in the B_4_C-SiC-Si composites. Since the density of silicon carbide is 3.21 g/cm^3^ and that of silicon is 2.32 g/cm^3^, the density of the produced B_4_C-SiC-Si composites increased progressively with the increase in the amount of silicon carbide produced. On the whole, the density of the B_4_C-SiC-Si composites was higher than the density of the corresponding B_4_C/C(graphite) composites. B_4_C-SiC-Si composites demonstrated a density value of 2.5855–2.9288 g/cm^3^ which is substantially higher than the pressureless sintered B_4_C/C(graphite) composites.

Figure 11 shows the impact of the amount of graphite on the relative density of the pressureless sintered B_4_C/C(graphite) composites and the B_4_C-SiC-Si composites. Figure 11 illustrates that the relative density of the pressureless sintered B_4_C/C(graphite) composites exhibited a gradual decline with the increase of graphite content. The relative density of the pressureless sintered B_4_C monolith was relatively high; however, the relative density of the pressureless sintered B_4_C/C(graphite) composites was rather low. This is presumably because the B_4_C/C(graphite) composites were produced via pressureless sintering and therefore exhibited a porous microstructure, causing their relative density to become extremely low, precisely 62–75%. The porosity of the pressureless sintered B_4_C/C(graphite) composites was about 25–38%.

Figure 11 also depicts that the relative density of the B_4_C-SiC-Si composites demonstrated a remarkable improvement in comparison with the pressureless sintered B_4_C/C(graphite) composites. In the B_4_C/C(graphite) composites, a gradual increase in the relative density was observed with the increase of graphite content. With the infiltration of liquid silicon into the interior of pressureless sintered B_4_C/C(graphite) composites, the B_4_C and graphite reacted with liquid silicon to produce the silicon carbide. The produced silicon carbide as well as the residual silicon filled the pores in the composite’s matrix, thereby resulting in a remarkable decrease in the porosity of the silicon infiltrated specimens and a consequent increase in their relative density. The residual silicon, boron carbide, and silicon carbide were sintered into dense and compact bulks. The density of silicon carbide is 3.21 g/cm^3^, hence the relative density of the composites based on B_4_C-SiC-Si became higher than the relative density of the B_4_C/C(graphite) composites obtained through pressureless sintering. The silicon infiltrated products B_4_C-SiC-Si composites presented compact and homogenous microstructural features. The relative density of the B_4_C-SiC-Si composites was about 98–99%. The density and relative density of the B_4_C-SiC-Si composites showed a significant improvement in comparison to the pressureless sintered B_4_C/C(graphite) composites.

#### 3.6.2. The Fracture Strength and Fracture Toughness of the Pressureless Sintered B_4_C/C(Graphite) Composites and the B_4_C-SiC-Si Composites

The impact of the amount of graphite on the fracture strength of the pressureless sintered B_4_C/C(graphite) composites and the B_4_C-SiC-Si composites has been shown in Figure 12. A gradual decrease in the fracture strength of the pressureless sintered B_4_C/C(graphite) composites was observed with the increase in the amount of graphite. While the fracture strength of the pressureless sintered B_4_C monolith was quite high, the pressureless sintered B_4_C/C(graphite) composites demonstrated a rather low fracture strength value. This is because the B_4_C/C(graphite) composites were fabricated by a pressureless sintering process, which imparted a porous microstructure, a low relative density (62–75%), and porosity of around 25–38% leaving them with an extremely low fracture strength. The fracture strength of the pressureless sintered B_4_C/C(graphite) composites lay within the 102–155 MPa range and reduced remarkably with the increase of graphite content.

Figure 12 depicts that the B_4_C-SiC-Si composites demonstrated a remarkable improvement in their fracture strength as compared to the pressureless sintered B_4_C/C(graphite) composites. The fracture strength of the B_4_C-SiC-Si composites increased progressively with the rise in graphite content. It occurs because liquid silicon infiltrated into the interior of pressureless sintered B_4_C/C(graphite) composites, and B_4_C and graphite reacted with liquid silicon producing silicon carbide. The resultant silicon carbide and the residual silicon and boron carbide were sintered together fabricating B_4_C-SiC-Si composites. The silicon carbide and residual silicon settled within the pores in the composite’s matrix, thereby causing a remarkable decrease in the porosity of the silicon infiltrated specimens and a corresponding increase in the relative density of the silicon infiltrated specimens. The relative density of the B_4_C-SiC-Si composites was found to be higher than 98%, the B_4_C-SiC-Si composites thus became denser and more compact and their fracture strength was higher than that of the pressureless sintered B_4_C/C(graphite) composites. B_4_C-SiC-Si composites have a fracture strength equivalent to 385–438 MPa. In addition, the produced silicon carbide could also improve the fracture strength of the B_4_C-SiC-Si composites, with the fracture strength increasing gradually with the increase of silicon carbide content. As an obvious fact, silicon carbide content underwent a gradual increase with the increase of graphite content in the B_4_C/C(graphite) composites. All these factors contribute to a remarkable improvement in the fracture strength of the B_4_C-SiC-Si composites as compared to the pressureless sintered B_4_C/C(graphite) composites.

The fracture toughness of the pressureless sintered B_4_C/C(graphite) composites and the B_4_C-SiC-Si composite is also affected by the amount of graphite as shown in Figure 13. As obvious from the figure, there was a gradual decline in the fracture toughness of the pressureless sintered B_4_C/C(graphite) composites with the increment in the amount of graphite. The fracture toughness of the pressureless sintered B_4_C/C(graphite) composites was rather low as compared to that of the pressureless sintered B_4_C monolith, presumably because the B_4_C/C(graphite) composites were produced via pressureless sintering and exhibited a porous microstructure. Additionally, the B_4_C/C(graphite) composites obtained via pressureless sintering had a rather low relative density and the porosity was 25–38%. It was also observed that the pressureless sintered B_4_C/C(graphite) composites demonstrated a remarkable decrease in their fracture toughness with the increase of graphite content. This is an outcome of the inherently low fracture toughness of graphitic materials. The observed fracture toughness of the pressureless sintered B_4_C/C(graphite) composites was about 1.568–2.35 MPa·m^1/2^.

It is also evident from Figure 13 that the fracture toughness of the B_4_C-SiC-Si composites underwent a marked improvement when compared with the pressureless sintered B_4_C/C(graphite) composites, demonstrating a gradual increase with the increase in the amount of graphite. This can be attributed to the infiltration of liquid silicon into the interior pressureless sintered B_4_C/C(graphite) composites and the subsequent reaction with B_4_C and graphite producing silicon carbide. Silicon carbide and residual silicon settled into the pores of the composite’s matrix, thereby decreasing its porosity and increasing the relative density up to >98%, rendering the B_4_C-SiC-Si composites even more dense and compact. Consequently, the fracture toughness of the B_4_C-SiC-Si composites was recorded as 6.05–7.55 MPa·m^1/2^ which is much higher in comparison to the fracture toughness of the pressureless sintered B_4_C/C(graphite) composites.

The silicon carbide produced following silicon infiltration is also assumed to improve the fracture toughness of the B_4_C-SiC-Si composites, which underwent a progressive increase progressively with the increase of the silicon carbide content. The content of produced silicon carbide in turn gradually increased with the increase of the amount of graphite in the B_4_C/C(graphite) composites. Additionally, the relative density of the B_4_C-SiC-Si composites was extremely high, about 98–99%. Thus, the fracture toughness and fracture strength of the B_4_C-SiC-Si composites were significantly enhanced as compared to the pressureless sintered B_4_C/C(graphite) composites. While the pressureless sintered B_4_C/C(graphite) composites were soft and brittle and could be easily broken, the B_4_C-SiC-Si composites were compact and hard and exhibited extremely high fracture strength and high fracture toughness.

#### 3.6.3. The Vickers Hardness and Elastic Modulus of the Pressureless Sintered B_4_C/C(Graphite) Composites and the B_4_C-SiC-Si Composites

The impact of the graphite content on the Vickers hardness of the pressureless sintered B_4_C/C(graphite) composites and the B_4_C-SiC-Si composites has been shown in Figure 14. The Vickers hardness of the pressureless sintered B_4_C/C(graphite) composites demonstrated a progressive decline with the increase of graphite content. While the Vickers hardness of the pressureless sintered B_4_C monolith was relatively high, its value was rather low for the pressureless sintered B_4_C/C(graphite) composites. Owing to their fabrication from the pressureless sintering process, the B_4_C/C(graphite) composites exhibited porous microstructures and have a low relative density (62–75%) and a porosity of about 25–38%. It has been known that the Vickers hardness of graphite is rather low. the laminated structured graphite materials have a softening effect on the ceramic composites. Consequently, the Vickers hardness of the pressureless sintered B_4_C/C(graphite) composites exhibited a gradual decrease with the increase in the amount of graphite. The pressureless sintered B_4_C/C(graphite) composites were soft and very brittle and could be easily broken. The Vickers hardness of the pressureless sintered B_4_C/C(graphite) composites was about 2.52–4.88 GPa.

Figure 14 also shows that the Vickers hardness of the B_4_C-SiC-Si composites was markedly enhanced when compared to the pressureless sintered B_4_C/C(graphite) composites and increased progressively with the increase in the amount of graphite. This can be ascribed to the infiltration of liquid silicon into the interior pressureless sintered B_4_C/C(graphite) composites, and the subsequent reaction with B_4_C and graphite, producing silicon carbide. Silicon carbide and residual silicon settle into the pores of the composite’s matrix, thereby decreasing its porosity and increasing the relative density up to >98%, rendering the B_4_C-SiC-Si composites even more dense and compact and subsequently increasing their Vickers’s hardness.

Besides, the Vickers hardness of boron carbide and silicon carbide is exceptionally high, 20 GPa to be specific. The specimen surface of the B_4_C-SiC-Si composites primarily comprises boron carbide, silicon carbide, and residual silicon. Consequently, the Vickers hardness of the B_4_C-SiC-Si composites following silicon infiltration is observed to be higher than pressureless sintered B_4_C/C(graphite) composites. B_4_C-SiC-Si composites demonstrated a Vickers hardness value as high as 22.885–26.768 GPa.

Moreover, the Vickers hardness of the B_4_C-SiC-Si composites demonstrated a gradual increase with the increase in the amount of silicon carbide. The content of silicon carbide increased steadily with the increment in graphite content in the B_4_C/C(graphite) composites, thereby enhancing the Vickers hardness of the B_4_C-SiC-Si composites to a remarkable extent. A high value of the Vickers hardness, in turn, guarantees an enhancement in the resistance of the B_4_C-SiC-Si composites to wear. Thus, the B_4_C-SiC-Si composites exhibit an outstanding wear resistance.

Figure 15 is a representation of the variation in elastic modulus of the pressureless sintered B_4_C/C(graphite) composites and the B_4_C-SiC-Si composites as a function of graphite content. The elastic modulus of the pressureless sintered B_4_C/C(graphite) composites reduced gradually with the increase in the amount of graphite. While the elastic modulus of the pressureless sintered B_4_C monolith was relatively high, that of the pressureless sintered B_4_C/C(graphite) composites was rather low. This is a consequence of the porous microstructure, low relative density (62–75%), and high porosity (25–38%) of the pressureless sintered B_4_C/C(graphite) composites.

The elastic modulus of boron carbide is high, and the elastic modulus of graphite was rather low. This leads to the introduction of softness and brittleness into the pressureless sintered B_4_C/C(graphite) composites. The pressureless sintered B_4_C/C(graphite) composites are highly porous and exhibit a low elastic modulus, with the elastic modulus gradually declining with the increment in graphite content. The exact value of elastic modulus of the pressureless sintered B_4_C/C(graphite) composites was about 106–145 GPa as illustrated in Figure 15.

When compared with the pressureless sintered B_4_C/C(graphite) composites, the B_4_C-SiC-Si composites demonstrate an improved value of the elastic modulus. There was a gradual increase in the elastic modulus of the B_4_C-SiC-Si composites with the increase in the amount of graphite in the B_4_C/C(graphite) composites as shown in Figure 15. This, again, is a consequence of liquid silicon seeping into the interior of pressureless sintered B_4_C/C(graphite) composites. Thus, the B_4_C and graphite undergo a chemical reaction with liquid silicon, generating silicon carbide. The silicon carbide thus generated takes up the porous spaces in the composite’s matrix along with the residual silicon, decreasing its porosity and increasing the relative density. The B_4_C-SiC-Si composites became denser and more compact with a relative density as high as >98%. Due to their compact and homogenous microstructure, B_4_C-SiC-Si composites have an elastic modulus in the range of 385–410 GPa, which is considerably higher than the elastic modulus of the pressureless sintered B_4_C/C(graphite) composites.

Both boron carbide and silicon carbide have an exceptionally high value of elastic modulus. Thus, the silicon carbide generated following silicon infiltration could also contribute to improving the B_4_C-SiC-Si composites in terms of their elastic modulus. A gradual increase in the elastic modulus of the B_4_C-SiC-Si composites was evident with the increase of the content of silicon carbide, which increases linearly with the increase in graphite content. B_4_C-SiC-Si composites thus demonstrated an extremely high elastic modulus which was substantially enhanced as opposed to the pressureless sintered B_4_C/C(graphite) composites.

### 3.7. The Wear Resistance of the Pressureless Sintered B_4_C/C(Graphite) Composites and the B_4_C-SiC-Si Composites

The pressureless sintered B_4_C/C(graphite) composites and the B_4_C-SiC-Si composites were examined for their wear resistance with the help of wear experiments. The vertical coordinate was wear weight loss, whereas the wear time represents the horizontal coordinate. Figure 16a presents the influence of wear time on the weight loss of the pressureless sintered B_4_C/C(graphite) composites due to wear. The wear weight loss of the pressureless sintered B_4_C/C(graphite) composites manifested a progressive increase with the increment in wear time. Additionally, the extent of wear weight loss increased with the gradual rise in graphite content. The pressureless sintered B_4_C monolith gave a rather low wear weight loss curve owing to the relatively high inherent Vickers hardness of B_4_C ceramics. The composites with the graphite content of 10 wt%, 20 wt%, 30 wt%, and 40 wt% exhibited higher wear weight loss curves which may be attributed to the low value of Vickers hardness of the pressureless sintered B_4_C/C(graphite) composites. The weight loss curves in Figure 16a, illustrate that the highest weight loss existed for sintered B_4_C/C(40 wt% graphite) composites while the lowest was observed for the B_4_C monolith. The findings suggest that for the pressureless sintered B_4_C/C(graphite) composites, the wear resistance was very poor, and the wear weight loss was high. The wear resistance of the pressureless sintered B_4_C/C(graphite) composites reduced gradually with the increase in the amount of graphite presumably on account of the low Vickers hardness of the pressureless sintered B_4_C/C(graphite) composites.

The influence of wear time on the wear weight loss of the B_4_C-SiC-Si composites has been shown in Figure 16b. Contrary to the B_4_C/C(graphite) composites, the B_4_C-SiC-Si composites demonstrated an increase in wear weight loss with the increase in wear time. The wear weight loss of the B_4_C-SiC-Si composites manifested a gradual decrease with the increment in graphite content in the corresponding B_4_C/C(graphite) composites. The B_4_C-SiC-Si composites with the graphite content of 10 wt%, 20 wt%, 30 wt%, and 40 wt% in the B_4_C/C(graphite) composites gave lower wear weight loss curves presumably on account of the high Vickers hardness of the B_4_C-SiC-Si composites. As shown in Figure 16b, the highest weight loss existed for silicon infiltrated B_4_C monolith, whereas the lowest was observed for the silicon infiltrated B_4_C/C(40 wt%graphite) composites. When the wear weight loss was low, the wear resistance was very high. Thus, the wear resistance of the B_4_C-SiC-Si composites was quite high on account of their high Vickers hardness which is equivalent to 22.885–26.768 GPa.

From Figure 16a,b, it could also be deduced that the wear weight loss of the B_4_C-SiC-Si composites was relatively lower in comparison to the wear weight loss of the pressureless sintered B_4_C/C(graphite) composites. This implies that the wear resistance of the B_4_C-SiC-Si composites demonstrated a substantial improvement as compared to pressureless sintered B_4_C/C(graphite) composites which may be attributed to their high Vickers hardness.

The wear weight loss curve was created using line fitness in Figure 16, and the slope of line fitness is the ratio of wear weight loss to wear time. The slope thus represents the wear loss rates. The following equation can be used to illustrate the relationship between wear weight decrease and wear time: R = M/t, where R represents wear loss rates, M represents wear weight loss and t represents wear duration. As a result, the slope of the wear weight loss curve was computed using the formulae R = M/t.

The wear resistance of pressureless sintered specimens and silicon infiltrated specimens were measured by friction wear experiments. A ratio of wear weight loss to wear time is referred to as the wear loss rate. Figure 17 presents the influence of graphite content on the wear loss rates of the pressureless sintered B_4_C/C(graphite) composites and the B_4_C-SiC-Si composites. It can be seen that the wear loss rates of the pressureless sintered B_4_C/C(graphite) composites gradually increased with the increase in the amount of graphite. This indicated that the wear resistance of the pressureless sintered B_4_C/C(graphite) composites underwent a gradual decrease with the increase in the amount of graphite. A possible explanation for this observation is that the B_4_C/C(graphite) composites were fabricated via a pressureless sintering method and have a low relative density and a porosity ranging between 25–38%. The Vickers hardness of the pressureless sintered B_4_C/C(graphite) composites reduced progressively with the increase in the amount of graphite. The surface hardness of the pressureless sintered B_4_C/C(graphite) composites was also exceptionally low. All these factors contribute to the reduced wear resistance of the pressureless sintered B_4_C/C(graphite) composites, which manifested a noticeable decline with the increase in the amount of graphite.

As regards the B_4_C-SiC-Si composites, their wear loss rates underwent a gradual decrease with the rise in graphite content as depicted in Figure 17. The outcome revealed that the B_4_C-SiC-Si composites specimen’s weight demonstrated a steady decrease with the increase in the amount of graphite in the B_4_C/C(graphite) composites. Figure 17 also revealed that the wear resistance of the B_4_C-SiC-Si composites was substantially enhanced when compared to the pressureless sintered B_4_C/C(graphite) composites, indicating that the silicon infiltrated specimen’s weight underwent a gradual decrement with the increment in the amount of graphite in the B_4_C/C(graphite) composites. The loss in weight observed for the silicon infiltrated specimens was exceptionally minimal. The B_4_C-SiC-Si composites manifested remarkable resistance to wear. The underlying phenomenon, in this case, is the infiltration of liquid silicon into the interior of the composites and subsequent reaction with B_4_C and graphite yielding silicon carbide. Silicon carbide and residual silicon ultimately occupy the pores in the composite’s matrix, decreasing its porosity and increasing its relative density. B_4_C-SiC-Si composites are thus rendered denser and more compact and their surface hardness and Vickers hardness were exceptionally high; hence, the wear resistance of the B_4_C-SiC-Si composites was also extremely high. Silicon carbide and boron carbide along with the residual silicon existing on the silicon infiltrated specimen’s surface also contribute towards the remarkable improvement in the resistance of the B_4_C-SiC-Si composites towards wear.

### 3.8. Formation Mechanism of the B_4_C-SiC-Si Composites via Silicon Infiltration Process

During the silicon infiltration, the liquid silicon infiltrated into the interior of the composite’s specimens, and subsequently reacted chemically with boron carbide producing silicon carbide according to the following chemical reactions [20,21,22,23,24,25,26,27,28,29,30,31,32,33,34,35,36,37,38,39,40,41,42,43,44,45,46,47,48,49,50,51,52,53,54,55]:5 Si + 3 B_4_C = 3 SiC + 2 SiB_6_;
2 Si + B_4_C = SiC + SiB_4_.

The liquid silicon combined with graphite to produce the silicon carbide according to the following chemical reactions [20,21,22,23,24,25,26,27,28,29,30,31,32,33,34,35,36,37,38,39,40,41,42,43,44,45,46,47,48,49,50,51,52,53,54,55]:Si + C = SiC.

The B_4_C-SiC-Si composites were primarily composed of boron carbide, resultant silicon carbide, and only a little content of silicon; the net content of boron carbide and silicon carbide was >90%, whereas the residual silicon was found to be <10%.

## 4. Conclusions

The B_4_C/C(graphite) composites were produced via a pressureless sintering process. At an elevated temperature, liquid silicon infiltration was conducted for improving the mechanical property of the pressureless sintered B_4_C/C(graphite) composites. Since the porosity of pressureless sintered B_4_C/C(graphite) composites was about 25–38%, the liquid silicon could infiltrate into the interior of B_4_C/C(graphite) composites, reacting with B_4_C and graphite to produce silicon carbide. The boron carbide, silicon carbide, and residual silicon are then sintered together giving rise to B_4_C-SiC-Si composites. This way the pressureless sintered B_4_C/C(graphite) composites transformed into the B_4_C-SiC-Si composites following silicon infiltration.

This work is based on an investigation of the microstructure, phase composition, and mechanical properties of the B_4_C/C(graphite) composites obtained via a pressureless sintering process. The XRD data demonstrated that the pressureless sintered bulks were primarily made up of the B_4_C phase and graphite phase. Besides, they have a porous microstructure, with the porosity lying in the range of 25–38%. The relative density of the pressureless sintered B_4_C/C(graphite) composites was about 62–75% and their mechanical property underwent a gradual decline with the increase in the amount of graphite. The density, relative density, fracture toughness, fracture strength, Vickers hardness, elastic modulus, and wear resistance of the pressureless sintered B_4_C/C(graphite) composites also manifested a progressive decrease with the increase in the amount of graphite.

Upon silicon infiltration, the pressureless sintered B_4_C/C(graphite) composites reacted with liquid silicon and transformed into B_4_C-SiC-Si composites. The microstructure, phase composition, and mechanical characteristics of the B_4_C-SiC-Si composites fabricated via the silicon infiltration process were then thoroughly investigated. XRD data suggest the existence of a B_4_C phase, a SiC phase, and a residual Si phase on the silicon infiltrated specimen’s surface. The silicon infiltrated products, i.e., B_4_C-SiC-Si composites exhibited a homogenous and compact microstructure, an extremely high relative density, and remarkably low porosity. The microstructure investigation results showed that the silicon infiltrated specimens were primarily made up of boron carbide, silicon carbide, and residual silicon. There was a substantial improvement in the B_4_C-SiC-Si composites in terms of their mechanical property when compared with the pressureless sintered B_4_C/C(graphite) composites. The density and relative density, fracture strength and fracture toughness, elastic modulus, Vickers hardness, and wear resistance of the B_4_C-SiC-Si composites were also significantly improved and exhibited a gradual increase with the increase in the amount of graphite in the composites. It can thus be concluded that the silicon infiltration process efficiently improves the mechanical property and wear resistance of the pressureless sintered B_4_C/C(graphite) composites, and the B_4_C-SiC-Si composites thus generated can find potentially useful applications due to their superior physical attributes.

## Figures and Tables

**Figure 1 materials-15-04853-f001:**
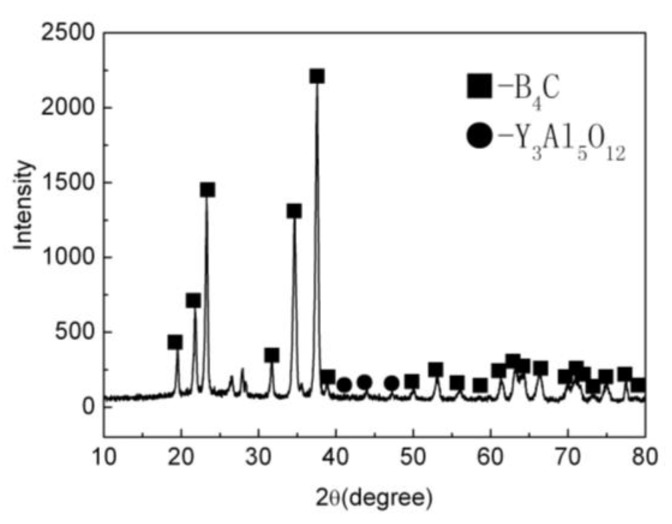
The XRD patterns of the B_4_C monolith bulks produced by pressureless sintering.

**Figure 2 materials-15-04853-f002:**
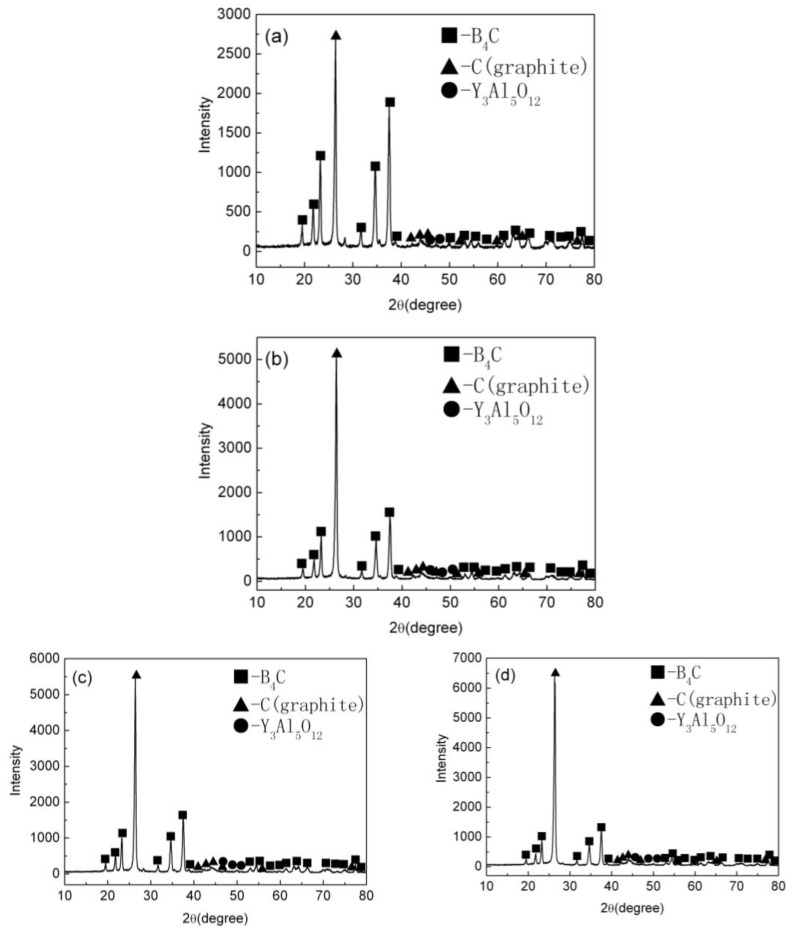
The XRD patterns of the B_4_C/C(graphite) composite’s bulks produced by pressureless sintering; (**a**–**d**) depict the XRD patterns of the (10 wt%, 20 wt%, 30 wt%, 40 wt% C(graphite)) B_4_C/C(graphite) composites sintered bulks, respectively.

**Figure 3 materials-15-04853-f003:**
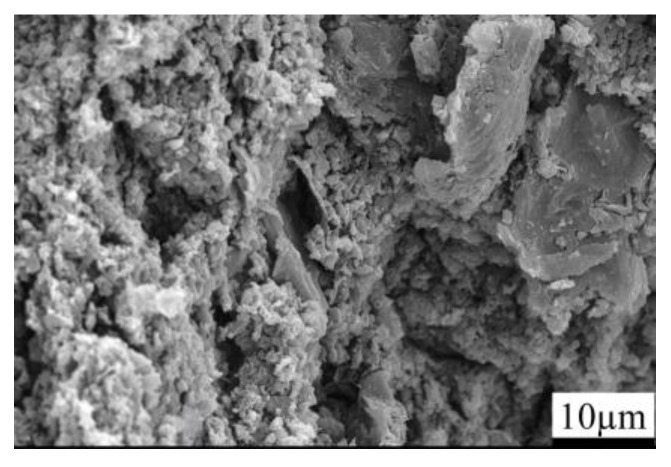
SEM images representing the fracture surface of the B_4_C monolith bulk fabricated by a pressureless sintering process.

**Figure 4 materials-15-04853-f004:**
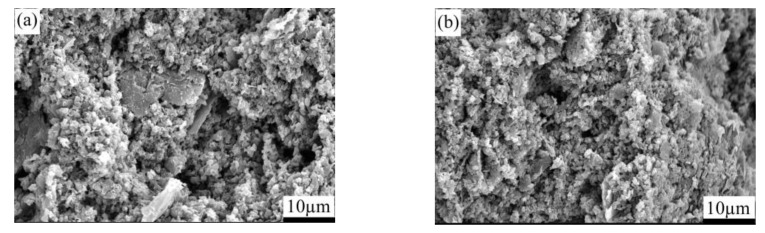
SEM images representing the fracture surface of the B_4_C/C(graphite) composite’s bulks fabricated by pressureless sintering; (**a**–**d**) depict the SEM micrographs of the (10 wt%, 20 wt%, 30 wt%, 40 wt% C(graphite)) B_4_C/C(graphite) composites sintered bulks, respectively.

**Figure 5 materials-15-04853-f005:**
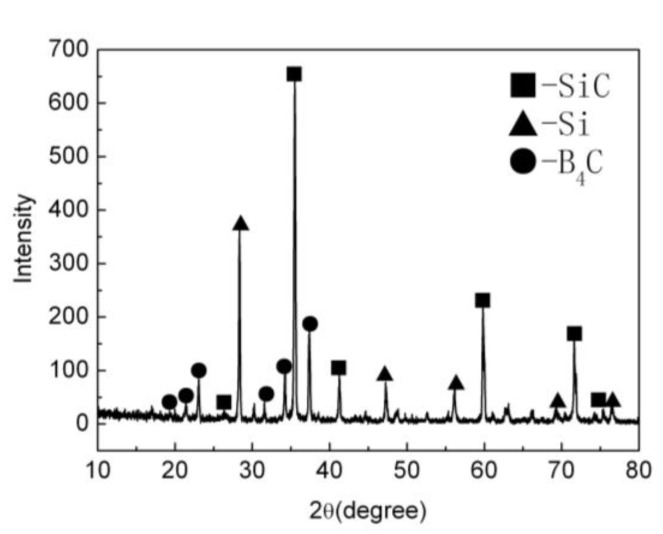
The XRD patterns of the silicon infiltrated B_4_C monolith (B_4_C-SiC-Si composites).

**Figure 6 materials-15-04853-f006:**
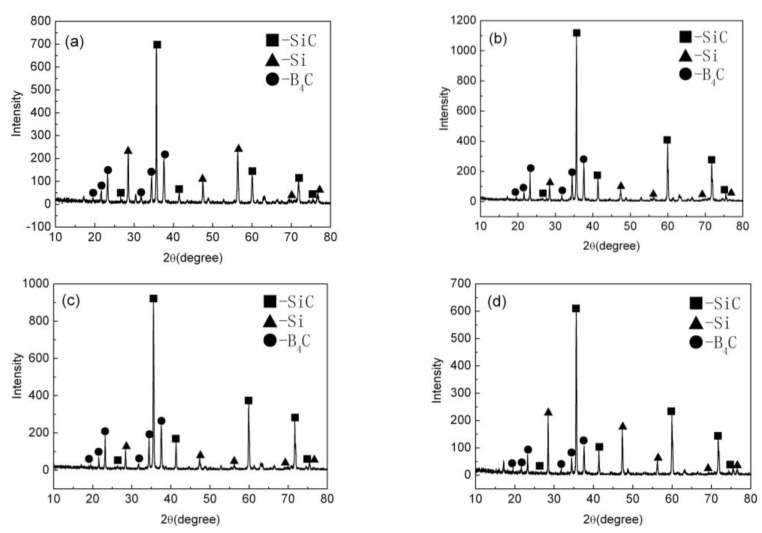
The XRD patterns of the silicon infiltrated B_4_C/C(graphite) composites (B_4_C-SiC-Si composites); (**a**–**d**) depict the XRD patterns of the (10 wt%, 20 wt%, 30 wt%, 40 wt%C(graphite)) silicon infiltrated B_4_C/C(graphite) composites, respectively.

**Figure 7 materials-15-04853-f007:**
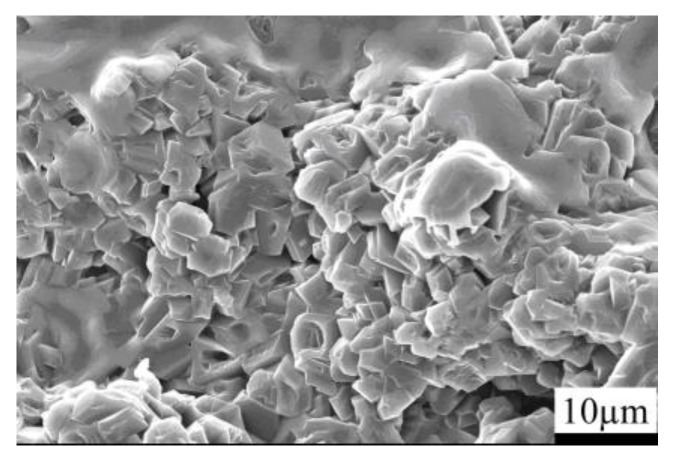
The SEM images representing the fracture surface of the silicon infiltrated B_4_C monolith (B_4_C-SiC-Si composites).

**Figure 8 materials-15-04853-f008:**
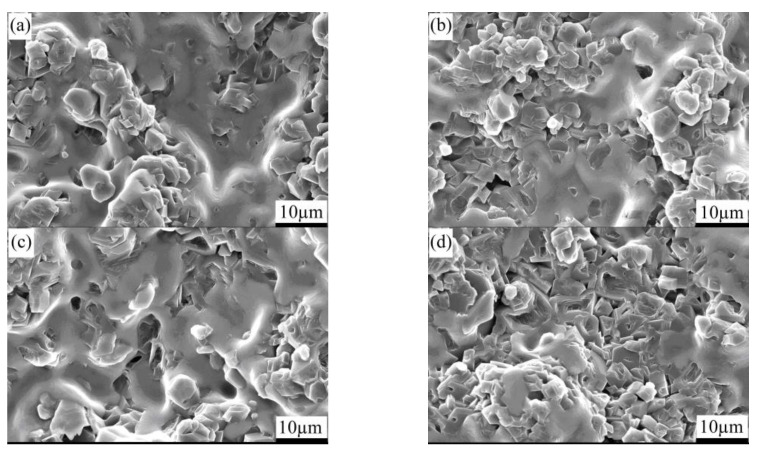
SEM images depicting the fracture surface of the silicon infiltrated B_4_C/C(graphite) composites (B_4_C-SiC-Si composites); (**a**–**d**) depict the SEM micrographs of the (10 wt%, 20 wt%, 30 wt%, 40 wt%C(graphite)) silicon infiltrated B_4_C/C(graphite) composites, respectively.

**Figure 9 materials-15-04853-f009:**
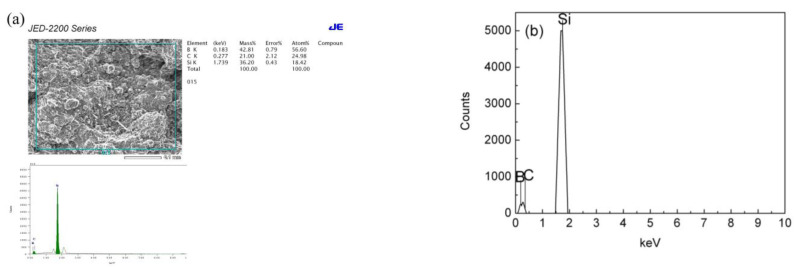
The (**a**) SEM micrographs and (**b**) EDS spectrum of the B_4_C-SiC-Si composites fabricated by the silicon infiltration process.

**Figure 10 materials-15-04853-f010:**
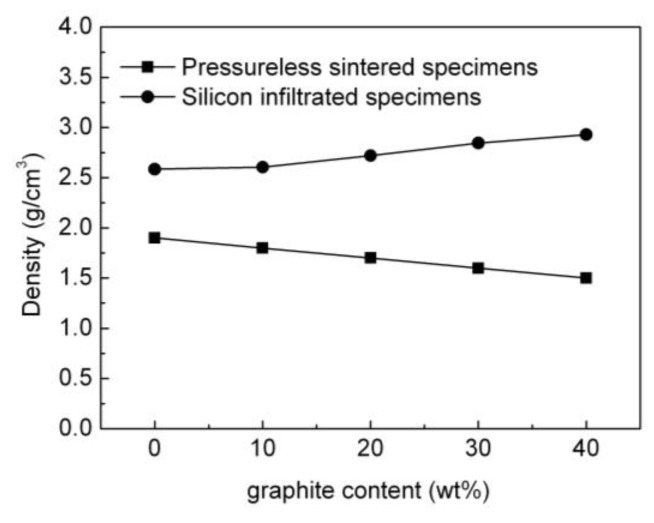
The influence of graphite content on the density of the pressureless sintered B_4_C/C(graphite) composites and the B_4_C-SiC-Si composites.

**Figure 11 materials-15-04853-f011:**
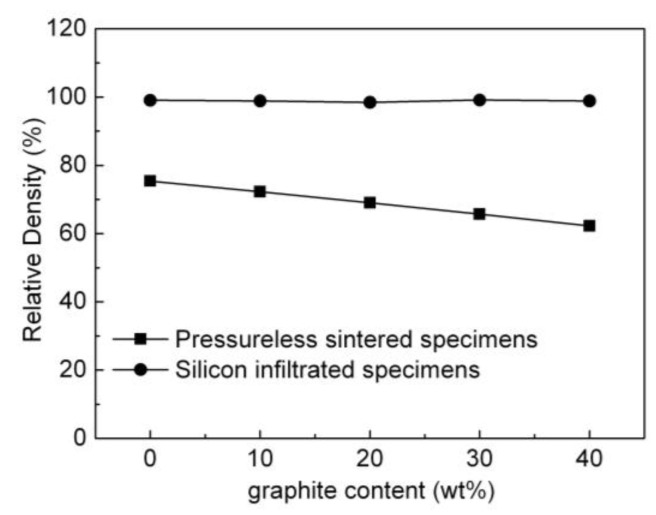
The impact of the graphite content on the relative density of the pressureless sintered B_4_C/C(graphite) composites and the B_4_C-SiC-Si composites.

**Figure 12 materials-15-04853-f012:**
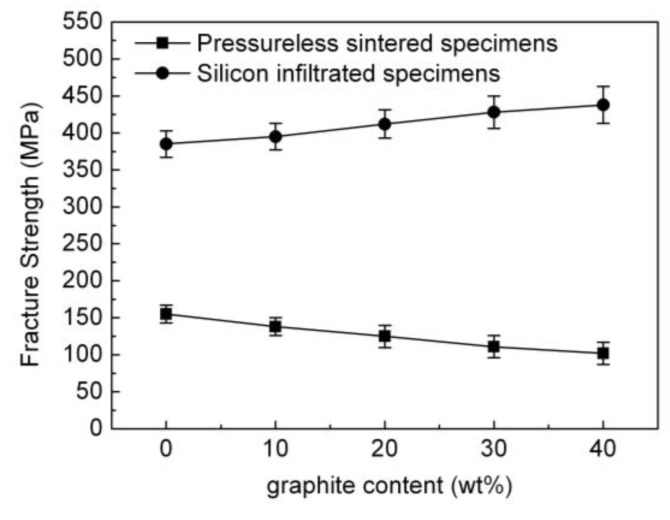
The impact of graphite content on the fracture strength of the pressureless sintered B_4_C/C(graphite) composites and the B_4_C-SiC-Si composites.

**Figure 13 materials-15-04853-f013:**
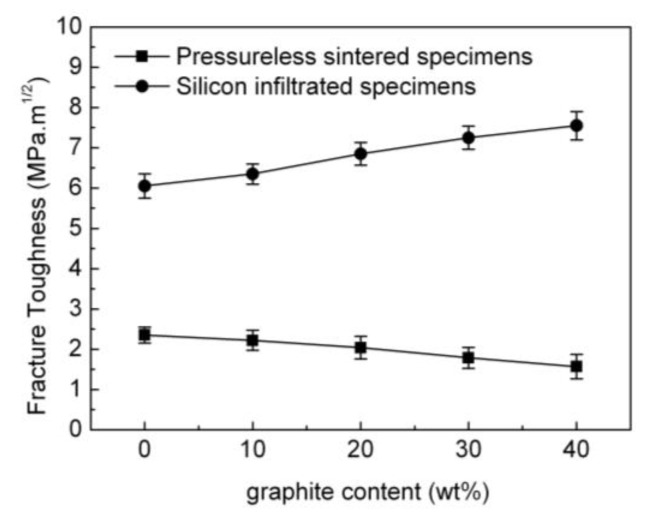
The effects of graphite content on the fracture toughness of the pressureless sintered B_4_C/C(graphite) composites and the B_4_C-SiC-Si composites.

**Figure 14 materials-15-04853-f014:**
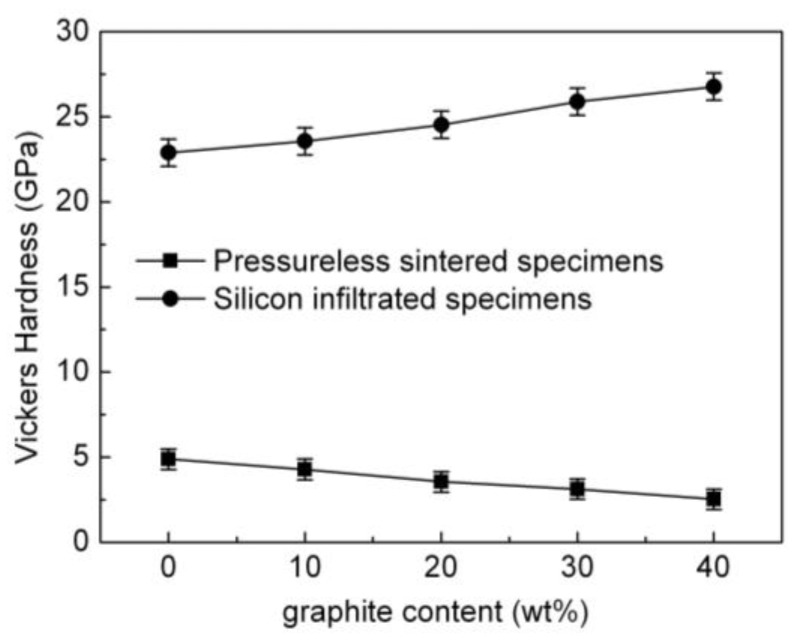
The effects of graphite content on the Vickers hardness of the pressureless sintered B_4_C/C(graphite) composites and the B_4_C-SiC-Si composites.

**Figure 15 materials-15-04853-f015:**
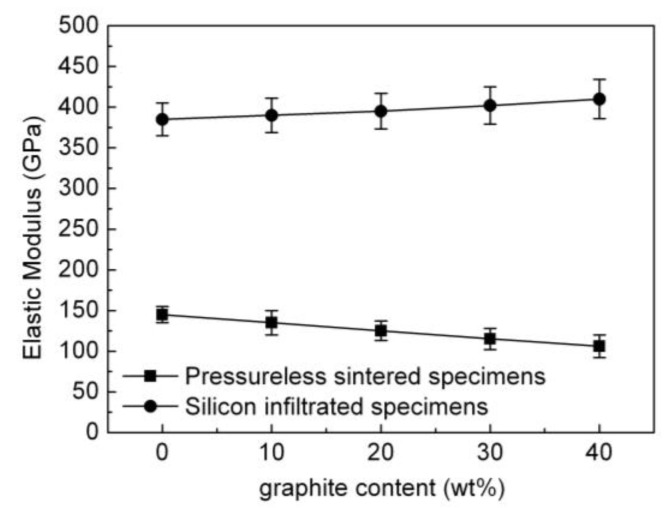
The influence of graphite content on the elastic modulus of the pressureless sintered B_4_C/C(graphite) composites and the B_4_C-SiC-Si composites.

**Figure 16 materials-15-04853-f016:**
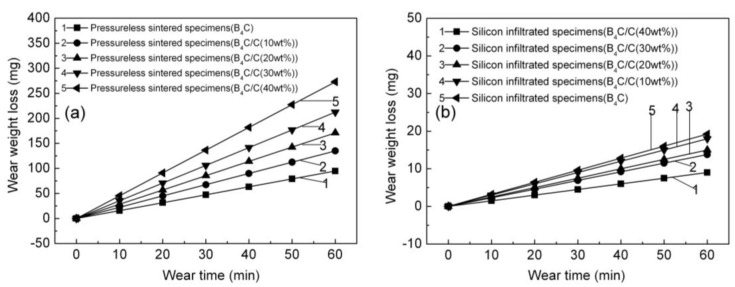
(**a**) The influence of wear time on the wear weight loss of the pressureless sintered B_4_C/C(graphite) composites; (**b**) the influence of wear time on the wear weight loss of the B_4_C-SiC-Si composites.

**Figure 17 materials-15-04853-f017:**
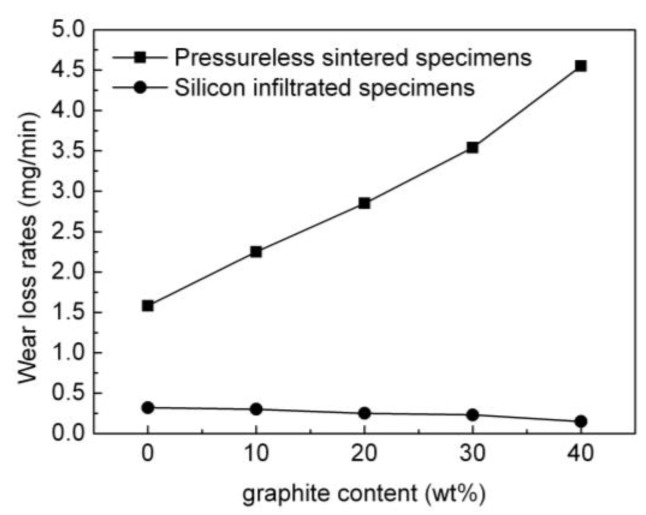
The influence of graphite content on the wear loss rates of the pressureless sintered B_4_C/C(graphite) composites and the B_4_C-SiC-Si composites.

## Data Availability

Not applicable.

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
