# Peer review of "Investigation of Microstructural Features and Mechanical Characteristics of the Pressureless Sintered B4C/C(Graphite) Composites and the B4C-SiC-Si Composites Fabricated by the Silicon Infiltration Process"

_materials, 2022, doi:10.3390/ma15144853_

Round 1
Reviewer 1 Report
In this submission to Materials, the authors give an experimental study on B4C/C(graphite) composites which they fabricated by employing a pressureless sintering process. The authors find that the pressureless sintered B4C/C(graphite) composites exhibited extremely low mechanical characteristics. The authors used a liquid silicon infiltration technique for enhancing the mechanical property of B4C/C(graphite) composites. The authors give additional details of the microstructure, phase composition, and mechanical characteristics of the pressureless sintered B4C/C(graphite) composites and B4C-SiC-Si composites. The authors carried out XRD techniques that demonstrated that the pressureless sintered bulks were composed of the B4C phase and graphite phase. The authors note that the density, relative density, fracture strength, fracture toughness, elastic modulus, and Vickers hardness of the B4C-SiC-Si composites were notably enhanced as compared to the pressureless sintered B4C/C(graphite) composites.
I find this topic to be of interest to structural materials researchers as well as readers of Materials. As such, I am relatively supportive of publication with a few minor comments that should be addressed in the next revision. In particular, there has been much work using first-principles approaches to understand the strength of these and other materials, which should be noted.
Phys. Rev. B 2011, 84, 014112
J. Chem. Theory Comput. 2015, 11, 5426–5435
In particular, these prior studies showed that density functional theory calculations can be used to understand atomistic details of structural mechanics and strength. I am not asking the authors to carry out DFT calculations, but it should be mentioned that these approaches can reveal mechanisms for the strength of structures in a more unbiased way, which should be noted in the next revision. With this revision, I would be willing to re-review this manuscript for possible publication.
Author Response
Response to reviewer’s comments
Reviewer’s comments: I find this topic to be of interest to structural materials researchers as well as readers of Materials. As such, I am relatively supportive of publication with a few minor comments that should be addressed in the next revision. In particular, there has been much work using first-principles approaches to understand the strength of these and other materials, which should be noted.
Phys. Rev. B 2011, 84, 014112
- Chem. Theory Comput. 2015, 11, 5426–5435
In particular, these prior studies showed that density functional theory calculations can be used to understand atomistic details of structural mechanics and strength. I am not asking the authors to carry out DFT calculations, but it should be mentioned that these approaches can reveal mechanisms for the strength of structures in a more unbiased way, which should be noted in the next revision. With this revision, I would be willing to re-review this manuscript for possible publication.
Author reply: Dear reviewer, how are you, I think my research direction is completely different from the reviewer's research direction. I think the reviewer's research direction should be first-principles calculation or density functional theory, and my research direction is the experiment, I haven't done first-principles calculations and simulations or density functional theory at all, I don't know what the first principles calculation are, and I don't even know what the density functional theory is. I think that the reviewers would ask me to adding the theory of first-principles calculations and simulations or density functional theory into my submission. My research direction is mainly experimental methods, and I use experiments to obtain data and pictures to write papers. Therefore, the reviewers asked me to perform first-principles simulation calculations or density functional theory calculations. I really can't perform first-principles simulation calculations or density functional theory calculations, and I won't perform first-principles simulation calculations or density functional theory calculations either. I have no way to perform theoretical calculations in accordance with the reviewer's opinion. I can only explain the relationship between the microstructure and mechanical properties of composites materials by means of experiments. I hope the reviewers can understand that I have never performed first-principles simulation calculations or density functional theory calculations. I myself have limited ability to perform first-principles simulation calculations and density functional theory calculations. These first-principles simulation calculations and density functional theory calculations have some effect on my paper, so I want to add something about theoretical calculations of first-principles simulation calculations and density functional theory calculations into the introduction of my submission, I can only explain the relationship between the microstructure and mechanical properties of composites materials through experimental methods. First-principles simulation calculations and density functional theory are too hard for me, I have never been exposed to this knowledge and content of first-principles simulation calculations and density functional theory calculations, and I can only complete the suggestions given to me by the reviewers through adding the conception, theory, methods and application of the first-principles calculations and simulations or density functional theory into the introduction of my submission. So I decided to add the first-principles simulation calculations and density functional theory calculations into the introduction in my submission. I expected the reviewer can understand that I have never study the first-principles simulation calculations and density functional theory, my research is always experiments, not first-principles simulation calculations and density functional theory calculations. I have some understanding about the first-principles simulation calculations and density functional theory calculations. So I can add the first-principles simulation calculations and density functional theory calculations into the introduction in my submission. So I decided to add the first-principles simulation calculations and density functional theory calculations into the introduction in my submission. I think that the first-principles simulation calculations and density functional theory calculations were suitable for use in my submission. So I decided to add the first-principles simulation calculations and density functional theory calculations into the introduction in my submission. So I decided to add the conception, theory, methods and application of the first-principles simulation calculations and density functional theory calculations into the introduction in my submission.
The reviewer provide the two references, according to the requirements of reviewers, I found out these two references showed as following:
[1]Roman Raucoules, Nathalie Vast, Emmanuel Betranhandy, and Jelena Sjakste. Mechanical properties of icosahedral boron carbide explained from first principles[J]. PHYSICAL REVIEW B, 2011, 84, 014112.
Abstract: An exhaustive study of the neutral structural defects of icosahedral B4C has been performed with the density functional theory. Vacancies have been determined to be boron vacancies in the C-B-C chains. Their presence is shown to yield a discontinuous variation of crystal volume upon increasing pressure, when the formation of a C-C bond occurs in the chains. The dynamical failure of shocked B4C is attributed to the formation of these C-C bonds.
[2]Niranjan V. Ilawe, Jonathan A. Zimmerman, and Bryan M. Wong. Breaking Badly: DFT-D2 Gives Sizeable Errors for Tensile Strengths in Palladium-Hydride Solids[J]. Journal of Chemical Theory and Computation, 2015, 11, 5426–5435.
Abstract: Dispersion interactions play a crucial role in noncovalently bound molecular systems, and recent studies have shown that dispersion effects are also critical for accurately describing covalently bound solids. While most studies on bulk solids have solely focused on equilibrium properties (lattice constants, bulk moduli, and cohesive energies), there has been little work on assessing the importance of dispersion effects for solid-state properties far from equilibrium. In this work, we present a detailed analysis of both equilibrium and highly nonequilibrium properties (tensile strengths leading to fracture) of various palladiumhydride systems using representative DFT methods within the LDA, GGA, DFT-D2, DFT-D3, and nonlocal vdw-DFT families. Among the various DFT methods, we surprisingly find that the empirically constructed DFT-D2 functional gives extremely anomalous and qualitatively incorrect results for tensile strengths in palladium-hydride bulk solids. We present a detailed analysis of these effects and discuss the ramifications of using these methods for predicting solid-state properties far from equilibrium. Most importantly, we suggest caution in using DFT-D2 (or other coarse-grained parametrizations obtained from DFT-D2) for computing material properties under large stress/strain loads or for evaluating solid-state properties under extreme structural conditions.
Dear reviewer, the first paper discussed the work using first-principles approaches to understand the strength of these and other materials, the second paper discussed the work about density functional theory calculations can be used to understand atomistic details of structural mechanics and strength. But I haven't done first-principles calculations and simulations or density functional theory at all, I don't know what the first principles calculation are, and I don't even know what density functional theory is. I myself have limited ability to perform first-principles simulation calculations and density functional theory calculations. These first-principles simulation calculations and density functional theory calculations have some effect on my paper, so I want to add something about theoretical calculations of first-principles simulation calculations and density functional theory calculations into the introduction in my submission. First-principles simulation calculations and density functional theory are too hard for me, I have never been exposed to this knowledge and content of first-principles simulation calculations and density functional theory calculations, and I can only complete the suggestions given to me by the reviewers through adding the conception, theory, methods and application of the first-principles calculations and simulations or density functional theory into the introduction in my submission. I have some understanding about the first-principles simulation calculations and density functional theory calculations. So I decided to add the first-principles simulation calculations and density functional theory calculations into the introduction in my submission. So I decided to add the conception, theory, methods and application of the first-principles simulation calculations and density functional theory calculations into the introduction in my submission.
I can only mention the first-principles simulation calculations and density functional theory calculations in the introduction in my revised submission, such as “In particular, there has been much work using first-principles approaches to understand the strength of these and other materials. In particular, these prior studies showed that density functional theory calculations can be used to understand atomistic details of structural mechanics and strength. These approaches can reveal mechanisms for the strength of structures in a more unbiased way. The first principle calculation played a very important role in studying the relationship between microstructure analysis and mechanical property of the composites materials. At the same time, the density functional calculation played a very important role in studying the relationship between microstructure analysis and mechanical property of the composites materials. The first principle calculation and the density functional calculation could be completely understand the relationship between the microstructure characterization and mechanical property of the materials by using the calculation methods. So it was very important to using the first principle calculation and the density functional calculation to understand the relationship between the microstructure characterization and mechanical property of the composites materials.
According to the principle of interaction between atomic nucleus and electron and its basic motion law, using the principle of quantum mechanics, starting from the specific requirements, the algorithm to directly solve the Schrödinger equation after some approximation processing is customarily called the first principle.
Density functional theory (DFT) is a method for studying the electronic structure of multi-electron systems. Density functional theory has a wide range of applications in physics and chemistry, especially for studying the properties of molecules and condensed matter. It is one of the most commonly used methods in the fields of computational materials science and computational chemistry in condensed matter physics.
I added the above discussion about the first-principles simulation calculations and density functional theory calculations in the introduction in the revised submission. So I decided to add the conception, theory, methods and application of the first-principles simulation calculations and density functional theory calculations into the introduction in my submission. I cited the above two published paper as the references in the revised submission.
I can only explain the relationship between the microstructure and mechanical properties of composites materials by means of experiments. Now I explain the relationship between the microstructure and mechanical properties of composites materials by means of experiments.
1 The effects of graphite content on the B4C weight fraction and SiC weight fraction of the B4C-SiC composites
This response to reviewer’s comments describes the mixing ratio of B4C to graphite in the pressureless sintered B4C/C(graphite) composites. The mass fractions of graphite of the pressureless sintered B4C/C(graphite) composites are 10wt%, 20wt%, 30wt%, 40wt%, respectively, and the mass fractions of boron carbide of the pressureless sintered B4C/C(graphite) composites are 90wt%, 80wt%, 70wt%, 60wt%, respectively. That is to say, the mass ratio of B4C to graphite is 90:10, 80:20, 70:30, and 60:40, respectively. According to the mass ratio of B4C to graphite are 90:10, according to the percentage, B4C accounts for 90, and graphite accounts for 10 respectively. Assuming that B4C does not react with silicon, only graphite reacts with liquid silicon, then according to the reaction formula: set the mass fractions of SiC generated to be x, and calculate the value of x according to the following formula. The content of silicon did not calculate.
C+Si=SiC
, then x=33.33, so the amount of SiC generated is 33.33.
Then the weight fractions of B4C =;
Then the weight fractions of SiC =.
According to the mass ratio of B4C to graphite being 90:10 respectively, it can be seen that the weight fractions of B4C is 72.97%, and the weight fractions of SiC is 27.03%. So in the B4C-SiC composites, the weight fractions of B4C is 72.97%, and the weight fractions of SiC is 27.03%.
According to the mass ratio of B4C to graphite is 80:20, according to the percentage, B4C accounts for 80, and graphite accounts for 20 respectively. Assuming that B4C does not react with silicon, only graphite reacts with liquid silicon, then according to the reaction formula: set the mass fractions of SiC generated to be x, and calculate the value of x according to the following formula. The content of silicon did not calculate.
C+Si=SiC
, then x=66.67, so the amount of SiC generated is 66.67.
Then the weight fractions of B4C =;
Then the weight fractions of SiC =.
According to the mass ratio of B4C to graphite being 80:20 respectively, it can be seen that the weight fractions of B4C is 54.54%, and the weight fractions of SiC is 45.46%. So in the B4C-SiC composites, the weight fractions of B4C is 54.54%, and the weight fractions of SiC is 45.46%.
According to the mass ratio of B4C to graphite is 70:30, according to the percentage, B4C accounts for 70, and graphite accounts for 30 respectively. Assuming that B4C does not react with silicon, only graphite reacts with liquid silicon, then according to the reaction formula: set the mass fractions of SiC generated to be x, and calculate the value of x according to the following formula. The content of silicon did not calculate.
C+Si=SiC
, then x=100, so the amount of SiC generated is 100.
Then the weight fractions of B4C =;
Then the weight fractions of SiC =.
According to the mass ratio of B4C to graphite being 70:30 respectively, it can be seen that the weight fractions of B4C is 41.18%, and the weight fractions of SiC is 58.82%. So in the B4C-SiC composites, the weight fractions of B4C is 41.18%, and the weight fractions of SiC is 58.82%.
According to the mass ratio of B4C to graphite is 60:40, according to the percentage, B4C accounts for 60, and graphite accounts for 40 respectively. Assuming that B4C powder does not react with silicon, only graphite reacts with liquid silicon, then according to the reaction formula: set the mass fractions of SiC generated to be x, and calculate the value of x according to the following formula. The content of silicon did not calculate.
C+Si=SiC
, then x=133.33, so the amount of SiC generated is 133.33.
Then the weight fractions of B4C =;
Then the weight fractions of SiC =.
According to the mass ratio of B4C to graphite being 60:40 respectively, it can be seen that the weight fractions of B4C is 31.04%, and the weight fractions of SiC is 68.96%. So in the B4C-SiC composites, the weight fractions of B4C is 31.04%, and the weight fractions of SiC is 68.96%.
Therefore, the weight fractions of B4C in the B4C-SiC composites is 31.04%-72.97%, and the weight fractions of SiC in the B4C-SiC composites is 27.03%-68.96%. It can be seen that the weight fractions of B4C is relatively high, while the weight fractions of SiC is also relatively high.
Table.1shows the effects of graphite content on the B4C weight fraction and SiC weight fraction of the B4C-SiC composites. Table.1 showed that, with the increase of graphite content of the pressurless sintered B4C/C(graphite) composites, the weight fraction of B4C in the B4C-SiC composites decreased gradually, while the weight fraction of SiC in the B4C-SiC composites increased gradually. The B4C-SiC composites exhibited homogenous and compact microstructure with the increase of SiC content. The mechanical property of the B4C-SiC composites increased gradually with the increase of SiC content. The density and relative density, fracture strength and fracture toughness, Vickers hardness and elastic modulus the B4C-SiC composites increased gradually with the increase of SiC content. The wear resistance of the B4C-SiC composites increased gradually with the increase of SiC content.
Table.1. The effects of graphite content on the B4C weight fraction and SiC weight fraction of the B4C-SiC composites.
|
Ratio of B4C to graphite |
B4C weight fractions (%) |
SiC weight fractions (%) |
|
Ratio of B4C to graphite=90:10 |
72.97 |
27.03 |
|
Ratio of B4C to graphite=80:20 |
54.54 |
45.46 |
|
Ratio of B4C to graphite=70:30 |
41.18 |
58.82 |
|
Ratio of B4C to graphite=60:40 |
31.04 |
68.96 |
Fig.1 shows the effects of graphite content on the B4C weight fraction and SiC weight fraction of the B4C-SiC composites. Fig.1 showed that, with the increase of graphite content of the pressurless sintered B4C/C(graphite) composites, the weight fraction of B4C in the B4C-SiC composites decreased gradually, while the weight fraction of SiC in the B4C-SiC composites increased gradually. The B4C-SiC composites exhibited homogenous and compact microstructure with the increase of SiC content. The mechanical property of the B4C-SiC composites increased gradually with the increase of SiC content. The density and relative density, fracture strength and fracture toughness, Vickers hardness and elastic modulus the B4C-SiC composites increased gradually with the increase of SiC content. The wear resistance of the B4C-SiC composites increased gradually with the increase of SiC content.
Fig.1. The effects of graphite content on the B4C weight fraction and SiC weight fraction of the B4C-SiC composites.
2 The SEM micrographs of the B4C-SiC-Si composites fabricated by silicon infiltration process
Now the SEM micrographs of the B4C-SiC-Si composites were shown in order to prove the B4C-SiC-Si composites exhibited homogenous and compact microstructure.
Fig. 2 presents the SEM photographs representing the fracture surface of the silicon infiltrated B4C monolith (B4C-SiC-Si composites). The pressureless sintered B4C monolith was transformed into the B4C-SiC-Si composites through the silicon infiltration process. These SEM images depict that the silicon infiltrated products B4C-SiC-Si composites possessed a compact and homogenous microstructure. B4C-SiC-Si composites exhibited an exceptionally high relative density (98-99%), whereas their porosity was only about 1-2%. The silicon carbide produced following silicon infiltration as well as the residual silicon filled the pores of the composites, so the porosity of the silicon infiltrated specimens decreased remarkably, and the B4C-SiC-Si composites became more compact and dense.
Fig. 2. The SEM images representing the fracture surface of the silicon infiltrated B4C monolith (B4C-SiC-Si composites).
Fig. 3 presents the SEM images depicting the fracture surface of the silicon infiltrated B4C/C(graphite) composites (B4C-SiC-Si composites). Fig. 3(a-d) respectively depict the SEM micrographs of the (10wt%, 20wt%, 30wt%, 40wt%C(graphite)) silicon infiltrated B4C/C(graphite) composites. These SEM data present a compact and homogenous microstructure of the silicon infiltrated B4C-SiC-Si composites. The relative density of the B4C-SiC-Si composites was extremely high (98-99%), whereas the porosity of the B4C-SiC-Si composites was about 1-2%. The produced silicon carbide and residual silicon filled the pores, so the porosity of the silicon infiltrated specimens decreased remarkably, and the B4C-SiC-Si composites became more compact and dense.
Fig. 3. SEM images depicting the fracture surface of the silicon infiltrated B4C/C(graphite) composites(B4C-SiC-Si composites); Fig. 3(a-d) respectively depict the SEM micrographs of the (10wt%, 20wt%, 30wt%, 40wt%C(graphite)) silicon infiltrated B4C/C(graphite) composites.
3 The density and relative density of the pressureless sintered B4C/C(graphite) composites and the B4C-SiC-Si composites
The density of the pressureless sintered B4C/C(graphite) composites demonstrated a gradual decrease with the increase of graphite content. The density of the pressureless sintered B4C/C(graphite) composites was about 1.5-1.9g/cm3. The density of the B4C-SiC-Si composites progressively increased with the increase in the amount of added graphite in the B4C/C(graphite) composites. Since the density of silicon carbide is 3.21g/cm3 and that silicon is 2.32g/cm3, the density of produced B4C-SiC-Si composites increased progressively with the increase in the amount of silicon carbide produced. B4C-SiC-Si composites demonstrated a density value of 2.5855-2.9288g/cm3 which is substantially higher than the pressureless sintered B4C/C(graphite) composites.
The relative density of the pressureless sintered B4C/C(graphite) composites exhibited a gradual decline with the increase of graphite content. This is presumably because the B4C/C(graphite) composites were produced via pressureless sintering and therefore exhibited a porous microstructure, thereby causing their relative density to become extremely low, precisely 62-75%. The porosity of the pressureless sintered B4C/C(graphite) composites was about 25-38%. The relative density of the B4C-SiC-Si composites demonstrated a remarkable improvement in comparison with the pressureless sintered B4C/C(graphite) composites. In the B4C/C(graphite) composites, a gradual increase in the relative density was observed with the increase of graphite content. The density of silicon carbide is 3.21g/cm3, hence the relative density of the composites based on B4C-SiC-Si became higher than the relative density of the B4C/C(graphite) composites obtained through pressureless sintering. The silicon infiltrated products B4C-SiC-Si composites presented compact and homogenous microstructural features. The relative density of the B4C-SiC-Si composites was about 98-99%. The density and relative density of the B4C-SiC-Si composites showed a significant improvement in comparison to the pressureless sintered B4C/C(graphite) composites.
4 The fracture strength and fracture toughness of the pressureless sintered B4C/C(graphite) composites and the B4C-SiC-Si composites
A gradual decrease in the fracture strength of the pressureless sintered B4C/C(graphite) composites was observed with the increase in the amount of graphite. The fracture strength of the pressureless sintered B4C/C(graphite) composites lay within the 102-155MPa range and reduced remarkably with the increase of graphite content. The B4C-SiC-Si composites demonstrated a remarkable improvement in their fracture strength as compared to the pressureless sintered B4C/C(graphite) composites. The fracture strength of the B4C-SiC-Si composites increased progressively with the rise in graphite content. The relative density of the B4C-SiC-Si composites was found to be higher than 98%, the B4C-SiC-Si composites thus became more dense and compact and their fracture strength was higher than that of the pressureless sintered B4C/C(graphite) composites. B4C-SiC-Si composites have a fracture strength equivalent to 385-438MPa. In addition, the produced silicon carbide could also improve the fracture strength of the B4C-SiC-Si composites, with the fracture strength increasing gradually with the increase of silicon carbide content. As an obvious fact, silicon carbide content underwent a gradual increase with the increase of graphite content in the B4C/C(graphite) composites. All these factors contribute to a remarkable improvement in the fracture strength of the B4C-SiC-Si composites as compared to the pressureless sintered B4C/C(graphite) composites.
There was a gradual decline in the fracture toughness of the pressureless sintered B4C/C(graphite) composites with the increment in the amount of graphite. The observed fracture toughness of the pressureless sintered B4C/C(graphite) composites was about 1.568MPa·m1/2-2.35MPa·m1/2. The fracture toughness of the B4C-SiC-Si composites underwent a marked improvement when compared with the pressureless sintered B4C/C(graphite) composites, demonstrating a gradual increase with the increase in the amount of graphite. Silicon carbide and residual silicon settled into the pores of the composite's matrix, thereby decreasing its porosity and increasing the relative density up to > 98%, rendering the B4C-SiC-Si composites even more dense and compact. Consequently, the fracture toughness of the B4C-SiC-Si composites was recorded as 6.05-7.55MPa·m1/2 which is much higher in comparison to the fracture toughness of the pressureless sintered B4C/C(graphite) composites. The silicon carbide produced following silicon infiltration is also assumed to improve the fracture toughness of the B4C-SiC-Si composites, which underwent a progressive increase progressively with the increase of the silicon carbide content. The content of produced silicon carbide in turn gradually increased with the increase of the amount of graphite in the B4C/C(graphite) composites. Also, the relative density of the B4C-SiC-Si composites was extremely high, about 98-99%. Thus the fracture toughness and fracture strength of the B4C-SiC-Si composites were significantly enhanced as compared to the pressureless sintered B4C/C(graphite) composites.
5 The Vickers hardness and elastic modulus of the pressureless sintered B4C/C(graphite) composites and the B4C-SiC-Si composites
The Vickers hardness of the pressureless sintered B4C/C(graphite) composites demonstrated a progressive decline with the increase of graphite content. The Vickers hardness of the pressureless sintered B4C/C(graphite) composites was about 2.52-4.88GPa. The Vickers hardness of the B4C-SiC-Si composites was markedly enhanced when compared to the pressureless sintered B4C/C(graphite) composites and increased progressively with the increase in the amount of graphite. The Vickers hardness of boron carbide and silicon carbide is exceptionally high; 20GPa to be specific. The specimen surface of the B4C-SiC-Si composites primarily comprises boron carbide, silicon carbide, and residual silicon. As a consequence, the Vickers hardness of the B4C-SiC-Si composites following silicon infiltration is observed to be higher than pressureless sintered B4C/C(graphite) composites. B4C-SiC-Si composites demonstrated a Vickers hardness value as high as 22.885-26.768GPa. The Vickers hardness of the B4C-SiC-Si composites demonstrated a gradual increase with the increase in the amount of silicon carbide. The content of silicon carbide increased steadily with the increment in graphite content in the B4C/C(graphite) composites, thereby enhancing the Vickers hardness of the B4C-SiC-Si composites to a remarkable extent.
The elastic modulus of the pressureless sintered B4C/C(graphite) composites reduced gradually with the increase in the amount of graphite. The exact value of elastic modulus of the pressureless sintered B4C/C(graphite) composites was about 106-145GPa. The B4C-SiC-Si composites demonstrate an improved value of the elastic modulus. There was a gradual increase in the elastic modulus of the B4C-SiC-Si composites with the increase in the amount of graphite in the B4C/C(graphite) composites. The B4C-SiC-Si composites became more dense and compact with a relative density as high as > 98%. Due to their compact and homogenous microstructure, B4C-SiC-Si composites have an elastic modulus in the range of 385-410GPa, which is considerably higher than the elastic modulus of the pressureless sintered B4C/C(graphite) composites. Both boron carbide and silicon carbide have an exceptionally high value of elastic modulus. Thus, the silicon carbide generated following silicon infiltration could also contribute to improving the B4C-SiC-Si composites in terms of their elastic modulus. A gradual increase in the elastic modulus of the B4C-SiC-Si composites was evident with the increase of the content of silicon carbide, which increases linearly with the increase in graphite content. B4C-SiC-Si composites thus demonstrated an extremely high elastic modulus which was substantially enhanced as opposed to the pressureless sintered B4C/C(graphite) composites.
6 The wear resistance of the pressureless sintered B4C/C(graphite) composites and the B4C-SiC-Si composites
The wear resistance of the pressureless sintered B4C/C(graphite) composites reduced gradually with the increase in the amount of graphite presumably on account of the low Vickers hardness of the pressureless sintered B4C/C(graphite) composites. The wear resistance of the B4C-SiC-Si composites demonstrated a substantial improvement as compared to pressureless sintered B4C/C(graphite) composites which may be attributed to their high Vickers hardness. B4C-SiC-Si composites are thus rendered denser and more compact, their surface hardness and Vickers hardness were exceptionally high, hence the wear resistance of the B4C-SiC-Si composites was also extremely high. Silicon carbide and boron carbide along with the residual silicon existing on the silicon infiltrated specimen’s surface also contribute towards the remarkable improvement in the resistance of the B4C-SiC-Si composites towards wear. Thus, the wear resistance of the B4C-SiC-Si composites was quite high on account of their high Vickers hardness which is equivalent to 22.885-26.768GPa.
7 The relationship of microstructure and the improved mechanical property and wear resistance of the B4C-SiC-Si composites
The B4C-SiC-Si composites exhibited a homogenous and compact microstructure, an extremely high relative density, and remarkably low porosity. The microstructure investigation results showed that the silicon infiltrated specimens were primarily made up of boron carbide, silicon carbide, and residual silicon. There was a substantial improvement in the B4C-SiC-Si composites in terms of their mechanical property when compared with the pressureless sintered B4C/C(graphite) composites. The density and relative density, fracture strength and fracture toughness, elastic modulus, Vickers hardness and wear resistance of the B4C-SiC-Si composites were also significantly improved and exhibited a gradual increase with the increase in the amount of graphite in the composites. It can thus be concluded that the silicon infiltration process efficiently improves the mechanical property and wear resistance of the pressureless sintered B4C/C(graphite) composites and B4C-SiC-Si composites thus generated can find potentially useful applications due to their superior physical attributes.
So the B4C-SiC-Si composites exhibited homogenous and compact microstructure, and the B4C-SiC-Si exhibited high mechanical property and excellent wear resistance. The density and relative density, fracture strength and fracture toughness, Vickers hardness and elastic modulus of the B4C-SiC-Si composites were significantly improved in comparison with the pressureless sintered B4C/C(graphite) composites. The wear resistance of the B4C-SiC-Si composites were significantly improved in comparison with the pressureless sintered B4C/C(graphite) composites.
8 I decided to add the first-principles simulation calculations and density functional theory calculations into the introduction in my submission. I cited the above two published paper as the references in the revised submission.
So I can explain the relationship between the microstructure and mechanical property of the B4C-SiC-Si composites by experimental methods. I can only explain the relationship between the microstructure and mechanical properties of composites materials through experimental methods. I expected the reviewer can understand that I have never study the first-principles simulation calculations and density functional theory, my research is always experiments, not first-principles simulation calculations and density functional theory calculations. I have some understanding about the first-principles simulation calculations and density functional theory calculations. So I can add the first-principles simulation calculations and density functional theory calculations into the introduction in my submission. So I decided to add the first-principles simulation calculations and density functional theory calculations into the introduction in my submission. I think that the first-principles simulation calculations and density functional theory calculations were suitable for use in my submission. So I decided to add the first-principles simulation calculations and density functional theory calculations into the introduction in my submission. I cited the above two published paper as the references in the revised submission.
I added the above discussion about the first-principles simulation calculations and density functional theory calculations in the introduction in the revised submission. So I decided to add the conception, theory, methods and application of the first-principles simulation calculations and density functional theory calculations into the introduction in my submission. I cited the above two published paper as the references in the revised submission.
Response to reviewer’s comments
Reviewer’s comments: I find this topic to be of interest to structural materials researchers as well as readers of Materials. As such, I am relatively supportive of publication with a few minor comments that should be addressed in the next revision. In particular, there has been much work using first-principles approaches to understand the strength of these and other materials, which should be noted.
Phys. Rev. B 2011, 84, 014112
- Chem. Theory Comput. 2015, 11, 5426–5435
In particular, these prior studies showed that density functional theory calculations can be used to understand atomistic details of structural mechanics and strength. I am not asking the authors to carry out DFT calculations, but it should be mentioned that these approaches can reveal mechanisms for the strength of structures in a more unbiased way, which should be noted in the next revision. With this revision, I would be willing to re-review this manuscript for possible publication.
Author reply: Dear reviewer, how are you, I think my research direction is completely different from the reviewer's research direction. I think the reviewer's research direction should be first-principles calculation or density functional theory, and my research direction is the experiment, I haven't done first-principles calculations and simulations or density functional theory at all, I don't know what the first principles calculation are, and I don't even know what the density functional theory is. I think that the reviewers would ask me to adding the theory of first-principles calculations and simulations or density functional theory into my submission. My research direction is mainly experimental methods, and I use experiments to obtain data and pictures to write papers. Therefore, the reviewers asked me to perform first-principles simulation calculations or density functional theory calculations. I really can't perform first-principles simulation calculations or density functional theory calculations, and I won't perform first-principles simulation calculations or density functional theory calculations either. I have no way to perform theoretical calculations in accordance with the reviewer's opinion. I can only explain the relationship between the microstructure and mechanical properties of composites materials by means of experiments. I hope the reviewers can understand that I have never performed first-principles simulation calculations or density functional theory calculations. I myself have limited ability to perform first-principles simulation calculations and density functional theory calculations. These first-principles simulation calculations and density functional theory calculations have some effect on my paper, so I want to add something about theoretical calculations of first-principles simulation calculations and density functional theory calculations into the introduction of my submission, I can only explain the relationship between the microstructure and mechanical properties of composites materials through experimental methods. First-principles simulation calculations and density functional theory are too hard for me, I have never been exposed to this knowledge and content of first-principles simulation calculations and density functional theory calculations, and I can only complete the suggestions given to me by the reviewers through adding the conception, theory, methods and application of the first-principles calculations and simulations or density functional theory into the introduction of my submission. So I decided to add the first-principles simulation calculations and density functional theory calculations into the introduction in my submission. I expected the reviewer can understand that I have never study the first-principles simulation calculations and density functional theory, my research is always experiments, not first-principles simulation calculations and density functional theory calculations. I have some understanding about the first-principles simulation calculations and density functional theory calculations. So I can add the first-principles simulation calculations and density functional theory calculations into the introduction in my submission. So I decided to add the first-principles simulation calculations and density functional theory calculations into the introduction in my submission. I think that the first-principles simulation calculations and density functional theory calculations were suitable for use in my submission. So I decided to add the first-principles simulation calculations and density functional theory calculations into the introduction in my submission. So I decided to add the conception, theory, methods and application of the first-principles simulation calculations and density functional theory calculations into the introduction in my submission.
The reviewer provide the two references, according to the requirements of reviewers, I found out these two references showed as following:
[1]Roman Raucoules, Nathalie Vast, Emmanuel Betranhandy, and Jelena Sjakste. Mechanical properties of icosahedral boron carbide explained from first principles[J]. PHYSICAL REVIEW B, 2011, 84, 014112.
Abstract: An exhaustive study of the neutral structural defects of icosahedral B4C has been performed with the density functional theory. Vacancies have been determined to be boron vacancies in the C-B-C chains. Their presence is shown to yield a discontinuous variation of crystal volume upon increasing pressure, when the formation of a C-C bond occurs in the chains. The dynamical failure of shocked B4C is attributed to the formation of these C-C bonds.
[2]Niranjan V. Ilawe, Jonathan A. Zimmerman, and Bryan M. Wong. Breaking Badly: DFT-D2 Gives Sizeable Errors for Tensile Strengths in Palladium-Hydride Solids[J]. Journal of Chemical Theory and Computation, 2015, 11, 5426–5435.
Abstract: Dispersion interactions play a crucial role in noncovalently bound molecular systems, and recent studies have shown that dispersion effects are also critical for accurately describing covalently bound solids. While most studies on bulk solids have solely focused on equilibrium properties (lattice constants, bulk moduli, and cohesive energies), there has been little work on assessing the importance of dispersion effects for solid-state properties far from equilibrium. In this work, we present a detailed analysis of both equilibrium and highly nonequilibrium properties (tensile strengths leading to fracture) of various palladiumhydride systems using representative DFT methods within the LDA, GGA, DFT-D2, DFT-D3, and nonlocal vdw-DFT families. Among the various DFT methods, we surprisingly find that the empirically constructed DFT-D2 functional gives extremely anomalous and qualitatively incorrect results for tensile strengths in palladium-hydride bulk solids. We present a detailed analysis of these effects and discuss the ramifications of using these methods for predicting solid-state properties far from equilibrium. Most importantly, we suggest caution in using DFT-D2 (or other coarse-grained parametrizations obtained from DFT-D2) for computing material properties under large stress/strain loads or for evaluating solid-state properties under extreme structural conditions.
Dear reviewer, the first paper discussed the work using first-principles approaches to understand the strength of these and other materials, the second paper discussed the work about density functional theory calculations can be used to understand atomistic details of structural mechanics and strength. But I haven't done first-principles calculations and simulations or density functional theory at all, I don't know what the first principles calculation are, and I don't even know what density functional theory is. I myself have limited ability to perform first-principles simulation calculations and density functional theory calculations. These first-principles simulation calculations and density functional theory calculations have some effect on my paper, so I want to add something about theoretical calculations of first-principles simulation calculations and density functional theory calculations into the introduction in my submission. First-principles simulation calculations and density functional theory are too hard for me, I have never been exposed to this knowledge and content of first-principles simulation calculations and density functional theory calculations, and I can only complete the suggestions given to me by the reviewers through adding the conception, theory, methods and application of the first-principles calculations and simulations or density functional theory into the introduction in my submission. I have some understanding about the first-principles simulation calculations and density functional theory calculations. So I decided to add the first-principles simulation calculations and density functional theory calculations into the introduction in my submission. So I decided to add the conception, theory, methods and application of the first-principles simulation calculations and density functional theory calculations into the introduction in my submission.
I can only mention the first-principles simulation calculations and density functional theory calculations in the introduction in my revised submission, such as “In particular, there has been much work using first-principles approaches to understand the strength of these and other materials. In particular, these prior studies showed that density functional theory calculations can be used to understand atomistic details of structural mechanics and strength. These approaches can reveal mechanisms for the strength of structures in a more unbiased way. The first principle calculation played a very important role in studying the relationship between microstructure analysis and mechanical property of the composites materials. At the same time, the density functional calculation played a very important role in studying the relationship between microstructure analysis and mechanical property of the composites materials. The first principle calculation and the density functional calculation could be completely understand the relationship between the microstructure characterization and mechanical property of the materials by using the calculation methods. So it was very important to using the first principle calculation and the density functional calculation to understand the relationship between the microstructure characterization and mechanical property of the composites materials.
According to the principle of interaction between atomic nucleus and electron and its basic motion law, using the principle of quantum mechanics, starting from the specific requirements, the algorithm to directly solve the Schrödinger equation after some approximation processing is customarily called the first principle.
Density functional theory (DFT) is a method for studying the electronic structure of multi-electron systems. Density functional theory has a wide range of applications in physics and chemistry, especially for studying the properties of molecules and condensed matter. It is one of the most commonly used methods in the fields of computational materials science and computational chemistry in condensed matter physics.
I added the above discussion about the first-principles simulation calculations and density functional theory calculations in the introduction in the revised submission. So I decided to add the conception, theory, methods and application of the first-principles simulation calculations and density functional theory calculations into the introduction in my submission. I cited the above two published paper as the references in the revised submission.
I can only explain the relationship between the microstructure and mechanical properties of composites materials by means of experiments. Now I explain the relationship between the microstructure and mechanical properties of composites materials by means of experiments.
1 The effects of graphite content on the B4C weight fraction and SiC weight fraction of the B4C-SiC composites
This response to reviewer’s comments describes the mixing ratio of B4C to graphite in the pressureless sintered B4C/C(graphite) composites. The mass fractions of graphite of the pressureless sintered B4C/C(graphite) composites are 10wt%, 20wt%, 30wt%, 40wt%, respectively, and the mass fractions of boron carbide of the pressureless sintered B4C/C(graphite) composites are 90wt%, 80wt%, 70wt%, 60wt%, respectively. That is to say, the mass ratio of B4C to graphite is 90:10, 80:20, 70:30, and 60:40, respectively. According to the mass ratio of B4C to graphite are 90:10, according to the percentage, B4C accounts for 90, and graphite accounts for 10 respectively. Assuming that B4C does not react with silicon, only graphite reacts with liquid silicon, then according to the reaction formula: set the mass fractions of SiC generated to be x, and calculate the value of x according to the following formula. The content of silicon did not calculate.
C+Si=SiC
, then x=33.33, so the amount of SiC generated is 33.33.
Then the weight fractions of B4C =;
Then the weight fractions of SiC =.
According to the mass ratio of B4C to graphite being 90:10 respectively, it can be seen that the weight fractions of B4C is 72.97%, and the weight fractions of SiC is 27.03%. So in the B4C-SiC composites, the weight fractions of B4C is 72.97%, and the weight fractions of SiC is 27.03%.
According to the mass ratio of B4C to graphite is 80:20, according to the percentage, B4C accounts for 80, and graphite accounts for 20 respectively. Assuming that B4C does not react with silicon, only graphite reacts with liquid silicon, then according to the reaction formula: set the mass fractions of SiC generated to be x, and calculate the value of x according to the following formula. The content of silicon did not calculate.
C+Si=SiC
, then x=66.67, so the amount of SiC generated is 66.67.
Then the weight fractions of B4C =;
Then the weight fractions of SiC =.
According to the mass ratio of B4C to graphite being 80:20 respectively, it can be seen that the weight fractions of B4C is 54.54%, and the weight fractions of SiC is 45.46%. So in the B4C-SiC composites, the weight fractions of B4C is 54.54%, and the weight fractions of SiC is 45.46%.
According to the mass ratio of B4C to graphite is 70:30, according to the percentage, B4C accounts for 70, and graphite accounts for 30 respectively. Assuming that B4C does not react with silicon, only graphite reacts with liquid silicon, then according to the reaction formula: set the mass fractions of SiC generated to be x, and calculate the value of x according to the following formula. The content of silicon did not calculate.
C+Si=SiC
, then x=100, so the amount of SiC generated is 100.
Then the weight fractions of B4C =;
Then the weight fractions of SiC =.
According to the mass ratio of B4C to graphite being 70:30 respectively, it can be seen that the weight fractions of B4C is 41.18%, and the weight fractions of SiC is 58.82%. So in the B4C-SiC composites, the weight fractions of B4C is 41.18%, and the weight fractions of SiC is 58.82%.
According to the mass ratio of B4C to graphite is 60:40, according to the percentage, B4C accounts for 60, and graphite accounts for 40 respectively. Assuming that B4C powder does not react with silicon, only graphite reacts with liquid silicon, then according to the reaction formula: set the mass fractions of SiC generated to be x, and calculate the value of x according to the following formula. The content of silicon did not calculate.
C+Si=SiC
, then x=133.33, so the amount of SiC generated is 133.33.
Then the weight fractions of B4C =;
Then the weight fractions of SiC =.
According to the mass ratio of B4C to graphite being 60:40 respectively, it can be seen that the weight fractions of B4C is 31.04%, and the weight fractions of SiC is 68.96%. So in the B4C-SiC composites, the weight fractions of B4C is 31.04%, and the weight fractions of SiC is 68.96%.
Therefore, the weight fractions of B4C in the B4C-SiC composites is 31.04%-72.97%, and the weight fractions of SiC in the B4C-SiC composites is 27.03%-68.96%. It can be seen that the weight fractions of B4C is relatively high, while the weight fractions of SiC is also relatively high.
Table.1shows the effects of graphite content on the B4C weight fraction and SiC weight fraction of the B4C-SiC composites. Table.1 showed that, with the increase of graphite content of the pressurless sintered B4C/C(graphite) composites, the weight fraction of B4C in the B4C-SiC composites decreased gradually, while the weight fraction of SiC in the B4C-SiC composites increased gradually. The B4C-SiC composites exhibited homogenous and compact microstructure with the increase of SiC content. The mechanical property of the B4C-SiC composites increased gradually with the increase of SiC content. The density and relative density, fracture strength and fracture toughness, Vickers hardness and elastic modulus the B4C-SiC composites increased gradually with the increase of SiC content. The wear resistance of the B4C-SiC composites increased gradually with the increase of SiC content.
Table.1. The effects of graphite content on the B4C weight fraction and SiC weight fraction of the B4C-SiC composites.
|
Ratio of B4C to graphite |
B4C weight fractions (%) |
SiC weight fractions (%) |
|
Ratio of B4C to graphite=90:10 |
72.97 |
27.03 |
|
Ratio of B4C to graphite=80:20 |
54.54 |
45.46 |
|
Ratio of B4C to graphite=70:30 |
41.18 |
58.82 |
|
Ratio of B4C to graphite=60:40 |
31.04 |
68.96 |
Fig.1 shows the effects of graphite content on the B4C weight fraction and SiC weight fraction of the B4C-SiC composites. Fig.1 showed that, with the increase of graphite content of the pressurless sintered B4C/C(graphite) composites, the weight fraction of B4C in the B4C-SiC composites decreased gradually, while the weight fraction of SiC in the B4C-SiC composites increased gradually. The B4C-SiC composites exhibited homogenous and compact microstructure with the increase of SiC content. The mechanical property of the B4C-SiC composites increased gradually with the increase of SiC content. The density and relative density, fracture strength and fracture toughness, Vickers hardness and elastic modulus the B4C-SiC composites increased gradually with the increase of SiC content. The wear resistance of the B4C-SiC composites increased gradually with the increase of SiC content.
Fig.1. The effects of graphite content on the B4C weight fraction and SiC weight fraction of the B4C-SiC composites.
2 The SEM micrographs of the B4C-SiC-Si composites fabricated by silicon infiltration process
Now the SEM micrographs of the B4C-SiC-Si composites were shown in order to prove the B4C-SiC-Si composites exhibited homogenous and compact microstructure.
Fig. 2 presents the SEM photographs representing the fracture surface of the silicon infiltrated B4C monolith (B4C-SiC-Si composites). The pressureless sintered B4C monolith was transformed into the B4C-SiC-Si composites through the silicon infiltration process. These SEM images depict that the silicon infiltrated products B4C-SiC-Si composites possessed a compact and homogenous microstructure. B4C-SiC-Si composites exhibited an exceptionally high relative density (98-99%), whereas their porosity was only about 1-2%. The silicon carbide produced following silicon infiltration as well as the residual silicon filled the pores of the composites, so the porosity of the silicon infiltrated specimens decreased remarkably, and the B4C-SiC-Si composites became more compact and dense.
Fig. 2. The SEM images representing the fracture surface of the silicon infiltrated B4C monolith (B4C-SiC-Si composites).
Fig. 3 presents the SEM images depicting the fracture surface of the silicon infiltrated B4C/C(graphite) composites (B4C-SiC-Si composites). Fig. 3(a-d) respectively depict the SEM micrographs of the (10wt%, 20wt%, 30wt%, 40wt%C(graphite)) silicon infiltrated B4C/C(graphite) composites. These SEM data present a compact and homogenous microstructure of the silicon infiltrated B4C-SiC-Si composites. The relative density of the B4C-SiC-Si composites was extremely high (98-99%), whereas the porosity of the B4C-SiC-Si composites was about 1-2%. The produced silicon carbide and residual silicon filled the pores, so the porosity of the silicon infiltrated specimens decreased remarkably, and the B4C-SiC-Si composites became more compact and dense.
Fig. 3. SEM images depicting the fracture surface of the silicon infiltrated B4C/C(graphite) composites(B4C-SiC-Si composites); Fig. 3(a-d) respectively depict the SEM micrographs of the (10wt%, 20wt%, 30wt%, 40wt%C(graphite)) silicon infiltrated B4C/C(graphite) composites.
3 The density and relative density of the pressureless sintered B4C/C(graphite) composites and the B4C-SiC-Si composites
The density of the pressureless sintered B4C/C(graphite) composites demonstrated a gradual decrease with the increase of graphite content. The density of the pressureless sintered B4C/C(graphite) composites was about 1.5-1.9g/cm3. The density of the B4C-SiC-Si composites progressively increased with the increase in the amount of added graphite in the B4C/C(graphite) composites. Since the density of silicon carbide is 3.21g/cm3 and that silicon is 2.32g/cm3, the density of produced B4C-SiC-Si composites increased progressively with the increase in the amount of silicon carbide produced. B4C-SiC-Si composites demonstrated a density value of 2.5855-2.9288g/cm3 which is substantially higher than the pressureless sintered B4C/C(graphite) composites.
The relative density of the pressureless sintered B4C/C(graphite) composites exhibited a gradual decline with the increase of graphite content. This is presumably because the B4C/C(graphite) composites were produced via pressureless sintering and therefore exhibited a porous microstructure, thereby causing their relative density to become extremely low, precisely 62-75%. The porosity of the pressureless sintered B4C/C(graphite) composites was about 25-38%. The relative density of the B4C-SiC-Si composites demonstrated a remarkable improvement in comparison with the pressureless sintered B4C/C(graphite) composites. In the B4C/C(graphite) composites, a gradual increase in the relative density was observed with the increase of graphite content. The density of silicon carbide is 3.21g/cm3, hence the relative density of the composites based on B4C-SiC-Si became higher than the relative density of the B4C/C(graphite) composites obtained through pressureless sintering. The silicon infiltrated products B4C-SiC-Si composites presented compact and homogenous microstructural features. The relative density of the B4C-SiC-Si composites was about 98-99%. The density and relative density of the B4C-SiC-Si composites showed a significant improvement in comparison to the pressureless sintered B4C/C(graphite) composites.
4 The fracture strength and fracture toughness of the pressureless sintered B4C/C(graphite) composites and the B4C-SiC-Si composites
A gradual decrease in the fracture strength of the pressureless sintered B4C/C(graphite) composites was observed with the increase in the amount of graphite. The fracture strength of the pressureless sintered B4C/C(graphite) composites lay within the 102-155MPa range and reduced remarkably with the increase of graphite content. The B4C-SiC-Si composites demonstrated a remarkable improvement in their fracture strength as compared to the pressureless sintered B4C/C(graphite) composites. The fracture strength of the B4C-SiC-Si composites increased progressively with the rise in graphite content. The relative density of the B4C-SiC-Si composites was found to be higher than 98%, the B4C-SiC-Si composites thus became more dense and compact and their fracture strength was higher than that of the pressureless sintered B4C/C(graphite) composites. B4C-SiC-Si composites have a fracture strength equivalent to 385-438MPa. In addition, the produced silicon carbide could also improve the fracture strength of the B4C-SiC-Si composites, with the fracture strength increasing gradually with the increase of silicon carbide content. As an obvious fact, silicon carbide content underwent a gradual increase with the increase of graphite content in the B4C/C(graphite) composites. All these factors contribute to a remarkable improvement in the fracture strength of the B4C-SiC-Si composites as compared to the pressureless sintered B4C/C(graphite) composites.
There was a gradual decline in the fracture toughness of the pressureless sintered B4C/C(graphite) composites with the increment in the amount of graphite. The observed fracture toughness of the pressureless sintered B4C/C(graphite) composites was about 1.568MPa·m1/2-2.35MPa·m1/2. The fracture toughness of the B4C-SiC-Si composites underwent a marked improvement when compared with the pressureless sintered B4C/C(graphite) composites, demonstrating a gradual increase with the increase in the amount of graphite. Silicon carbide and residual silicon settled into the pores of the composite's matrix, thereby decreasing its porosity and increasing the relative density up to > 98%, rendering the B4C-SiC-Si composites even more dense and compact. Consequently, the fracture toughness of the B4C-SiC-Si composites was recorded as 6.05-7.55MPa·m1/2 which is much higher in comparison to the fracture toughness of the pressureless sintered B4C/C(graphite) composites. The silicon carbide produced following silicon infiltration is also assumed to improve the fracture toughness of the B4C-SiC-Si composites, which underwent a progressive increase progressively with the increase of the silicon carbide content. The content of produced silicon carbide in turn gradually increased with the increase of the amount of graphite in the B4C/C(graphite) composites. Also, the relative density of the B4C-SiC-Si composites was extremely high, about 98-99%. Thus the fracture toughness and fracture strength of the B4C-SiC-Si composites were significantly enhanced as compared to the pressureless sintered B4C/C(graphite) composites.
5 The Vickers hardness and elastic modulus of the pressureless sintered B4C/C(graphite) composites and the B4C-SiC-Si composites
The Vickers hardness of the pressureless sintered B4C/C(graphite) composites demonstrated a progressive decline with the increase of graphite content. The Vickers hardness of the pressureless sintered B4C/C(graphite) composites was about 2.52-4.88GPa. The Vickers hardness of the B4C-SiC-Si composites was markedly enhanced when compared to the pressureless sintered B4C/C(graphite) composites and increased progressively with the increase in the amount of graphite. The Vickers hardness of boron carbide and silicon carbide is exceptionally high; 20GPa to be specific. The specimen surface of the B4C-SiC-Si composites primarily comprises boron carbide, silicon carbide, and residual silicon. As a consequence, the Vickers hardness of the B4C-SiC-Si composites following silicon infiltration is observed to be higher than pressureless sintered B4C/C(graphite) composites. B4C-SiC-Si composites demonstrated a Vickers hardness value as high as 22.885-26.768GPa. The Vickers hardness of the B4C-SiC-Si composites demonstrated a gradual increase with the increase in the amount of silicon carbide. The content of silicon carbide increased steadily with the increment in graphite content in the B4C/C(graphite) composites, thereby enhancing the Vickers hardness of the B4C-SiC-Si composites to a remarkable extent.
The elastic modulus of the pressureless sintered B4C/C(graphite) composites reduced gradually with the increase in the amount of graphite. The exact value of elastic modulus of the pressureless sintered B4C/C(graphite) composites was about 106-145GPa. The B4C-SiC-Si composites demonstrate an improved value of the elastic modulus. There was a gradual increase in the elastic modulus of the B4C-SiC-Si composites with the increase in the amount of graphite in the B4C/C(graphite) composites. The B4C-SiC-Si composites became more dense and compact with a relative density as high as > 98%. Due to their compact and homogenous microstructure, B4C-SiC-Si composites have an elastic modulus in the range of 385-410GPa, which is considerably higher than the elastic modulus of the pressureless sintered B4C/C(graphite) composites. Both boron carbide and silicon carbide have an exceptionally high value of elastic modulus. Thus, the silicon carbide generated following silicon infiltration could also contribute to improving the B4C-SiC-Si composites in terms of their elastic modulus. A gradual increase in the elastic modulus of the B4C-SiC-Si composites was evident with the increase of the content of silicon carbide, which increases linearly with the increase in graphite content. B4C-SiC-Si composites thus demonstrated an extremely high elastic modulus which was substantially enhanced as opposed to the pressureless sintered B4C/C(graphite) composites.
6 The wear resistance of the pressureless sintered B4C/C(graphite) composites and the B4C-SiC-Si composites
The wear resistance of the pressureless sintered B4C/C(graphite) composites reduced gradually with the increase in the amount of graphite presumably on account of the low Vickers hardness of the pressureless sintered B4C/C(graphite) composites. The wear resistance of the B4C-SiC-Si composites demonstrated a substantial improvement as compared to pressureless sintered B4C/C(graphite) composites which may be attributed to their high Vickers hardness. B4C-SiC-Si composites are thus rendered denser and more compact, their surface hardness and Vickers hardness were exceptionally high, hence the wear resistance of the B4C-SiC-Si composites was also extremely high. Silicon carbide and boron carbide along with the residual silicon existing on the silicon infiltrated specimen’s surface also contribute towards the remarkable improvement in the resistance of the B4C-SiC-Si composites towards wear. Thus, the wear resistance of the B4C-SiC-Si composites was quite high on account of their high Vickers hardness which is equivalent to 22.885-26.768GPa.
7 The relationship of microstructure and the improved mechanical property and wear resistance of the B4C-SiC-Si composites
The B4C-SiC-Si composites exhibited a homogenous and compact microstructure, an extremely high relative density, and remarkably low porosity. The microstructure investigation results showed that the silicon infiltrated specimens were primarily made up of boron carbide, silicon carbide, and residual silicon. There was a substantial improvement in the B4C-SiC-Si composites in terms of their mechanical property when compared with the pressureless sintered B4C/C(graphite) composites. The density and relative density, fracture strength and fracture toughness, elastic modulus, Vickers hardness and wear resistance of the B4C-SiC-Si composites were also significantly improved and exhibited a gradual increase with the increase in the amount of graphite in the composites. It can thus be concluded that the silicon infiltration process efficiently improves the mechanical property and wear resistance of the pressureless sintered B4C/C(graphite) composites and B4C-SiC-Si composites thus generated can find potentially useful applications due to their superior physical attributes.
So the B4C-SiC-Si composites exhibited homogenous and compact microstructure, and the B4C-SiC-Si exhibited high mechanical property and excellent wear resistance. The density and relative density, fracture strength and fracture toughness, Vickers hardness and elastic modulus of the B4C-SiC-Si composites were significantly improved in comparison with the pressureless sintered B4C/C(graphite) composites. The wear resistance of the B4C-SiC-Si composites were significantly improved in comparison with the pressureless sintered B4C/C(graphite) composites.
8 I decided to add the first-principles simulation calculations and density functional theory calculations into the introduction in my submission. I cited the above two published paper as the references in the revised submission.
So I can explain the relationship between the microstructure and mechanical property of the B4C-SiC-Si composites by experimental methods. I can only explain the relationship between the microstructure and mechanical properties of composites materials through experimental methods. I expected the reviewer can understand that I have never study the first-principles simulation calculations and density functional theory, my research is always experiments, not first-principles simulation calculations and density functional theory calculations. I have some understanding about the first-principles simulation calculations and density functional theory calculations. So I can add the first-principles simulation calculations and density functional theory calculations into the introduction in my submission. So I decided to add the first-principles simulation calculations and density functional theory calculations into the introduction in my submission. I think that the first-principles simulation calculations and density functional theory calculations were suitable for use in my submission. So I decided to add the first-principles simulation calculations and density functional theory calculations into the introduction in my submission. I cited the above two published paper as the references in the revised submission.
I added the above discussion about the first-principles simulation calculations and density functional theory calculations in the introduction in the revised submission. So I decided to add the conception, theory, methods and application of the first-principles simulation calculations and density functional theory calculations into the introduction in my submission. I cited the above two published paper as the references in the revised submission.
Reviewer 2 Report
Manuscript ID: materials-1760411
Type of manuscript: Article
Title: Investigation of Microstructural Features and Mechanical Characteristics of the Pressureless Sintered B4C/C(graphite) Composites and the B4C-SiC-Si Composites Fabricated by the Silicon Infiltration Process
Dear Ms. Derniza Cozorici
Assistant Editor of materials
This manuscript performs a study on the fabrication of the B4C/C(graphite) composites by using a pressureless sintering process. However, the fabricated composite exhibited extremely low mechanical characteristics. The Authors used from the liquid silicon infiltration technique to improve its mechanical properties. As mentioned in the literature, this article lacks the necessary innovation, but I am still waiting for the author's opinion for a final decision.
Comments:
1. First of all, it is necessary to determine the novelty of this article compared to the previous article of the author (Tao Jiang, Manman Han, Jia Fu. "Influence of
2. silicon infiltration on the wear and oxidation resistance of hot-pressed B4C/C(graphite) composites", Ceramics International, 2021).
3. In abstract: There are duplicate sentences in the text that should be removed such as:
“The pressureless sintered B4C/C(graphite) composites exhibited extremely low mechanical 11 characteristics.” and “The pressureless sintered B4C/C(graphite) composites exhibited a porous microstructure, an extremely low mechanical property, and low wear resistance.”
I think that the present manuscript has a significant overlap with the above article.
4. The above previous article has been published by three authors, while in the present manuscript, there is only one Author. Please explain about author statement.
5. References should be purposeful. Avoid block referencing (i.e. [1-10])
6. The characteristics of used equipment should be mentioned in “materials and methods” (i.e. bending and hardness testing and …)
7. There are many writing problems such as spacing in the text and therefore it is necessary to read the article carefully.
Author Response
Response to reviewer’s comments
Reviewer’s Comments:
- First of all, it is necessary to determine the novelty of this article compared to the previous article of the author (Tao Jiang, Manman Han, Jia Fu. "Influence of silicon infiltration on the wear and oxidation resistance of hot-pressed B4C/C(graphite) composites", Ceramics International, 2021).
Author reply:
PART ONE
My previous article published in Ceramics International was shown as follows:
Tao Jiang, Manman Han, Jia Fu. Influence of silicon infiltration on the wear and oxidation resistance of hot-pressed B4C/C(graphite) composites[J]. Ceramics International, 2021, 47(24): 34927-34939.
Abstract: B4C/C(graphite) composites were produced using Hot-pressing. However, the composites exhibited low surface hardness, low wear resistance as well as low oxidation resistance. The hot-pressed B4C/C(graphite) composites were processed via silicon infiltration procedure at 1550oC for 2h in the vacuum condition to refine the wear resistance, surface hardness, and oxidation resistance of the B4C/C(graphite) composites. The silicon infiltration process resulted in the fabrication of a surface layer based on silicon carbide and silicon upon the B4C/C(graphite) composites surface. The surface layer of the silicon infiltrated B4C/C(graphite) composites, primarily comprising silicon carbide and silicon, was examined for its phase composition and microstructure. The oxidation resistance, wear resistance, and surface hardness of B4C/C(graphite) composites subjected to silicon infiltration was additionally examined. The XRD results confirmed that silicon carbide and silicon-based layer existed on the surface of B4C/C(graphite) composites, which was produced as a result of the silicon infiltration process. The surface of silicon infiltrated B4C/C(graphite) composites exhibited a 300-400µm thick covering of silicon carbide and silicon. The deposited layer also exhibited a dense and compact microstructure. When compared to hot-pressed B4C/C(graphite) composites, the surface hardness and wear resistance of the composites resulting after silicon infiltration were significantly enhanced. Silicon infiltrated B4C/C(graphite) composites possess a surface hardness of 16-17GPa. The oxidation resistance of the silicon infiltrated B4C/C(graphite) composites was considerably enhanced as compared to the hot-pressed B4C/C(graphite) composites. So the silicon infiltrated B4C/C(graphite) composites exhibited high surface hardness, excellent wear resistance and excellent oxidation resistance in comparison with the hot-pressed B4C/C(graphite) composites.
1 Phase compositions of the silicon infiltrated B4C/C(graphite) composites
XRD patterns of the Si-infiltrated B4C/C(graphite) composites containing 10-40wt% of graphite showed some new set of peaks: first one at 20.78o, 26.67o, 35.60o, 41.38o, 59.97o, 71.77o, 75.49o, belonging to SiC, and the second one at 28.44o, 47.30o, 56.12o, 69.13o, and 76.37o, belonging to Si (see Fig. 1). Thus, B4C and graphite reacted with the liquid Si during the infiltration experiments forming SiC. Some unreacted Si remained in the B4C/C(graphite) composites as well, which translated into Si XRD peaks. Thus, after the infiltration experiments, a SiC/Si layer formed on the B4C/C(graphite) composites surface.
Fig. 1. XRD patterns of the B4C/C(graphite) composites subjected to the Si-infiltration process and containing (a) 10 wt%, (b) 20 wt%, (c) 30 wt% and (d) 40 wt% of graphite.
2 SEM characterization of the silicon infiltrated B4C/C(graphite) composites
SEM analysis of the Si-infiltrated B4C/C(graphite) composites containing 10-40wt% of graphite showed that the thickness of the SiC/Si coating generated on the sample surface following silicon infiltration lies in the 300-400μm range. The coating was tightly bonded to the matrix without an obvious interface. The SiC/Si coating also appeared to be very compact without the existence of any pores or defects (see Fig. 2).
Fig. 2. SEM micrographs of SiC/Si coating on the surface of the B4C/C(graphite) composites subjected to the liquid Si-infiltration process and containing (a) 10 wt%, (b) 20 wt%, (c) 30 wt%, and (d) 40 wt% of graphite.
3 The Vickers hardness of hot-pressed B4C/C(graphite) composites specimens and silicon infiltrated B4C/C(graphite) composites specimens
In the case of silicon infiltrated B4C/C(graphite) composites specimens, Fig. 3 depicts how the Vickers hardness is influenced by the net graphite content. It is evident that the Vickers hardness of specimens surface of B4C/C(graphite) composites subjected to silicon infiltration, manifested a significant improvement when compared against the original hot-pressed B4C/C(graphite) composites. The Vickers hardness values of the surface of the specimen of the silicon infiltrated B4C/C(graphite) composites were about 16-17GPa. The hot-pressed B4C/C(graphite) composites having (10wt%C(graphite)) had a Vickers hardness value equivalent to 10.266GPa, whereas the Vickers hardness of the B4C/C(graphite) composites(10wt%C(graphite)) upon silicon infiltration was approximately 17.5243GPa. With (20wt%C(graphite)), the Vickers hardness of the hot-pressed B4C/C(graphite) composites was while the Vickers hardness of the silicon infiltrated B4C/C(graphite) composites was 6.875GPa, and 17.0474GPa respectively. In the case of the hot-pressed B4C/C(graphite) composites having (30wt%C(graphite)), the Vickers hardness was 4.586GPa, while that of the silicon infiltrated B4C/C(graphite) composites having identical graphite content was 16.5898GPa. For (40wt%C(graphite)), the respective values were 3.587GPa and 16.1504GPa for hot-pressed B4C/C(graphite) composites with and without silicon infiltration. The above results demonstrated that the Vickers hardness of the surface of the specimen of the B4C/C(graphite) composites underwent a significant improvement following the silicon infiltration process. These results were because that the silicon carbide and silicon-based coating on the silicon infiltrated B4C/C(graphite) composites specimens exhibited high Vickers hardness, so the surface hardness of silicon infiltrated B4C/C(graphite) composites specimens was exceptionally enhanced. The silicon infiltrated B4C/C(graphite) composites, thus, exhibited a high net value for Vickers hardness as compared to the hot-pressed B4C/C(graphite) composites. Subsequently, the improved Vickers hardness of the silicon infiltrated B4C/C(graphite) composites led to an obvious improvement in their resistance to abrasion and wear.
Fig.3. The influence of the content of graphite on the Vickers hardness of the hot-pressed B4C/C(graphite) composites specimens and the silicon infiltrated B4C/C(graphite) composites specimens.
4 The wear resistance of hot-pressed B4C/C(graphite) composites specimens and silicon infiltrated B4C/C(graphite) composites specimens
The wear resistance of the hot-pressed B4C/C(graphite) composites and the silicon infiltrated B4C/C(graphite) composites were investigated by wear experiments. The horizontal coordinates was the wear time, the vertical coordinates was wear weight loss. Fig. 4 shows the effects of wear time on the wear weight loss of the hot-pressed B4C/C(graphite) composites and the silicon infiltrated B4C/C(graphite) composites. Fig.4(a) shows the effects of wear time on the wear weight loss of the hot-pressed B4C/C(graphite) composites. Fig.4(a) showed that the wear weight loss of the hot-pressed B4C/C(graphite) composites increased gradually with increasing of wear time. The wear weight loss of the hot-pressed B4C/C(graphite) composites increased gradually with increasing of graphite content. The B4C monolith exhibited the rather low wear weight loss curve due to the high Vickers hardness of B4C ceramics. The B4C/C(graphite) composites with the graphite content of 10wt%, 20wt%, 30wt%, 40wt% exhibited high wear weight loss curves due to the low Vickers hardness of the B4C/C(graphite) composites. For the Fig.4(a), from top to bottom, the wear weight loss curves were the B4C/C(40wt%graphite) composites, the B4C/C(30wt%graphite) composites, the B4C/C(20wt%graphite) composites, the B4C/C(10wt%graphite) composites, the B4C monolith. These results indicated that the wear weight loss was high, the wear resistance was very poor. So the hot-pressed B4C/C(graphite) composites exhibited low wear resistance. The wear resistance of the hot-pressed B4C/C(graphite) composites decreased gradually with the increasing of graphite content due to the low Vickers hardness of the B4C/C(graphite) composites. So the wear weight loss of the hot-pressed B4C/C(graphite) composites was serious, this result indicated that the wear resistance of the hot-pressed B4C/C(graphite) composites was extremely low. The hot-pressed B4C/C(graphite) composites exhibited extremely low wear resistance.
Fig.4(b) shows the effects of wear time on the wear weight loss of the silicon infiltrated B4C/C(graphite) composites. The wear weight loss of the silicon infiltrated B4C/C(graphite) composites increased gradually with increasing of wear time. The wear weight loss of the silicon infiltrated B4C/C(graphite) composites increased gradually with increasing of graphite content. The silicon infiltrated B4C/C(graphite) composites with the graphite content of 10wt%, 20wt%, 30wt%, 40wt% exhibited rather low wear weight loss curves due to the high Vickers hardness of the silicon infiltrated B4C/C(graphite) composites. For the Fig.4(b), from top to bottom, the wear weight loss curves were the silicon infiltrated B4C/C(40wt%graphite) composites, the silicon infiltrated B4C/C(30wt%graphite) composites, the silicon infiltrated B4C/C(20wt%graphite) composites, the silicon infiltrated B4C/C(10wt%graphite) composites. These results indicated that the wear weight loss was low, the wear resistance was very high. So the wear resistance of the silicon infiltrated B4C/C(graphite) composites exhibited high wear resistance due to the high Vickers hardness of the silicon infiltrated B4C/C(graphite) composites. The wear weight loss of the silicon infiltrated B4C/C(graphite) composites also increased gradually with the increase of wear time. The wear weight loss of the silicon infiltrated B4C/C(graphite) composites also increased gradually with the increase of graphite content. The wear resistance of the silicon infiltrated B4C/C(graphite) composites also decreased gradually with the increasing of graphite content. From the Fig.4(a) and Fig.4(b), it could be concluded that the wear weight loss of the silicon infiltrated B4C/C(graphite) composites was rather lower than the wear weight loss of the hot-pressed B4C/C(graphite) composites. These results indicated the wear resistance of the silicon infiltrated B4C/C(graphite) composites was remarkably improved in comparison with the hot-pressed B4C/C(graphite) composites. The silicon infiltrated B4C/C(graphite) composites exhibited excellent wear resistance due to the high Vickers hardness of the silicon infiltrated B4C/C(graphite) composites. So the silicon infiltrated B4C/C(graphite) composites exhibited excellent wear resistance in comparison with the hot-pressed B4C/C(graphite) composites.
Fig. 4 The effects of wear time on the wear weight loss of the hot-pressed B4C/C(graphite) composites and the silicon infiltrated B4C/C(graphite) composites; (a) hot-pressed B4C/C(graphite) composites; (b) silicon infiltrated B4C/C(graphite) composites.
According to the Fig.4, the line fitness was adopted to produce the wear weight loss curve, so the slope of line fitness was the ratio of wear weight loss to wear time. The slope was the wear loss rates. The relationship of wear weight loss and wear time can be expressed as the following equation:, was wear weight loss, R was wear loss rates, t was wear time. So the slope of wear weight loss curve was calculated according the following equations:, so the R was the wear loss rates, was wear weight loss, t was wear time.
Wear-resistant characteristics directly ascertain the safe usage duration of mechanical parts and equipment. Wear experiments were used to determine the wear resistance of the hot-pressed B4C/C(graphite) composites specimens and the B4C/C(graphite) composites specimens processed through silicon infiltration. The load force was 20N and the rotational speed was 220rpm. The rate of weight loss due to wearing, for the hot-pressed B4C/C(graphite) composites specimens and the silicon infiltrated B4C/C(graphite) composites specimens were measured. The wear loss rates were the ratio of weight loss (M) to wear time (t), the wear loss rate(R) was calculated as following equation: R=M/t. For the hot-pressed B4C/C(graphite) composites specimens and the silicon infiltrated B4C/C(graphite) composites specimens, the wear loss rates as a function of graphite content are shown in Fig. 5. Fig. 5 depicts that, for the hot-pressed B4C/C(graphite) composites, the wear loss rates rose gradually with the increment in graphite content. This result indicated that the weight loss of hot-pressed B4C/C(graphite) composites specimens was substantially increased with the increase of graphite content. The wear resistance of the hot-pressed B4C/C(graphite) composites dropped steadily as the graphite concentration increased, according to these findings. The low surface hardness of the hot-pressed B4C/C(graphite) composites can be blamed for the decreased wear resistance. Since the Vickers hardness of graphite materials was exceptionally low and had a softening impact on the Vickers hardness of boron carbide ceramics composites, the surface hardness of the hot-pressed B4C/C(graphite) composites was extremely low and consequently, the abrasive wear resistance of the hot-pressed B4C/C(graphite) composites loss lowers due to an exceptionally low value. Thus incorporating the graphite materials into the boron carbide ceramics matrix would remarkably decrease the wear resistance of the B4C/C(graphite) composites. At a graphite content greater than 20wt%, the wear loss rates of the B4C/C(graphite) composites were found to be quite high, thereby decreasing the wear resistance of the B4C/C(graphite) composites.
The wear loss rates of the B4C/C (graphite) composites specimens following silicon infiltration, as a function of graphite content are illustrated in Fig. 5. The rate of wear loss of the silicon infiltrated samples were estimated and Fig. 5 shows that with increasing graphite content, the wear loss rates of silicon penetrated B4C/C(graphite) composites underwent an increase. This finding suggested that as the graphite content increased, the weight loss of silicon infiltrated specimens happened only at a moderate rate. An obvious outcome of a reduction in the rate of wear loss is the significant improvement in the wear resistance of the silicon infiltrated B4C/C(graphite) composites as compared to the hot-pressed B4C/C(graphite) composites specimens. The above outcome implies that the silicon infiltration process led to a substantial enhancement in the wear resistance of the surface of the silicon infiltrated B4C/C(graphite) composites. Such an outcome is a result of the high magnitude of Vickers hardness and exceptionally good wear resistance of the silicon carbide and silicon-based layer on the surface of silicon infiltrated B4C/C(graphite) composites. Furthermore, the surface layer adhered and clung closely with the matrix of the composite; hence this observation of the remarkable improvement in the wear resistance of B4C/C(graphite) composites following silicon infiltration in comparison to the bare B4C/C(graphite) composites.
Fig. 5. The wear loss rates of the hot-pressed B4C/C(graphite) composites and the silicon infiltrated B4C/C(graphite) composites specimens as a function of the graphite content.
The main content of my previous article published in Ceramics International was shown as follows: Tao Jiang, Manman Han, Jia Fu. Influence of silicon infiltration on the wear and oxidation resistance of hot-pressed B4C/C(graphite) composites[J]. Ceramics International, 2021, 47(24): 34927-34939.
This paper mainly investigated the influence of silicon infiltration on the surface hardness, wear resistance and oxidation resistance of hot-pressed B4C/C(graphite) composites. (1) The B4C/C(graphite) composites were fabricated by hot-pressing process, the hot-pressed B4C/C(graphite) composites exhibited low surface hardness and low wear resistance and low oxidation resistance. (2) In order to improve the surface hardness, wear resistance and low oxidation resistance, the liquid silicon infiltration process was adopted to improve the surface hardness, wear resistance and low oxidation resistance of the hot-pressed B4C/C(graphite) composites. The liquid silicon could react with specimens surface of the B4C/C(graphite) composites, the liquid silicon could react with boron carbide to produce the silicon carbide, and the liquid silicon could react with graphite to produce the silicon carbide, and the residual silicon existed on the silicon infiltrated specimens, so the silicon carbide and silicon coating were fabricated on the surface of silicon infiltrated specimens. The thickness of silicon carbide and silicon coating was about 300-400µm. (3) Because the silicon carbide and silicon coating on the surface of silicon infiltrated B4C/C(graphite) composites exhibited high Vickers hardness, so the silicon infiltrated B4C/C(graphite) composites exhibited high Vickers hardness. The Vickers hardness of the silicon infiltrated B4C/C(graphite) composites were significantly improved in comparison with the hot-pressed B4C/C(graphite) composites. The wear resistance of the silicon infiltrated B4C/C(graphite) composites were significantly improved in comparison with the hot-pressed B4C/C(graphite) composites. Because the silicon carbide and silicon coating exhibited excellent oxidation resistance, so the oxidation resistance of the silicon infiltrated B4C/C(graphite) composites were significantly improved in comparison with the hot-pressed B4C/C(graphite) composites. The silicon infiltrated B4C/C(graphite) composites exhibited high Vickers hardness, excellent wear resistance and excellent oxidation resistance in comparison with the hot-pressed B4C/C(graphite) composites.
Now I provide some schematic diagram of the silicon carbide and silicon coating on the specimens surface of the silicon infiltrated B4C/C(graphite) composites.
Fig.6 shows the schematic diagram of the silicon carbide and silicon coating on the specimens surface of the silicon infiltrated B4C/C(graphite) composites. The liquid silicon could react with specimens surface of the B4C/C(graphite) composites, the liquid silicon could react with boron carbide to produce the silicon carbide, and the liquid silicon could react with graphite to produce the silicon carbide, and the residual silicon existed on the silicon infiltrated specimens, so the silicon carbide and silicon coating were fabricated on the surface of silicon infiltrated specimens. The thickness of silicon carbide and silicon coating was about 300-400µm. The silicon carbide and silicon coating bonded closely with the composites matrix.
Fig.6. The schematic diagram of the silicon carbide and silicon coating on the specimens surface of the silicon infiltrated B4C/C(graphite) composites.
Fig.7 shows the schematic diagram of the silicon carbide and silicon coating on the specimens surface of the silicon infiltrated B4C/C(graphite) composites. Fig.7 showed that the silicon carbide and silicon coating coated on the specimens surface of the B4C/C(graphite) composites. The specimens surface of the B4C/C(graphite) composites was coated by silicon carbide and silicon coating. The silicon carbide and silicon coating inside was the B4C/C(graphite) composites matrix.
Fig.7. The schematic diagram of the silicon carbide and silicon coating on the specimens surface of the silicon infiltrated B4C/C(graphite) composites.
Fig.8 shows the formation process of silicon carbide and silicon coating on the specimens surface of the hot-pressed B4C/C(graphite) composites fabricated by silicon infiltration process. Fig.8 showed that the B4C/C(graphite) composites were immerged in the liquid, so the liquid silicon could react with specimens surface of the B4C/C(graphite) composites, the liquid silicon could react with boron carbide to produce the silicon carbide, and the liquid silicon could react with graphite to produce the silicon carbide, and the residual silicon existed on surface of the silicon infiltrated specimens, so the silicon carbide and silicon coating were fabricated on the surface of silicon infiltrated B4C/C(graphite) composites specimens.
Fig.8. The formation process of silicon carbide and silicon coating on the specimens surface of the hot-pressed B4C/C(graphite) composites fabricated by silicon infiltration process.
Fig.9 shows the fabrication process and technology roadmap of silicon carbide and silicon coating on the specimens surface of the hot-pressed B4C/C(graphite) composites by silicon infiltration process. The B4C/C(graphite) composites were fabricated by hot-pressing process, the hot-pressed B4C/C(graphite) composites were fabricated into stripe samples, the B4C/C(graphite) composites stripe samples were processed by silicon infiltration process, the silicon carbide and silicon coating were produced on the surface of the silicon infiltrated B4C/C(graphite) composites. The phase composition of the silicon carbide and silicon coating was investigated. The microstructure of the silicon carbide and silicon coating was investigated. The Vickers hardness of the silicon infiltrated B4C/C(graphite) composites and the hot-pressed B4C/C(graphite) composites were investigated. The Wear resistance of the silicon infiltrated B4C/C(graphite) composites and the hot-pressed B4C/C(graphite) composites were investigated. The oxidation resistance of the silicon infiltrated B4C/C(graphite) composites and the hot-pressed B4C/C(graphite) composites were investigated.
Fig.9. The fabrication process and technology roadmap of silicon carbide and silicon coating on the specimens surface of the hot-pressed B4C/C(graphite) composites by silicon infiltration process.
PART TWO
But in my present submission to Materials, the paper title: “Investigation of Microstructural Features and Mechanical Characteristics of the Pressureless Sintered B4C/C(graphite) Composites and the B4C-SiC-Si Composites Fabricated by the Silicon Infiltration Process”. This present paper submitted to MATERIALS in 2022 was completely different with the above paper which published in CERAMICS INTERNATIONAL in 2021.
Abstract: The B4C/C(graphite) composites were fabricated by employing a pressureless sintering process. The pressureless sintered B4C/C(graphite) composites exhibited extremely low mechanical characteristics. The liquid silicon infiltration technique was employed for enhancing the mechanical property of B4C/C(graphite) composites. Since the porosity of B4C/C(graphite) composites was about 25%-38%, the liquid silicon was able to infiltrate into the interior composites, thereby reacting with B4C and graphite to generate silicon carbide. Thus, boron carbide, silicon carbide, and residual silicon were sintered together forming B4C-SiC-Si composites. The pressureless sintered B4C/C(graphite) composites were transformed into the B4C-SiC-Si composites following the silicon infiltration process. This work comprises an investigation of the microstructure, phase composition, and mechanical characteristics of the pressureless sintered B4C/C(graphite) composites and B4C-SiC-Si composites. The XRD data demonstrated that the pressureless sintered bulks were composed of the B4C phase and graphite phase. The pressureless sintered B4C/C(graphite) composites exhibited a porous microstructure, an extremely low mechanical property, and low wear resistance. The XRD data of the B4C-SiC-Si specimens showed that silicon infiltrated specimens comprised a B4C phase, SiC phase, and residual Si. The B4C-SiC-Si composites manifested a compact and homogenous microstructure. The mechanical property of the B4C-SiC-Si composites was substantially enhanced in comparison to the pressureless sintered B4C/C(graphite) composites. The density, relative density, fracture strength, fracture toughness, elastic modulus, and Vickers hardness of the B4C-SiC-Si composites were notably enhanced as compared to the pressureless sintered B4C/C(graphite) composites. The B4C-SiC-Si composites also manifested outstanding resistance to wear as a consequence of silicon infiltration. The B4C-SiC-Si composites demonstrated excellent wear resistance and superior mechanical characteristics.
1 The phase composition of the B4C-SiC-Si composites fabricated by the silicon infiltration process
1.1 The phase composition of the B4C-SiC-Si composites fabricated by silicon infiltration of pressureless sintered B4C monolith
Fig. 1 presents the XRD patterns corresponding to the silicon infiltrated B4C monolith (B4C-SiC-Si composites). The pressureless sintered B4C monolith was transformed into the B4C-SiC-Si composites through the silicon infiltration process. Fig. 1 shows that following the silicon infiltration process, the diffractive peaks of the B4C phase disappeared partially. Besides, some new diffraction peaks were apparent in the XRD pattern, which presumably arise due to the existence of silicon carbide and silicon phase. As seen before, the diffraction peaks of the B4C phase were observed at 19.71o, 22.02o, 23.50o, 31.90o, 34.95o, and 37.82o. The diffraction peaks of the SiC phase and Si phase were observed respectively at 20.78o, 26.67o, 35.60o, 41.38o, 59.97o, 71.77o, 75.49o, and 28.44o, 47.30o, 56.12o, 69.13o, 76.37o. The XRD results have also demonstrated the existence of a boron carbide phase, silicon carbide phase, and residual silicon phase on the surface of the specimen obtained after silicon infiltration. These XRD analysis results also confirmed our assumption that the liquid silicon upon reaction with the boron carbide produced silicon carbide, leaving the residual silicon on the surface of silicon infiltrated specimens. The residual boron carbide also existed in the matrix of the composite. The boron carbide, silicon carbide, and residual silicon were sintered together yielding composites of B4C-SiC-Si type. This way the pressureless sintered B4C monolith was transformed into the B4C-SiC-Si composites following the silicon infiltration process. The porosity of the pressureless sintered B4C monolith was about 20-30%. The apparent decrease in porosity can be ascribed to the silicon carbide and residual silicon (generated following silicon infiltration), filling in the pores of the matrix of the composite.
Fig. 1. The XRD patterns of the silicon infiltrated B4C monolith (B4C-SiC-Si composites).
1.2 The phase composition of the B4C-SiC-Si composites fabricated by silicon infiltration of pressureless sintered B4C/C(graphite) composites
Fig. 2 represents the X-ray diffraction patterns of the silicon infiltrated B4C/C(graphite) composites (B4C-SiC-Si composites). Fig. 2(a-d) respectively depict the XRD patterns of the (10wt%, 20wt%, 30wt%, 40wt%C(graphite)) silicon infiltrated B4C/C(graphite) composites. The pressureless sintered B4C/C(graphite) composites transformed into the B4C-SiC-Si composites following silicon infiltration. The silicon infiltrated B4C/C(graphite) composites have a precise chemical composition of the type B4C-SiC-Si. Fig. 2(a-d) demonstrated that the diffractive peaks of the B4C phase and graphite phase vanished following the infiltration of silicon. While the diffraction peaks of the B4C phase disappeared only partially, those of graphite vanished completely. The appearance of some new diffraction peaks in the data corresponds to the existence of the silicon carbide phase and silicon phase. The diffraction peaks of the B4C phase appear at 19.71o, 22.02o, 23.50o, 31.90o, 34.95o, and 37.82o, the diffraction peaks of the SiC phase can be observed at 20.78o, 26.67o, 35.60o, 41.38o, 59.97o, 71.77o, 75.49o, whereas those of the Si phase are apparent at 28.44o, 47.30o, 56.12o, 69.13o, 76.37o. The XRD data points towards the existence of the boron carbide phase, silicon carbide phase, and residual silicon phase upon the surface of the specimens obtained after silicon infiltration. The XRD analysis suggests the infiltration of liquid silicon into the inner composite matrix and subsequent reaction with boron carbide and graphite producing silicon carbide. Some residual silicon stayed on the specimen surface following silicon infiltration. The boron carbide, silicon carbide, and residual silicon were sintered together forming the B4C-SiC-Si composites. The porosity of the pressureless sintered B4C/C(graphite) composites was about 25-38%. The apparent decrease in the porosity is because produced silicon carbide and residual silicon filled the pores in the matrix of the composite during the silicon infiltration process.
Fig. 2. The XRD patterns of the silicon infiltrated B4C/C(graphite) composites(B4C-SiC-Si composites); Fig. 2(a-d) respectively depict the XRD patterns of the (10wt%, 20wt%, 30wt%, 40wt%C(graphite)) silicon infiltrated B4C/C(graphite) composites.
2 The microstructural features of the B4C-SiC-Si composites fabricated by the silicon infiltration process
2.1 The microstructural features of the B4C-SiC-Si composites fabricated by silicon infiltration of pressureless sintered B4C monolith
Fig. 3 presents the SEM photographs representing the fracture surface of the silicon infiltrated B4C monolith (B4C-SiC-Si composites). The pressureless sintered B4C monolith was transformed into the B4C-SiC-Si composites through the silicon infiltration process. These SEM images depict that the silicon infiltrated products B4C-SiC-Si composites possessed a compact and homogenous microstructure. The B4C particles and SiC particles were homogenously dispersed within the composite matrix. The mean particles size of B4C particles and SiC particles was about 4-5µm. The existence of some residual silicon in the matrix of the composite is also evident. Because the pressureless sintered B4C monolith was a porous material; the porosity being 20-30%, the liquid silicon could infiltrate into the interior of the composite matrix during the silicon infiltration process. Boron carbide reacted with liquid silicon producing silicon carbide particles, and the residual boron carbide, as well as silicon, existed in the composite's matrix. B4C-SiC-Si composites exhibited an exceptionally high relative density (98-99%), whereas their porosity was only about 1-2%. The silicon carbide produced following silicon infiltration as well as the residual silicon filled the pores of the composites, so the porosity of the silicon infiltrated specimens decreased remarkably, and the B4C-SiC-Si composites became more compact and dense.
Fig. 3. The SEM images representing the fracture surface of the silicon infiltrated B4C monolith (B4C-SiC-Si composites).
2.2 The microstructural features of the B4C-SiC-Si composites fabricated by silicon infiltration of pressureless sintered B4C/C(graphite) composites
Fig. 4 presents the SEM images depicting the fracture surface of the silicon infiltrated B4C/C(graphite) composites (B4C-SiC-Si composites). Fig. 4(a-d) respectively depict the SEM micrographs of the (10wt%, 20wt%, 30wt%, 40wt%C(graphite)) silicon infiltrated B4C/C(graphite) composites. These SEM data present a compact and homogenous microstructure of the silicon infiltrated B4C-SiC-Si composites. The B4C particles and SiC particles were distributed homogeneously in the composite's matrix, and some residual silicon also existed in the composite's matrix. The mean particles size of B4C particles and SiC particles was about 4-5µm. Because the pressureless sintered B4C/C(graphite) composites were porous, (the porosity of the pressureless sintered B4C/C(graphite) composites was about 25-38%), the liquid silicon could infiltrate into the inner composite's matrix during silicon infiltration. The reaction between boron carbide/graphite and liquid silicon produced silicon carbide particles, and the residual boron carbide and silicon existed in the composite's matrix. The relative density of the B4C-SiC-Si composites was extremely high (98-99%), whereas the porosity of the B4C-SiC-Si composites was about 1-2%. The produced silicon carbide and residual silicon filled the pores, so the porosity of the silicon infiltrated specimens decreased remarkably, and the B4C-SiC-Si composites became more compact and dense.
Fig. 4. SEM images depicting the fracture surface of the silicon infiltrated B4C/C(graphite) composites(B4C-SiC-Si composites); Fig. 4(a-d) respectively depict the SEM micrographs of the (10wt%, 20wt%, 30wt%, 40wt%C(graphite)) silicon infiltrated B4C/C(graphite) composites.
3 The mechanical property of the pressureless sintered B4C/C(graphite) composites and the B4C-SiC-Si composites
3.1 The density and relative density of the pressureless sintered B4C/C(graphite) composites and the B4C-SiC-Si composites
Fig. 5 represents the impact of graphite content on the density of the pressureless sintered B4C/C(graphite) composites and the B4C-SiC-Si composites. The density of the pressureless sintered B4C/C(graphite) composites demonstrated a gradual decrease with the increase of graphite content. The density of the pressureless sintered B4C monolith was relatively high, but the density of the pressureless sintered B4C/C(graphite) composites was rather low. The underlying reason for this observation is that B4C materials have a density of 2.52g/cm3, whereas graphite has a density of about 2.26g/cm3, hence with the increase in the amount of graphite, the density of the pressureless sintered B4C/C(graphite) composites demonstrated a gradual decline. The porous microstructure of the pressureless sintered specimens also contributed to the extremely low density. The density of the pressureless sintered B4C/C(graphite) composites was about 1.5-1.9g/cm3.
On the contrary, the density of the B4C-SiC-Si composites was phenomenally improved as compared to the pressureless sintered B4C/C(graphite) composites as depicted in Fig. 5. The density of the B4C-SiC-Si composites progressively increased with the increase in the amount of added graphite in the B4C/C(graphite) composites. This observation was mainly because the liquid silicon infiltrated into the interior of the pressureless sintered B4C/C(graphite) composites and reacted with B4C and graphite to produce silicon carbide. Thus, with the increment in the amount of graphite in the B4C/C(graphite) composites, the graphite completely reacted with liquid silicon to produce silicon carbide, thus markedly increasing the content of produced silicon carbide in the B4C-SiC-Si composites. Since the density of silicon carbide is 3.21g/cm3 and that silicon is 2.32g/cm3, the density of produced B4C-SiC-Si composites increased progressively with the increase in the amount of silicon carbide produced. On the whole, the density of the B4C-SiC-Si composites was higher than the density of the corresponding B4C/C(graphite) composites. B4C-SiC-Si composites demonstrated a density value of 2.5855-2.9288g/cm3 which is substantially higher than the pressureless sintered B4C/C(graphite) composites.
Fig. 5. The influence of graphite content on the density of the pressureless sintered B4C/C(graphite) composites and the B4C-SiC-Si composites.
Fig. 6 shows the impact of the amount of graphite on the relative density of the pressureless sintered B4C/C(graphite) composites and the B4C-SiC-Si composites. Fig. 6 illustrates that the relative density of the pressureless sintered B4C/C(graphite) composites exhibited a gradual decline with the increase of graphite content. The relative density of the pressureless sintered B4C monolith was relatively high, however, the relative density of the pressureless sintered B4C/C(graphite) composites was rather low. This is presumably because the B4C/C(graphite) composites were produced via pressureless sintering and therefore exhibited a porous microstructure, thereby causing their relative density to become extremely low, precisely 62-75%. The porosity of the pressureless sintered B4C/C(graphite) composites was about 25-38%.
Fig. 6 also depicts that the relative density of the B4C-SiC-Si composites demonstrated a remarkable improvement in comparison with the pressureless sintered B4C/C(graphite) composites. In the B4C/C(graphite) composites, a gradual increase in the relative density was observed with the increase of graphite content. With the infiltration of liquid silicon into the interior of pressureless sintered B4C/C(graphite) composites, the B4C and graphite reacted with liquid silicon to produce the silicon carbide. The silicon carbide thus produced as well as the residual silicon filled into the pores in the composite's matrix, thereby resulting in a remarkable decrease in the porosity of the silicon infiltrated specimens and a consequent increase in their relative density. The residual silicon, boron carbide, and silicon carbide were sintered into dense and compact bulks. The density of silicon carbide is 3.21g/cm3, hence the relative density of the composites based on B4C-SiC-Si became higher than the relative density of the B4C/C(graphite) composites obtained through pressureless sintering. The silicon infiltrated products B4C-SiC-Si composites presented compact and homogenous microstructural features. The relative density of the B4C-SiC-Si composites was about 98-99%. The density and relative density of the B4C-SiC-Si composites showed a significant improvement in comparison to the pressureless sintered B4C/C(graphite) composites.
Fig. 6. The impact of graphite content on the relative density of the pressureless sintered B4C/C(graphite) composites and the B4C-SiC-Si composites.
3.2 The fracture strength and fracture toughness of the pressureless sintered B4C/C(graphite) composites and the B4C-SiC-Si composites
The impact of the amount of graphite on the fracture strength of the pressureless sintered B4C/C(graphite) composites and the B4C-SiC-Si composites has been shown in Fig. 7. A gradual decrease in the fracture strength of the pressureless sintered B4C/C(graphite) composites was observed with the increase in the amount of graphite. While the fracture strength of the pressureless sintered B4C monolith was quite high, the pressureless sintered B4C/C(graphite) composites demonstrated a rather low fracture strength value. This is because the B4C/C(graphite) composites were fabricated by a pressureless sintering process, which imparted a porous microstructure, a low relative density (62-75%), and porosity of around 25-38% leaving them with an extremely low fracture strength. The fracture strength of the pressureless sintered B4C/C(graphite) composites lay within the 102-155MPa range and reduced remarkably with the increase of graphite content.
Fig. 7 depicts that the B4C-SiC-Si composites demonstrated a remarkable improvement in their fracture strength as compared to the pressureless sintered B4C/C(graphite) composites. The fracture strength of the B4C-SiC-Si composites increased progressively with the rise in graphite content. It occurs because liquid silicon infiltrated into the interior of pressureless sintered B4C/C(graphite) composites, and B4C and graphite reacted with liquid silicon producing silicon carbide. The resultant silicon carbide and the residual silicon and boron carbide were sintered together fabricating B4C-SiC-Si composites. The silicon carbide and residual silicon settled within the pores in the composite's matrix, thereby causing a remarkable decrease in the porosity of the silicon infiltrated specimens and a corresponding increase in the relative density of the silicon infiltrated specimens. The relative density of the B4C-SiC-Si composites was found to be higher than 98%, the B4C-SiC-Si composites thus became more dense and compact and their fracture strength was higher than that of the pressureless sintered B4C/C(graphite) composites. B4C-SiC-Si composites have a fracture strength equivalent to 385-438MPa. In addition, the produced silicon carbide could also improve the fracture strength of the B4C-SiC-Si composites, with the fracture strength increasing gradually with the increase of silicon carbide content. As an obvious fact, silicon carbide content underwent a gradual increase with the increase of graphite content in the B4C/C(graphite) composites. All these factors contribute to a remarkable improvement in the fracture strength of the B4C-SiC-Si composites as compared to the pressureless sintered B4C/C(graphite) composites.
Fig. 7. The impact of graphite content on the fracture strength of the pressureless sintered B4C/C(graphite) composites and the B4C-SiC-Si composites.
The fracture toughness of the pressureless sintered B4C/C(graphite) composites and the B4C-SiC-Si composite is also affected by the amount of graphite as shown in Fig. 8. As obvious from the figure, there was a gradual decline in the fracture toughness of the pressureless sintered B4C/C(graphite) composites with the increment in the amount of graphite. The fracture toughness of the pressureless sintered B4C/C(graphite) composites was rather low as compared to that of the pressureless sintered B4C monolith, presumably because the B4C/C(graphite) composites were produced via pressureless sintering, and exhibited a porous microstructure. Also, the B4C/C(graphite) composites obtained via pressureless sintering had a rather low relative density and the porosity was 25%-38%. It was also observed that the pressureless sintered B4C/C(graphite) composites demonstrated a remarkable decrease in their fracture toughness with the increase of graphite content. This is an outcome of the inherently low fracture toughness of graphitic materials. The observed fracture toughness of the pressureless sintered B4C/C(graphite) composites was about 1.568MPa·m1/2-2.35MPa·m1/2.
It is also evident from Fig. 8 that the fracture toughness of the B4C-SiC-Si composites underwent a marked improvement when compared with the pressureless sintered B4C/C(graphite) composites, demonstrating a gradual increase with the increase in the amount of graphite. This can be attributed to the infiltration of liquid silicon into the interior pressureless sintered B4C/C(graphite) composites, and the subsequent reaction with B4C and graphite producing silicon carbide. Silicon carbide and residual silicon settled into the pores of the composite's matrix, thereby decreasing its porosity and increasing the relative density up to > 98%, rendering the B4C-SiC-Si composites even more dense and compact. Consequently, the fracture toughness of the B4C-SiC-Si composites was recorded as 6.05-7.55MPa·m1/2 which is much higher in comparison to the fracture toughness of the pressureless sintered B4C/C(graphite) composites.
The silicon carbide produced following silicon infiltration is also assumed to improve the fracture toughness of the B4C-SiC-Si composites, which underwent a progressive increase progressively with the increase of the silicon carbide content. The content of produced silicon carbide in turn gradually increased with the increase of the amount of graphite in the B4C/C(graphite) composites. Also, the relative density of the B4C-SiC-Si composites was extremely high, about 98-99%. Thus the fracture toughness and fracture strength of the B4C-SiC-Si composites were significantly enhanced as compared to the pressureless sintered B4C/C(graphite) composites. While the pressureless sintered B4C/C(graphite) composites were soft and brittle and could be easily broken, the B4C-SiC-Si composites were compact and hard and exhibited extremely high fracture strength and high fracture toughness.
Fig. 8. The effects of graphite content on the fracture toughness of the pressureless sintered B4C/C(graphite) composites and the B4C-SiC-Si composites.
3.3 The Vickers hardness and elastic modulus of the pressureless sintered B4C/C(graphite) composites and the B4C-SiC-Si composites
The impact of graphite content on the Vickers hardness of the pressureless sintered B4C/C(graphite) composites and the B4C-SiC-Si composites has been shown in Fig. 9. The Vickers hardness of the pressureless sintered B4C/C(graphite) composites demonstrated a progressive decline with the increase of graphite content. While the Vickers hardness of the pressureless sintered B4C monolith was relatively high, its value was rather low for the pressureless sintered B4C/C(graphite) composites. Owing to their fabrication from the pressureless sintering process, the B4C/C(graphite) composites exhibited porous microstructure and have a low relative density (62-75%) and a porosity of about 25-38%. It has been known that the Vickers hardness of graphite is rather low; the laminated structured graphite materials have a softening effect on the ceramics composites. Consequently, the Vickers hardness of the pressureless sintered B4C/C(graphite) composites exhibited a gradual decrease with the increase in the amount of graphite. The pressureless sintered B4C/C(graphite) composites were soft and very brittle and could be easily broken. The Vickers hardness of the pressureless sintered B4C/C(graphite) composites was about 2.52-4.88GPa.
Fig. 9 also shows that the Vickers hardness of the B4C-SiC-Si composites was markedly enhanced when compared to the pressureless sintered B4C/C(graphite) composites and increased progressively with the increase in the amount of graphite. This can be ascribed to the infiltration of liquid silicon into the interior pressureless sintered B4C/C(graphite) composites, and the subsequent reaction with B4C and graphite, producing silicon carbide. Silicon carbide and residual silicon settle into the pores of the composite's matrix, thereby decreasing its porosity and increasing the relative density up to > 98%, rendering the B4C-SiC-Si composites even more dense and compact and subsequently increasing their Vickers’s hardness.
Besides, the Vickers hardness of boron carbide and silicon carbide is exceptionally high; 20GPa to be specific. The specimen surface of the B4C-SiC-Si composites primarily comprises boron carbide, silicon carbide, and residual silicon. As a consequence, the Vickers hardness of the B4C-SiC-Si composites following silicon infiltration is observed to be higher than pressureless sintered B4C/C(graphite) composites. B4C-SiC-Si composites demonstrated a Vickers hardness value as high as 22.885-26.768GPa.
Moreover, the Vickers hardness of the B4C-SiC-Si composites demonstrated a gradual increase with the increase in the amount of silicon carbide. The content of silicon carbide increased steadily with the increment in graphite content in the B4C/C(graphite) composites, thereby enhancing the Vickers hardness of the B4C-SiC-Si composites to a remarkable extent. A high value of the Vickers hardness, in turn, guarantees an enhancement in the resistance of the B4C-SiC-Si composites to wear. Thus, the B4C-SiC-Si composites exhibit an outstanding wear resistance.
Fig. 9. The effects of graphite content on the Vickers hardness of the pressureless sintered B4C/C(graphite) composites and the B4C-SiC-Si composites.
Figure 10 is a representation of the variation in elastic modulus of the pressureless sintered B4C/C(graphite) composites and the B4C-SiC-Si composites as a function of graphite content. The elastic modulus of the pressureless sintered B4C/C(graphite) composites reduced gradually with the increase in the amount of graphite. While the elastic modulus of the pressureless sintered B4C monolith was relatively high, that of the pressureless sintered B4C/C(graphite) composites was rather low. This is a consequence of porous microstructure, low relative density (62%-75%), and a high porosity (25%-38%), of the pressureless sintered B4C/C(graphite) composites.
The elastic modulus of boron carbide is high, and the elastic modulus of graphite was rather low. This leads to the introduction of softness and brittleness in the pressureless sintered B4C/C(graphite) composites. The pressureless sintered B4C/C(graphite) composites are highly porous and exhibit a low elastic modulus, with the elastic modulus gradually declining with the increment in graphite content. The exact value of elastic modulus of the pressureless sintered B4C/C(graphite) composites was about 106-145GPa as illustrated in Fig. 10.
When compared with the pressureless sintered B4C/C(graphite) composites, the B4C-SiC-Si composites demonstrate an improved value of the elastic modulus. There was a gradual increase in the elastic modulus of the B4C-SiC-Si composites with the increase in the amount of graphite in the B4C/C(graphite) composites as shown in Fig. 10. This, again, is a consequence of liquid silicon seeping into the interior of pressureless sintered B4C/C(graphite) composites. Thus the B4C and graphite undergo a chemical reaction with liquid silicon, generating silicon carbide. The silicon carbide thus generated takes up the porous spaces in the composite's matrix along with the residual silicon, decreasing its porosity and increasing the relative density. The B4C-SiC-Si composites became more dense and compact with a relative density as high as > 98%. Due to their compact and homogenous microstructure, B4C-SiC-Si composites have an elastic modulus in the range of 385-410GPa, which is considerably higher than the elastic modulus of the pressureless sintered B4C/C(graphite) composites.
Both boron carbide and silicon carbide have an exceptionally high value of elastic modulus. Thus, the silicon carbide generated following silicon infiltration could also contribute to improving the B4C-SiC-Si composites in terms of their elastic modulus. A gradual increase in the elastic modulus of the B4C-SiC-Si composites was evident with the increase of the content of silicon carbide, which increases linearly with the increase in graphite content. B4C-SiC-Si composites thus demonstrated an extremely high elastic modulus which was substantially enhanced as opposed to the pressureless sintered B4C/C(graphite) composites.
Fig. 10. The influence of graphite content on the elastic modulus of the pressureless sintered B4C/C(graphite) composites and the B4C-SiC-Si composites.
4 The wear resistance of the pressureless sintered B4C/C(graphite) composites and the B4C-SiC-Si composites
The pressureless sintered B4C/C(graphite) composites and the B4C-SiC-Si composites were examined for their wear resistance with the help of wear experiments. The vertical coordinate was wear weight loss, whereas the wear time represents the horizontal coordinate. Fig.11(a) presents the influence of wear time on the weight loss of the pressureless sintered B4C/C(graphite) composites due to wear. The wear weight loss of the pressureless sintered B4C/C(graphite) composites manifested a progressive increase with the increment in wear time. Also, the extent of wear weight loss increased with the gradual rise in graphite content. The pressureless sintered B4C monolith gave a rather low wear weight loss curve owing to the relatively high inherent Vickers hardness of B4C ceramics. Whereas the composites with the graphite content of 10wt%, 20wt%, 30wt%, and 40wt% exhibited higher wear weight loss curves which may be attributed to the low value of Vickers hardness of the pressureless sintered B4C/C(graphite) composites. The weight loss curves in Fig.11(a), illustrate that the highest weight loss existed for sintered B4C/C(40wt%graphite) composites while the lowest was observed for the B4C monolith. The findings suggested that for the pressureless sintered B4C/C(graphite) composites, the wear resistance was very poor and the wear weight loss was high. The wear resistance of the pressureless sintered B4C/C(graphite) composites reduced gradually with the increase in the amount of graphite presumably on account of the low Vickers hardness of the pressureless sintered B4C/C(graphite) composites.
The influence of wear time on the wear weight loss of the B4C-SiC-Si composites has been shown in Fig.11(b). Contrary to the B4C/C(graphite) composites, the B4C-SiC-Si composites demonstrated an increase in wear weight loss with the increase in wear time. The wear weight loss of the B4C-SiC-Si composites manifested a gradual decrease with the increment in graphite content in the corresponding B4C/C(graphite) composites. The B4C-SiC-Si composites with the graphite content of 10wt%, 20wt%, 30wt%, and 40wt% in the B4C/C(graphite) composites gave lower wear weight loss curves presumably on account of the high Vickers hardness of the B4C-SiC-Si composites. As shown in Fig.11(b), the highest weight loss existed for silicon infiltrated B4C monolith, whereas the lowest was observed for the silicon infiltrated B4C/C(40wt%graphite) composites. When the wear weight loss was low, the wear resistance was very high. Thus, the wear resistance of the B4C-SiC-Si composites was quite high on account of their high Vickers hardness which is equivalent to 22.885-26.768GPa.
From Fig.11(a) and Fig.11(b), it could also be deduced that the wear weight loss of the B4C-SiC-Si composites was relatively lower in comparison to the wear weight loss of the pressureless sintered B4C/C(graphite) composites. This implies that the wear resistance of the B4C-SiC-Si composites demonstrated a substantial improvement as compared to pressureless sintered B4C/C(graphite) composites which may be attributed to their high Vickers hardness.
Fig. 11. (a) the influence of wear time on the wear weight loss of the pressureless sintered B4C/C(graphite) composites; (b) the influence of wear time on the wear weight loss of the B4C-SiC-Si composites.
The wear weight loss curve was created using line fitness in Fig.11, and the slope of line fitness is the ratio of wear weight loss to wear time. The slope thus represents the wear loss rates. The following equation can be used to illustrate the relationship between wear weight decrease and wear time: R=M/t, where R represents wear loss rates, M represents wear weight loss and t represents wear duration. As a result, the slope of the wear weight loss curve was computed using the formulae R=M/t.
The wear resistance of pressureless sintered specimens and silicon infiltrated specimens were measured by friction wear experiments. A ratio of wear weight loss to wear time is referred to as the wear loss rate. Fig. 12 presents the influence of graphite content on the wear loss rates of the pressureless sintered B4C/C(graphite) composites and the B4C-SiC-Si composites. It can be seen that the wear loss rates of the pressureless sintered B4C/C(graphite) composites gradually increased with the increase in the amount of graphite. This indicated that the wear resistance of the pressureless sintered B4C/C(graphite) composites underwent a gradual decrease with the increase in the amount of graphite. A possible explanation of this observation is that the B4C/C(graphite) composites were fabricated via a pressureless sintering method and have a low relative density and a porosity ranging between 25%-38%. The Vickers hardness of the pressureless sintered B4C/C(graphite) composites reduced progressively with the increase in the amount of graphite. The surface hardness of the pressureless sintered B4C/C(graphite) composites was also exceptionally low. All these factors contribute to the reduced wear resistance of the pressureless sintered B4C/C(graphite) composites, which manifested a noticeable decline with the increase in the amount of graphite.
As regards the B4C-SiC-Si composites, their wear loss rates underwent a gradual decrease with the rise in graphite content as depicted in Fig. 12. The outcome revealed that the B4C-SiC-Si composites specimen’s weight demonstrated a steady decrease with the increase in the amount of graphite in the B4C/C(graphite) composites. Fig. 12 also revealed that the wear resistance of the B4C-SiC-Si composites was substantially enhanced when compared to the pressureless sintered B4C/C(graphite) composites, indicating that the silicon infiltrated specimen’s weight underwent a gradual decrement with the increment in the amount of graphite in the B4C/C(graphite) composites. The loss in weight observed for the silicon infiltrated specimens was exceptionally minimal. The B4C-SiC-Si composites manifested remarkable resistance to wear. The underlying phenomenon, in this case, is the infiltration of liquid silicon into the interior of the composites and subsequent reaction with B4C and graphite yielding silicon carbide. Silicon carbide and residual silicon ultimately occupy the pores in the composite's matrix, decreasing its porosity and increasing its relative density. B4C-SiC-Si composites are thus rendered denser and more compact, their surface hardness and Vickers hardness were exceptionally high, hence the wear resistance of the B4C-SiC-Si composites was also extremely high. Silicon carbide and boron carbide along with the residual silicon existing on the silicon infiltrated specimen’s surface also contribute towards the remarkable improvement in the resistance of the B4C-SiC-Si composites towards wear.
Fig. 12. The influence of graphite content on the wear loss rates of the pressureless sintered B4C/C(graphite) composites and the B4C-SiC-Si composites.
PART THREE
The difference between my previous paper published in CERAMICS INTERNATIONAL and my present paper submitted to Materials
[1]My previous paper published in CERAMICS INTERNATIONAL: Tao Jiang, Manman Han, Jia Fu. Influence of silicon infiltration on the wear and oxidation resistance of hot-pressed B4C/C(graphite) composites[J]. Ceramics International, 2021, 47(24): 34927-34939.
[2]My present paper submitted to Materials: Tao Jiang. Investigation of Microstructural Features and Mechanical Characteristics of the Pressureless Sintered B4C/C(graphite) Composites and the B4C-SiC-Si Composites Fabricated by the Silicon Infiltration Process[J]. Materials, 2022.
Now I explain the difference between my previous paper published in CERAMICS INTERNATIONAL and my present paper submitted to Materials from following aspects:
(1) My previous paper published in CERAMICS INTERNATIONAL: Influence of silicon infiltration on the wear and oxidation resistance of hot-pressed B4C/C(graphite) composites[J]. Ceramics International, 2021, 47(24): 34927-34939.
This paper mainly investigated the influence of silicon infiltration on the surface hardness, wear resistance and oxidation resistance of hot-pressed B4C/C(graphite) composites. (1) The B4C/C(graphite) composites were fabricated by hot-pressing process, the hot-pressed B4C/C(graphite) composites exhibited low surface hardness and low wear resistance and low oxidation resistance. (2) In order to improve the surface hardness, wear resistance and low oxidation resistance, the liquid silicon infiltration process was adopted to improve the surface hardness, wear resistance and low oxidation resistance of the hot-pressed B4C/C(graphite) composites. The liquid silicon could react with specimens surface of the B4C/C(graphite) composites, the liquid silicon could react with boron carbide to produce the silicon carbide, and the liquid silicon could react with graphite to produce the silicon carbide, and the residual silicon existed on the silicon infiltrated specimens, so the silicon carbide and silicon coating were fabricated on the surface of silicon infiltrated specimens. The thickness of silicon carbide and silicon coating was about 300-400µm. (3) Because the silicon carbide and silicon coating exhibited high Vickers hardness, so the silicon infiltrated B4C/C(graphite) composites exhibited high Vickers hardness. The Vickers hardness of the silicon infiltrated B4C/C(graphite) composites were significantly improved in comparison with the hot-pressed B4C/C(graphite) composites. The wear resistance of the silicon infiltrated B4C/C(graphite) composites were significantly improved in comparison with the hot-pressed B4C/C(graphite) composites. Because the silicon carbide and silicon coating exhibited excellent oxidation resistance, so the oxidation resistance of the silicon infiltrated B4C/C(graphite) composites were significantly improved in comparison with the hot-pressed B4C/C(graphite) composites. The silicon infiltrated B4C/C(graphite) composites exhibited high Vickers hardness, excellent wear resistance and excellent oxidation resistance in comparison with the hot-pressed B4C/C(graphite) composites.
This previous paper mainly investigated the B4C/C(graphite) composites were fabricated by hot-pressing process, and the silicon carbide and silicon coating were obtained on the specimens surface of hot-pressed B4C/C(graphite) composites by silicon infiltration process. This previous paper mainly investigated the phase composition and microstructure of produced silicon carbide and silicon coating on the surface of silicon infiltrated B4C/C(graphite) composites. This previous paper mainly investigated the Vickers hardness, wear resistance and oxidation resistance of silicon infiltrated B4C/C(graphite) composites.
(2) My present paper submitted to Materials: Investigation of Microstructural Features and Mechanical Characteristics of the Pressureless Sintered B4C/C(graphite) Composites and the B4C-SiC-Si Composites Fabricated by the Silicon Infiltration Process[J]. Materials, 2022.
This present paper mainly investigated the Microstructural Features and Mechanical Characteristics of the Pressureless Sintered B4C/C(graphite) Composites and the B4C-SiC-Si Composites Fabricated by the Silicon Infiltration Process. (1) The B4C/C(graphite) composites were fabricated by pressureless sintering process, the pressureless sintered B4C/C(graphite) composites exhibited porous microstructure, and the pressureless sintered B4C/C(graphite) composites exhibited low mechanical property and low wear resistance. (2) In order to improve the mechanical property and wear resistance, the liquid silicon infiltration process was adopted to improve the mechanical property and wear resistance of the pressureless sintered B4C/C(graphite) composites. The liquid silicon could react with the interior B4C/C(graphite) composites, the liquid silicon could react with boron carbide to produce the silicon carbide, and the liquid silicon could react with graphite to produce the silicon carbide, and the residual silicon existed in the silicon infiltrated specimens, so the boron carbide, silicon carbide and silicon were sintered together to fabricate into the B4C-SiC-Si composites. The B4C-SiC-Si composites exhibited homogenous and compact microstructure. The relative density of the B4C-SiC-Si composites was about 98-99%. (3) The pressureless sintered B4C/C(graphite) composites reacted with liquid silicon and transformed into the B4C-SiC-Si composites by silicon infiltration process. The phase composition, microstructure and mechanical property of the B4C-SiC-Si composites fabricated by silicon infiltration process were investigated. The XRD patterns results showed that there existed the B4C phase, SiC phase and residual Si phase on the surface of silicon infiltrated specimens. The silicon infiltrated products B4C-SiC-Si composites was mainly composed of boron carbide, silicon carbide and residual silicon. The silicon infiltrated products B4C-SiC-Si composites exhibited homogenous and compact microstructure. The B4C-SiC-Si composites exhibited extremely high relative density and the porosity of the silicon infiltrated specimens was extremely low. The microstructure investigation results showed that the silicon infiltrated specimens was mainly composed of boron carbide, silicon carbide and residual silicon. The mechanical property of the B4C-SiC-Si composites were significantly improved in comparison with the pressureless sintered B4C/C(graphite) composites. The density and relative density of the B4C-SiC-Si composites were significantly improved in comparison with the pressureless sintered B4C/C(graphite) composites. The fracture strength and fracture toughness of the B4C-SiC-Si composites were significantly improved in comparison with the pressureless sintered B4C/C(graphite) composites. The elastic modulus and Vickers hardness of the B4C-SiC-Si composites were significantly improved in comparison with the pressureless sintered B4C/C(graphite) composites. The density and relative density of the B4C-SiC-Si composites increased gradually with the increase of graphite content in the B4C/C(graphite) composites. The fracture strength and fracture toughness of the B4C-SiC-Si composites increased gradually with the increase of graphite content in the B4C/C(graphite) composites. The elastic modulus and Vickers hardness of the B4C-SiC-Si composites increased gradually with the increase of graphite content in the B4C/C(graphite) composites. The wear resistance of the B4C-SiC-Si composites were remarkably improved in comparison with the pressureless sintered B4C/C(graphite) composites. The wear resistance of the B4C-SiC-Si composites increased gradually with the increase of graphite content in the B4C/C(graphite) composites. So the silicon infiltrated products B4C-SiC-Si composites exhibited high mechanical property and excellent wear resistance. The B4C-SiC-Si composites exhibited high mechanical property due to the homogenous and compact microstructure. The B4C-SiC-Si composites exhibited excellent wear resistance due to the extremely high Vickers hardness of the B4C-SiC-Si composites. The mechanical property and wear resistance of the B4C-SiC-Si composites were significantly improved in comparison with the pressureless sintered B4C/C(graphite) composites.
This paper mainly investigated the B4C/C(graphite) composites were fabricated by pressureless sintering process, and the pressureless sintered B4C/C(graphite) composites were transformed into the B4C-SiC-Si composites by silicon infiltration process. This paper mainly investigated the phase composition and microstructure of the B4C-SiC-Si composites fabricated by silicon infiltration process. This present paper mainly investigated the mechanical property and wear resistance of the B4C-SiC-Si composites fabricated by silicon infiltration process. This paper mainly investigated the density and relative density, fracture strength and fracture toughness, Vickers hardness and elastic modulus of the B4C-SiC-Si composites fabricated by silicon infiltration process. This paper mainly investigated wear resistance of the B4C-SiC-Si composites fabricated by silicon infiltration process.
So My present paper submitted to Materials: “Investigation of Microstructural Features and Mechanical Characteristics of the Pressureless Sintered B4C/C(graphite) Composites and the B4C-SiC-Si Composites Fabricated by the Silicon Infiltration Process” have innovative, and have novelty in comparison with my previous paper published in CERAMICS INTERNATIONAL.
Now I provide some schematic diagram of formation process of the B4C-SiC-Si composites fabricated by silicon infiltration process of the pressureless sintered B4C/C(graphite) composites.
Fig.13 shows the formation process of the B4C-SiC-Si composites fabricated by silicon infiltration process of the pressureless sintered B4C/C(graphite) composites. The B4C/C(graphite) composites specimens were fabricated by pressureless sintering process. The pressureless sintered specimens were fabricated into the stripe samples. The B4C/C(graphite) composites fabricated by pressureless sintering process with the graphite content of 10wt%, 20wt%, 30wt%, 40wt% were processed by the silicon infiltration process. These stripe specimens were put into the graphite crucible and put the graphite crucible into vacuum sintering furnace. These stripe samples were covered with silicon powders. The silicon infiltration process was performed at 1550oC for 2h in vacuum condition. Because that the melting point of silicon was about 1410oC, therefore, the solid silicon was completely transformed into the liquid silicon at 1550oC during the silicon infiltration process. During the silicon infiltration process, the stripe samples were immerged in the liquid silicon, the porosity of pressureless sintered specimens was about 25%-38%, so that the liquid silicon would be infiltrated into the interior composites specimens from these pores. So the liquid silicon reacted with B4C and graphite to produce the silicon carbide, the boron carbide partially reacted with the liquid silicon to produce the silicon carbide, and the graphite completely reacted with the liquid silicon to produced the silicon carbide, so the residual boron carbide existed in the composites matrix, and residual silicon existed in the composites matrix, the boron carbide, silicon carbide and residual silicon were sintered together and formed the B4C-SiC-Si composites, so the B4C-SiC-Si composites were fabricated by silicon infiltration process. So the liquid silicon would be infiltrated into the specimens. So the pressureless sintered B4C/C(graphite) composites transformed into the B4C-SiC-Si composites by silicon infiltration process. So the silicon infiltrated product was the B4C-SiC-Si composites.
Fig.13. The formation process of the B4C-SiC-Si composites fabricated by silicon infiltration process of the pressureless sintered B4C/C(graphite) composites.
Fig.14 shows the fabrication process and technology roadmap of the B4C-SiC-Si composites by silicon infiltration process of the pressureless sintered B4C/C(graphite) composites. The B4C/C(graphite) composites were fabricated by pressureless sintering process, the pressureless sintered B4C/C(graphite) composites were fabricated into stripe samples, the pressureless sintered B4C/C(graphite) composites stripe samples were processed by silicon infiltration process, the pressureless sintered B4C/C(graphite) composites transformed into the B4C-SiC-Si composites by silicon infiltration process. The phase composition of the B4C-SiC-Si composites and the pressureless sintered B4C/C(graphite) composites were investigated. The microstructure of the B4C-SiC-Si composites and the pressureless sintered B4C/C(graphite) composites were investigated. The density and relative density, fracture strength and fracture toughness, Vickers hardness and elastic modulus of the B4C-SiC-Si composites and the pressureless sintered B4C/C(graphite) composites were investigated. The wear resistance of the B4C-SiC-Si composites and the pressureless sintered B4C/C(graphite) composites were investigated.
Fig.14. The fabrication process and technology roadmap of the B4C-SiC-Si composites by silicon infiltration process of the pressureless sintered B4C/C(graphite) composites.
- In abstract: There are duplicate sentences in the text that should be removed such as:“The pressureless sintered B4C/C(graphite) composites exhibited extremely low mechanical 11 characteristics.” and “The pressureless sintered B4C/C(graphite) composites exhibited a porous microstructure, an extremely low mechanical property, and low wear resistance.”
Author reply: The duplicate sentences in the text that have been removed from the manuscript. The abstract became concise and clear. The abstract of this manuscript was revised carefully.
- I think that the present manuscript has a significant overlap with the above article.
Author reply: This present manuscript has not overlap with the above article. This present manuscript was completely different with the above article. The phase composition of this present manuscript and the above article were completely different. The microstructure of this present manuscript and the above article were completely different. The mechanical property of this present manuscript and the above article were completely different. The mechanical property of this present manuscript mainly investigated the density and relative density, fracture strength and fracture toughness, Vickers hardness and elastic modulus of the B4C-SiC-Si composites. The wear resistance of this present manuscript mainly investigated the wear resistance of the B4C-SiC-Si composites. The mechanical property of the above article mainly investigated the Vickers hardness and wear resistance of the silicon carbide and silicon coating on the specimens surface of silicon infiltrated B4C/C(graphite) composites. The oxidation resistance of the above article mainly investigated the oxidation resistance of the silicon carbide and silicon coating on the specimens surface of silicon infiltrated B4C/C(graphite) composites. The research target of this present manuscript and the above article were completely different. The research target of this present manuscript were pressureless sintered B4C/C(graphite) composites and the silicon infiltrated products B4C-SiC-Si composites. The research target of the above article were hot-pressed B4C/C(graphite) composites and the silicon carbide and silicon coating on the specimens surface of the silicon infiltrated B4C/C(graphite) composites. The above article mainly investigated the phase composition and microstructure of the silicon carbide and silicon coating on the specimens surface of the silicon infiltrated B4C/C(graphite) composites. The above article mainly investigated the Vickers hardness, wear resistance and oxidation resistance of the silicon carbide and silicon coating on the specimens surface of the silicon infiltrated B4C/C(graphite) composites. This present manuscript mainly investigated the phase composition and microstructure of the pressureless sintered B4C/C(graphite) composites and the silicon infiltrated products B4C-SiC-Si composites. This present manuscript mainly investigated the density and relative density, fracture strength and fracture toughness, Vickers hardness, elastic modulus of the pressureless sintered B4C/C(graphite) composites and the silicon infiltrated products B4C-SiC-Si composites. This present manuscript mainly investigated the wear resistance of the pressureless sintered B4C/C(graphite) composites and the silicon infiltrated products B4C-SiC-Si composites. So this present manuscript has not overlap with the above article. The data and pictures of this present paper were completely different with the above article.
If the reviewer carefully read the above articles published in Ceramics International and this present manuscript submitted to Materials, you would find that the present manuscript was completely different with the above articles published in Ceramics International. This present manuscript has not overlap with the above article. So I submitted the present manuscript to the journal of Materials.
(1) My previous paper published in CERAMICS INTERNATIONAL: Influence of silicon infiltration on the wear and oxidation resistance of hot-pressed B4C/C(graphite) composites[J]. Ceramics International, 2021, 47(24): 34927-34939.
This previous paper mainly investigated the influence of silicon infiltration on the surface hardness, wear resistance and oxidation resistance of hot-pressed B4C/C(graphite) composites. (1) The B4C/C(graphite) composites were fabricated by hot-pressing process, the hot-pressed B4C/C(graphite) composites exhibited low surface hardness and low wear resistance and low oxidation resistance. (2) In order to improve the surface hardness, wear resistance and low oxidation resistance, the liquid silicon infiltration process was adopted to improve the surface hardness, wear resistance and low oxidation resistance of the hot-pressed B4C/C(graphite) composites. The liquid silicon could react with specimens surface of the B4C/C(graphite) composites, the liquid silicon could react with boron carbide to produce the silicon carbide, and the liquid silicon could react with graphite to produce the silicon carbide, and the residual silicon existed on the silicon infiltrated specimens, so the silicon carbide and silicon coating were fabricated on the surface of silicon infiltrated specimens. The thickness of silicon carbide and silicon coating was about 300-400µm. (3) Because the silicon carbide and silicon coating exhibited high Vickers hardness, so the silicon infiltrated B4C/C(graphite) composites exhibited high Vickers hardness. The Vickers hardness of the silicon infiltrated B4C/C(graphite) composites were significantly improved in comparison with the hot-pressed B4C/C(graphite) composites. The wear resistance of the silicon infiltrated B4C/C(graphite) composites were significantly improved in comparison with the hot-pressed B4C/C(graphite) composites. Because the silicon carbide and silicon coating exhibited excellent oxidation resistance, so the oxidation resistance of the silicon infiltrated B4C/C(graphite) composites were significantly improved in comparison with the hot-pressed B4C/C(graphite) composites. The silicon infiltrated B4C/C(graphite) composites exhibited high Vickers hardness, excellent wear resistance and excellent oxidation resistance in comparison with the hot-pressed B4C/C(graphite) composites.
This previous paper mainly investigated the B4C/C(graphite) composites were fabricated by hot-pressing process, and the silicon carbide and silicon coating were obtained on the specimens surface of hot-pressed B4C/C(graphite) composites by silicon infiltration process. This previous paper mainly investigated the phase composition and microstructure of produced silicon carbide and silicon coating on the surface of silicon infiltrated B4C/C(graphite) composites. This previous paper mainly investigated the Vickers hardness, wear resistance and oxidation resistance of silicon infiltrated B4C/C(graphite) composites.
(2) My present paper submitted to Materials: Investigation of Microstructural Features and Mechanical Characteristics of the Pressureless Sintered B4C/C(graphite) Composites and the B4C-SiC-Si Composites Fabricated by the Silicon Infiltration Process[J]. Materials, 2022.
This present paper mainly investigated the Microstructural Features and Mechanical Characteristics of the Pressureless Sintered B4C/C(graphite) Composites and the B4C-SiC-Si Composites Fabricated by the Silicon Infiltration Process. (1) The B4C/C(graphite) composites were fabricated by pressureless sintering process, the pressureless sintered B4C/C(graphite) composites exhibited porous microstructure, and the pressureless sintered B4C/C(graphite) composites exhibited low mechanical property and low wear resistance. (2) In order to improve the mechanical property and wear resistance, the liquid silicon infiltration process was adopted to improve the mechanical property and wear resistance of the pressureless sintered B4C/C(graphite) composites. The liquid silicon could react with the interior B4C/C(graphite) composites, the liquid silicon could react with boron carbide to produce the silicon carbide, and the liquid silicon could react with graphite to produce the silicon carbide, and the residual silicon existed in the silicon infiltrated specimens, so the boron carbide, silicon carbide and silicon were sintered together to fabricate into the B4C-SiC-Si composites. The B4C-SiC-Si composites exhibited homogenous and compact microstructure. The relative density of the B4C-SiC-Si composites was about 98-99%. (3) The pressureless sintered B4C/C(graphite) composites reacted with liquid silicon and transformed into the B4C-SiC-Si composites by silicon infiltration process. The phase composition, microstructure and mechanical property of the B4C-SiC-Si composites fabricated by silicon infiltration process were investigated. The XRD patterns results showed that there existed the B4C phase, SiC phase and residual Si phase on the surface of silicon infiltrated specimens. The silicon infiltrated products B4C-SiC-Si composites was mainly composed of boron carbide, silicon carbide and residual silicon. The silicon infiltrated products B4C-SiC-Si composites exhibited homogenous and compact microstructure. The B4C-SiC-Si composites exhibited extremely high relative density and the porosity of the silicon infiltrated specimens was extremely low. The microstructure investigation results showed that the silicon infiltrated specimens was mainly composed of boron carbide, silicon carbide and residual silicon. The mechanical property of the B4C-SiC-Si composites were significantly improved in comparison with the pressureless sintered B4C/C(graphite) composites. The density and relative density of the B4C-SiC-Si composites were significantly improved in comparison with the pressureless sintered B4C/C(graphite) composites. The fracture strength and fracture toughness of the B4C-SiC-Si composites were significantly improved in comparison with the pressureless sintered B4C/C(graphite) composites. The elastic modulus and Vickers hardness of the B4C-SiC-Si composites were significantly improved in comparison with the pressureless sintered B4C/C(graphite) composites. The density and relative density of the B4C-SiC-Si composites increased gradually with the increase of graphite content in the B4C/C(graphite) composites. The fracture strength and fracture toughness of the B4C-SiC-Si composites increased gradually with the increase of graphite content in the B4C/C(graphite) composites. The elastic modulus and Vickers hardness of the B4C-SiC-Si composites increased gradually with the increase of graphite content in the B4C/C(graphite) composites. The wear resistance of the B4C-SiC-Si composites were remarkably improved in comparison with the pressureless sintered B4C/C(graphite) composites. The wear resistance of the B4C-SiC-Si composites increased gradually with the increase of graphite content in the B4C/C(graphite) composites. So the silicon infiltrated products B4C-SiC-Si composites exhibited high mechanical property and excellent wear resistance. The B4C-SiC-Si composites exhibited high mechanical property due to the homogenous and compact microstructure. The B4C-SiC-Si composites exhibited excellent wear resistance due to the extremely high Vickers hardness of the B4C-SiC-Si composites. The mechanical property and wear resistance of the B4C-SiC-Si composites were significantly improved in comparison with the pressureless sintered B4C/C(graphite) composites.
This present paper mainly investigated the B4C/C(graphite) composites were fabricated by pressureless sintering process, and the pressureless sintered B4C/C(graphite) composites were transformed into the B4C-SiC-Si composites fabricated by silicon infiltration process. This present paper mainly investigated the phase composition and microstructure of the B4C-SiC-Si composites fabricated by silicon infiltration process. This present paper mainly investigated the mechanical property and wear resistance of the B4C-SiC-Si composites fabricated by silicon infiltration process. This present paper mainly investigated the density and relative density, fracture strength and fracture toughness, Vickers hardness and elastic modulus of the B4C-SiC-Si composites fabricated by silicon infiltration process. This present paper mainly investigated wear resistance of the B4C-SiC-Si composites fabricated by silicon infiltration process.
So My present paper submitted to Materials: “Investigation of Microstructural Features and Mechanical Characteristics of the Pressureless Sintered B4C/C(graphite) Composites and the B4C-SiC-Si Composites Fabricated by the Silicon Infiltration Process” were completely different with my previous paper published in CERAMICS INTERNATIONAL. This present manuscript has not overlap with the above article. The data and pictures of this present paper were completely different with the above article. So I submitted the present manuscript to the Journal of Materials.
- The above previous article has been published by three authors, while in the present manuscript, there is only one Author. Please explain about author statement.
Author reply: The above previous article has been published in Ceramics International were completely different with the present manuscript submitted to Journal of Materials. The above previous article has been published by three authors, the first author is me, this paper is my own personal scientific research achievement, and the second author is my postgraduate student, in order to let this student graduate as a postgraduate, I added the student into the author. The third author is a teacher in my department. The second author and the third author have little contribution to my above previous article, so I added the second author and the third author into my above previous article as author. So the above previous article has been published by three authors.
But in the present manuscript, there is only one Author, the author is myself, because the present manuscript belongs to my own personal scientific research achievement, so I write myself as the author. My postgraduate student has graduated from Xi’an Shiyou University, so I cannot set my postgraduate student as the author. The third author in the above previous article has leave from Xi’an Shiyou University and went to the other school, so I cannot set this person as the author. The second author and third author in the above previous article have no contribution to my present manuscript, so I did not added the second author and the third author into the present manuscript as author. Because the present manuscript belongs to my own personal scientific research achievement, I did not want to add any other person as the authors. So in the present manuscript, there is only one Author, the author is myself. I did not want to add any other person as the authors in my present manuscript. I obtained the XRD date and SEM micrographs by myself through the experiments, and I obtained the mechanical property date and wear resistance data by myself through experiments. This present manuscript belongs to my own personal scientific research achievement, so I write myself as the author. I did not want to add any other person as the authors in my present manuscript. So in the present manuscript, there is only one Author, the author is myself. This present manuscript belongs to my own personal scientific research achievement, so I write myself as the author. I did not want to add any other person as the authors in my present manuscript. This present manuscript belongs to my own personal scientific research achievement, so I write myself as the author.
- References should be purposeful. Avoid block referencing (i.e. [1-10])
Author reply: the references were cited properly, and the block referencing was removed, all the references were cited correctly.
- The characteristics of used equipment should be mentioned in “materials and methods” (i.e. bending and hardness testing and …)
Author reply: The composites were evaluated for their fracture strength by a three-point bending test utilizing specimens of sizes 3mm×4mm×30mm, the span was kept at 16mm and the speed of the crosshead was 0.5mm/min(Testing machine: Instron 1195). The fracture toughness of the composites was estimated using a single edge notch beam (SENB) using samples of 3mm×4mm×30mm, the respective depth and width of the notch were 1.5mm and 0.2mm, the span was 16mm whereas the crosshead speed was 0.05mm/min(Testing machine: Instron 1195). The elastic modulus of the pressureless sintered B4C/C(graphite) composites as well as B4C-SiC-Si composites obtained via silicon infiltration was estimated using a three-point bending test with the dimensions of the specimen being 3mm×4mm×30mm, the span and the crosshead speed being 16mm and 0.5mm/min respectively(Testing machine: Instron 1195). Vickers hardness meter equipped with a load force of 49N and holding time of the 20s was employed to test the composites for their Vickers’s hardness (HV-5 Vickers hardness meter). The wear experiment was preformed at MM-W1B friction and wear tester. Friction wear tests were carried out to measure the wear resistance of the pressureless sintered B4C/C(graphite) composites and B4C-SiC-Si composites obtained via silicon infiltration, wherein the load force was 20N and the rotational speed was 220rpm. The wear loss rates of pressureless sintered specimens were also measured, the wear loss rate (R) was the ratio of wear weight loss(M) to wear time(t), thus the wear loss rate is computed using the following equation: R=M/t. The mechanical property and wear resistance of the pressureless sintered B4C monolith were measured similarly using the methods employed for the pressureless sintered B4C/C(graphite) composites.
- There are many writing problems such as spacing in the text and therefore it is necessary to read the article carefully.
Author reply: the writing problems such as spacing in the text have been revised according to the template. The author has been carefully read the article thoroughly.

Round 2
Reviewer 2 Report
This manuscript can be accepted in the present form.